# Conformal Inference under High-Dimensional Covariate Shifts via Likelihood-Ratio Regularization

**Sunay Joshi***
University of Pennsylvania
sunayj@sas.upenn.edu

**Shayan Kiyani***
University of Pennsylvania
shayank@seas.upenn.edu

**George Pappas**
University of Pennsylvania
pappasg@seas.upenn.edu

**Edgar Dobriban**
University of Pennsylvania
dobriban@wharton.upenn.edu

**Hamed Hassani**
University of Pennsylvania
hassani@seas.upenn.edu

## Abstract

We consider the problem of conformal prediction under covariate shift. Given labeled data from a source domain and unlabeled data from a covariate shifted target domain, we seek to construct prediction sets with valid marginal coverage in the target domain. Most existing methods require estimating the unknown likelihood ratio function, which can be prohibitive for high-dimensional data such as images. To address this challenge, we introduce the likelihood ratio regularized quantile regression (LR-QR) algorithm, which combines the pinball loss with a novel choice of regularization in order to construct a threshold function without directly estimating the unknown likelihood ratio. We show that the LR-QR method has coverage at the desired level in the target domain, up to a small error term that we can control. Our proofs draw on a novel analysis of coverage via stability bounds from learning theory. Our experiments demonstrate that the LR-QR algorithm outperforms existing methods on high-dimensional prediction tasks, including a regression task for the Communities and Crime dataset, an image classification task from the WILDS repository, and an LLM question-answering task on the MMLU benchmark.

## 1 Introduction

Conformal prediction is a framework to construct distribution-free prediction sets for black-box predictive models [e.g., 45, 60, 61, etc]. Given a pretrained prediction model $f : \mathcal{X} \to \mathcal{Y}$ mapping features $x \in \mathcal{X}$ to labels $y \in \mathcal{Y}$, and $n_1$ calibration datapoints $(X_i, Y_i) : i \in [n_1]$ sampled i.i.d. from a calibration distribution $\mathbb{P}_1$, we seek to construct a prediction set $C(X_{\text{test}}) \subseteq \mathcal{Y}$ for test features $X_{\text{test}}$ sampled from a marginal test distribution $\mathbb{P}_{2,X}$. We aim to cover the true label $Y_{\text{test}}$ with probability at least $1 - \alpha$ for some $\alpha \in (0, 1)$: that is, $\mathbb{P}(Y_{\text{test}} \in C(X_{\text{test}})) \geqslant 1 - \alpha$. The left-hand side of this inequality is the marginal coverage of the prediction set $C$, averaged over the randomness of both the calibration datapoints and the test datapoint $(X_{\text{test}}, Y_{\text{test}}) \sim \mathbb{P}_2$. In the case that the calibration and test distributions coincide ($\mathbb{P}_1 = \mathbb{P}_2$), there are numerous conformal prediction algorithms that

---

*Equal Contribution. Correspondence to: sunayj@sas.upenn.edu, shayank@seas.upenn.edu.

39th Conference on Neural Information Processing Systems (NeurIPS 2025).

construct distribution-free prediction sets with valid marginal coverage; e.g., split and full conformal prediction [e.g., 29, 36].

However, in practice, it is often the case that test data is sampled from a different distribution than calibration data. This general phenomenon is known as distribution shift [e.g., 41, 52]. One particularly common type of distribution shift is covariate shift [50], where the conditional distribution of $Y|X$ stays fixed, but the marginal distribution of features changes from calibration to test time. For instance, in the setting of image classification for autonomous vehicles, the calibration and test data might have been collected under different weather conditions [25, 65]. Under covariate shift, ordinary conformal prediction algorithms may lose coverage.

Recently, a number of methods have been proposed to adapt conformal prediction to covariate shift, e.g., in [15, 19, 37, 38, 40, 55, 64]. Most existing approaches attempt to estimate the likelihood ratio function $r : \mathcal{X} \to \mathbb{R}$, defined as $r(x) = (d\mathbb{P}_{2,X}/d\mathbb{P}_{1,X})(x)$ for all $x \in \mathcal{X}$, where $\mathbb{P}_{i,X}$ denotes the marginal distribution of $\mathbb{P}_i$ over the features. One can construct an estimate $\hat{r}$ of the likelihood ratio if one has access to additional unlabeled datapoints sampled i.i.d from the test distribution $\mathbb{P}_2$. Methods for likelihood ratio estimation include using Bayes' rule to express it as a ratio of classifiers [12, 40] and domain adaptation [13, 37]. However, such estimates may be inaccurate for high-dimensional data. This error propagates to the coverage of the resulting conformal predictor, and the prediction sets may no longer attain the nominal coverage level. Thus, it is natural to ask the following question: *Can one design a conformal prediction algorithm that attains valid coverage in the target domain, without estimating the entire function $r$?*

In this paper, we present a method that answers this question in the affirmative. We construct our prediction sets by introducing and solving a regularized quantile regression problem, which combines the pinball loss with a novel data-dependent regularization term that can be computed from one-dimensional projections of the likelihood ratio $r$. Crucially, the objective function can be estimated at the parametric rate, with only a mild dependence on the dimension of the feature space. This regularization is specifically chosen to ensure that the first order conditions of the pinball loss lead to coverage at test-time. Geometrically, the regularization aligns the selected threshold function with the true likelihood ratio $r$. The resulting method, which we call likelihood ratio regularized quantile regression (LR-QR), outperforms existing methods on high-dimensional datasets with covariate shift.

Our contributions include the following:

- We propose the LR-QR algorithm, which constructs a conformal predictor that adapts to covariate shift without directly estimating the likelihood ratio.

- We show that the minimizers of the population LR-QR objective have coverage in the test distribution. We also show that the minimizers of the empirical LR-QR objective lead to coverage up to a small error term that we can control. Our theoretical results draw on a novel analysis of coverage via stability bounds from learning theory.

- We demonstrate the effectiveness of the LR-QR algorithm on high-dimensional datasets under covariate shift, including the Communities and Crime dataset, the RxRx1 dataset from the WILDS repository, and the MMLU benchmark. Here, we crucially leverage our theory by choosing the regularization parameter proportional to the theoretically optimal value.

## 1.1 Related work

Here we only list prior work most closely related to our method; we provide more references in Appendix C. The early ideas of conformal prediction were developed in Saunders et al. [45], Vovk et al. [61]. With the rise of machine learning, conformal prediction has emerged as a widely used framework for constructing prediction sets [e.g., 36, 58, 59]. Classical conformal prediction guarantees validity when the calibration and test data are drawn from the same distribution. In contrast, when there is distribution shift between the calibration and test data [e.g., 5, 41, 50, 52, 54], coverage may not hold. Covariate shift is a type of dataset shift that arises in many settings, e.g., when predicting disease risk for individuals whose features may evolve over time, while the outcome distribution conditioned on the features remains stable [41].

Numerous works have addressed conformal prediction under distribution shift [37, 38, 40, 51, 55]. For example, Tibshirani et al. [55] investigated conformal prediction under covariate shift, assuming the likelihood ratio between source and target covariates is known. Lei and Candès [32] allowed

the likelihood ratio to be estimated. Park et al. [37] developed prediction sets with a calibration-set conditional (PAC) property under covariate shift. Qiu et al. [40], Yang et al. [64] developed prediction sets with asymptotic coverage that are doubly robust in the sense that their coverage error is bounded by the product of the estimation errors of the quantile function of the score and the likelihood ratio. Cauchois et al. [7] construct prediction sets based on distributionally robust optimization. In contrast, our method entirely avoids estimating the likelihood ratio function.

To achieve coverage under a predefined set of covariate shifts, Gibbs et al. [15] develop an approach based on minimizing the quantile loss over a linear hypothesis class. We build on their quantile regression framework, but develop a novel regularization scheme that allows us to effectively optimize over a data-driven class, adaptive to the unknown shift $r$. A similar regularization is used in [66], which performs supervised learning under covariate shift by minimizing an upper bound of the test risk. However, when one sets the loss function to equal the pinball loss, minimizing the objective in [66] is not guaranteed to provide coverage at test-time, whereas our construction has asymptotically valid coverage.

## 2 Problem Formulation

**Preliminaries and notations.** For $\alpha \in (0, 1)$, the quantile loss $\ell_\alpha$ is defined for all $c, s \in \mathbb{R}$ as

$$\ell_\alpha(c, s) := \begin{cases} (1 - \alpha)(s - c) \text{ if } s \geq c, \\ \alpha(c - s) \text{ if } s < c. \end{cases} \tag{1}$$

Let the *source* or *calibration* distribution be denoted $\mathbb{P}_1 = \mathbb{P}_{1,X} \times \mathbb{P}_{Y|X}$, where $\mathbb{P}_{1,X}$ is the marginal distribution of $\mathbb{P}_1$ over features. Let the *target* or *test* distribution be denoted $\mathbb{P}_2 = \mathbb{P}_{2,X} \times \mathbb{P}_{Y|X}$, where $\mathbb{P}_{2,X}$ is the marginal distribution of $\mathbb{P}_2$ over features. Since the conditional distribution of labels given features $\mathbb{P}_{Y|X}$ is common to $\mathbb{P}_1$ and $\mathbb{P}_2$, the test distribution is a covariate shifted version of the calibration distribution. Let $\mathbb{E}_i$ denote the expectation over $\mathbb{P}_i$, $i = 1, 2$. Let $x \mapsto r(x) = (d\mathbb{P}_{2,X}/d\mathbb{P}_{1,X})(x)$ denote the unknown likelihood ratio function.

We consider both discrete and continuous label spaces $\mathcal{Y}$. When $\mathcal{Y} = \mathbb{R}$, prediction sets correspond to prediction intervals. Recall that a prediction set $C : \mathcal{X} \to 2^{\mathcal{Y}}$ has marginal $(1 - \alpha)$-coverage in the test domain if $\mathbb{P}_2 [Y \in C(X)] \geqslant 1 - \alpha$. Let $S : (x, y) \mapsto S(x, y)$ denote the nonconformity score associated to a pair $(x, y) \in \mathcal{X} \times \mathcal{Y}$. Given a *threshold function* $q : \mathcal{X} \to \mathbb{R}$, we consider the corresponding prediction set $C : \mathcal{X} \to 2^{\mathcal{Y}}$ given by $C(x) = \{y \in \mathcal{Y} : S(x, y) \leqslant q(x)\}$ for all $x \in \mathcal{X}$. Thus a threshold function $q$ yields a conformal predictor with marginal $(1 - \alpha)$-coverage in the *test* domain if $\mathbb{P}_2[S(X, Y) \leqslant q(X)] \geqslant 1 - \alpha$. Note that the use of an adaptive threshold function is common in the conformal prediction literature, going back to [55]. We assume that $\alpha \leqslant 0.5$. For our theory, we consider $[0, 1]$-valued scores; however, in Appendix O, we comment on conditions under which unbounded scores can be handled.

In this paper, a linear hypothesis class refers to a linear subspace of functions from $\mathcal{X} \to \mathbb{R}$ that are square-integrable with respect to $\mathbb{P}_{1,X}$. An example is the space of functions representable by a pretrained model with a scalar read-out layer. If $\Phi : \mathcal{X} \to \mathbb{R}^d$ denotes the last hidden-layer feature map of the pretrained model, where $\Phi = (\phi_1, \ldots, \phi_d)$ for $\phi_i : \mathcal{X} \to \mathbb{R}$ for all $i \in [d]$, then the linear class of functions representable by the network is given by $\{\langle \gamma, \Phi \rangle : \gamma \in \mathbb{R}^d\}$, where $\langle \cdot, \cdot \rangle$ is the $\ell^2$ inner product on $\mathbb{R}^d$.

**Problem statement.** We observe $n_1$ labeled calibration (or, source) datapoints $\{(X_i, Y_i) : i \in [n_1]\}$ drawn i.i.d. from the source distribution $\mathbb{P}_1$, and an additional $n_3$ unlabeled calibration datapoints $\mathcal{S}_3$. We also have $n_2$ *unlabeled* (target) datapoints $\mathcal{S}_2$ drawn i.i.d. from the target distribution $\mathbb{P}_2$. Given $\alpha \in (0, 1)$, our goal is to construct a threshold function $q : \mathcal{X} \to \mathbb{R}$ that achieves marginal $(1 - \alpha)$-coverage in the test domain: $\mathbb{P}_2[S(X, Y) \leqslant q(X)] \geqslant 1 - \alpha$.

## 3 Algorithmic Principles

Here we present the intuition behind our approach. Our goal is to construct a prediction set of the form $C(x) = \{y \in \mathcal{Y} : S(x, y) \leqslant q(x)\}$, where $q$ should be close to a conditional quantile of $S$ given $X = x$. The quantile loss $\ell_\alpha$ is designed such that for any random variable $Z$, the minimizers of the objective $\kappa \mapsto \mathbb{E}\ell_\alpha(\kappa, Z)$ are the $(1 - \alpha)$th quantiles of $Z$. This has motivated prior work

[15, 22], where the authors minimize the objective $h \mapsto \mathbb{E}\ell_\alpha(h(X), S(X, Y))$ for $h$ in some linear hypothesis class $\mathcal{H}$. At a minimizer $h^*$, the derivatives in all directions $g \in \mathcal{H}$ should be zero. Since the derivative of the pinball loss with respect to its first argument is given by

$$\partial_1 \ell_\alpha(c, s) = -(1-\alpha)\mathbf{1}[s > c] + \alpha \mathbf{1}[s \leqslant c] = \mathbf{1}[s \leqslant c] - (1-\alpha),$$

the chain rule implies that the directional derivative of $h \mapsto \mathbb{E}\ell_\alpha(h(X), S(X, Y))$ in the direction $g$ equals

$$\left.\frac{\partial}{\partial \epsilon}\right|_{\varepsilon=0} \mathbb{E}_1[\ell_\alpha(h(X) + \epsilon g(X), S(X, Y))] = \mathbb{E}_1\left[\left.\frac{\partial}{\partial \epsilon}\right|_{\varepsilon=0} (h(X) + \epsilon g(X)) \cdot \partial_1 \ell_\alpha(h(X), S(X, Y))\right]$$
$$= \mathbb{E}_1[g(X)(\mathbf{1}[S(X, Y) \leqslant h(X)] - (1-\alpha))],$$

where in the first step we interchanged derivative and expectation, applied the chain rule, and evaluated at $\varepsilon = 0$, and in the second step we used the formula for the derivative of the pinball loss. Setting this equal to zero, if $g$ takes the form $g(x) = d\mathbb{Q}_X/d\mathbb{P}_{1,X}(x)$ for some distribution $\mathbb{Q}_X$, then[2] this equality reads $\mathbb{E}_\mathbb{Q}[\mathbf{1}[S(X, Y) \leqslant h^*(X)] - (1-\alpha)] = 0$, which can be viewed as exact coverage under the covariate shift induced by $g$ for the prediction set $x \mapsto \{y \in \mathcal{Y} : S(x, y) \leqslant h^*(x)\}$. In other words, if the test distribution is $\mathbb{Q} = \mathbb{Q}_X \times \mathbb{P}_{Y|X}$, then we have the exact coverage result

$$\mathbb{E}_\mathbb{Q}[\mathbf{1}[S(X, Y) \leqslant h^*(X)]] = \mathbb{Q}[S(X, Y) \leqslant h^*(X)] = 1 - \alpha.$$

Therefore, if the hypothesis class $\mathcal{H}$ is large enough to include the true likelihood ratio $r = d\mathbb{P}_{2,X}/d\mathbb{P}_{1,X}$, then the threshold function $h^*$ attains valid coverage in the test domain $\mathbb{P}_2$, as desired.

### 3.1  Our approach

**An adaptive choice of the hypothesis class.** The above approach requires special assumptions on the hypothesis class $\mathcal{H}$. The choice of the hypothesis class poses a challenge in practice: if $\mathcal{H}$ is too small, then coverage may fail, while if $\mathcal{H}$ is too large, then finite-sample performance may suffer due to large estimation errors.

To address this challenge, our idea is to choose $\mathcal{H}$ adaptively. We start by considering the class of hypotheses $h$ that are close to the true likelihood ratio $r$, as measured by $\mathbb{E}_1[(h(X) - r(X))^2]$ being small. By our remarks above, if we minimize $\mathbb{E}_1[\ell_\alpha(h(X), S(X, Y))]$ for $h$ restricted to this set, we obtain a threshold function with valid coverage under the covariate shift $r$.

**Removing the explicit dependence on the likelihood ratio.** The quantity $\mathbb{E}_1[(h(X) - r(X))^2]$ depends on the unknown $r$. However, we can expand this to obtain

$$\mathbb{E}_1[(h(X) - r(X))^2] = \mathbb{E}_1[h(X)^2] + \mathbb{E}_1[-2r(X)h(X)] + \mathbb{E}_1[r(X)^2].$$

The term $\mathbb{E}_1[r(X)^2]$ does not depend on the optimization variable $h$, so it is enough to consider the first two terms. Due to the change-of-measure identity $\mathbb{E}_1[r(X)h(X)] = \mathbb{E}_2[h(X)]$, the sum of these terms equals

$$\mathbb{E}_1[h(X)^2] + \mathbb{E}_1[-2r(X)h(X)] = \mathbb{E}_1[h(X)^2] + \mathbb{E}_2[-2h(X)].$$

A key observation is that neither of the terms $\mathbb{E}_1[h(X)^2]$ or $\mathbb{E}_2[-2h(X)]$ explicitly involve $r$, and thus they can be estimated by sample averages over the source and target data, respectively. Thus, we can minimize $\mathbb{E}_1[\ell_\alpha(h(X), S(X, Y))]$ over $h \in \mathcal{H}$ while keeping $\mathbb{E}_1[h(X)^2] + \mathbb{E}_2[-2h(X)]$ bounded. The threshold $h^*$ will have valid coverage under the covariate shift $r$.

**Introducing a normalizing scalar.** We also need to make sure that $h$ is a valid likelihood ratio under $d\mathbb{P}_{1,X}$, of the form $g(x) = d\mathbb{Q}_X/d\mathbb{P}_{1,X}(x)$ for some distribution $\mathbb{Q}_X$. This imposes the constraint $\int h(x)d\mathbb{P}_{1,X}(x) = 1$, which can be equivalently achieved for any non-negative $h$ by scaling it with an appropriate scalar $\beta$. In our analysis, it turns out to be convenient to use the optimization variable $\beta h$ and consider the class of functions $h$ such that $\mathbb{E}_1[(\beta h(X) - r(X))^2]$ is bounded for some scalar $\beta \in \mathbb{R}$. By the above discussion, the term $\mathbb{E}_1[r(X)^2]$ is immaterial and it is sufficient to impose the constraint that $\min_{\beta \in \mathbb{R}}(\mathbb{E}_1[\beta^2 h(X)^2] + \mathbb{E}_2[-2\beta h(X)])$ is bounded.

---

[2]This holds due to the change of measure identity $\mathbb{E}_\mathbb{P}[d\mathbb{Q}/d\mathbb{P}(X) \cdot h(X)] = \mathbb{E}_\mathbb{Q}[h(X)]$ for all integrable functions $h$.

---

**Algorithm 1** Likelihood-ratio regularized quantile regression

---

**Input:** $n_1$ labeled source datapoints, $n_2$ unlabeled target datapoints, $n_3$ unlabeled source datapoints

1: Compute scores $S_i = S(x_i, y_i)$ for all $i \in [n_1]$

2: Solve $(\hat{h}, \hat{\beta}) \in \arg\min_{h \in \mathcal{H}, \beta \in \mathbb{R}} \hat{\mathbb{E}}_1[\ell_\alpha(h(X), S(X, Y))] + \lambda \hat{\mathbb{E}}_3[\beta^2 h(X)^2] + \lambda \hat{\mathbb{E}}_2[-2\beta h(X)]$,
where $\hat{\mathbb{E}}_1, \hat{\mathbb{E}}_2, \hat{\mathbb{E}}_3$ denote expectations over the source, unlabeled target, and unlabeled source data;

**Return:** Prediction set $\hat{C}(x) \leftarrow \{y \in \mathcal{Y} : S(x, y) \leqslant \hat{h}(x)\}$ with asymptotic $1 - \alpha$ coverage in the target distribution

---

**Replacing the constraint with a regularization.** Instead of imposing a constraint on $\min_{\beta \in \mathbb{R}}(\mathbb{E}_1[\beta^2 h(X)^2] + \mathbb{E}_2[-2\beta h(X)])$, we can use this term as a regularizer. Given a regularization strength $\lambda \geqslant 0$, we can solve

$$\min_{h \in \mathcal{H}} \left\{ \mathbb{E}_1[\ell_\alpha(h(X), S(X, Y))] + \lambda \min_{\beta \in \mathbb{R}}(\mathbb{E}_1[\beta^2 h(X)^2] + \mathbb{E}_2[-2\beta h(X)]) \right\}.$$

Since the first term does not depend on $\beta$, this is equivalent to the joint optimization problem

$$\min_{h \in \mathcal{H}, \beta \in \mathbb{R}} \left\{ L_\lambda(h, \beta) := \mathbb{E}_1[\ell_\alpha(h(X), S(X, Y))] + \lambda(\mathbb{E}_1[\beta^2 h(X)^2] - \mathbb{E}_2[2\beta h(X)]) \right\}. \quad \text{(LR-QR)}$$

### 3.2 Algorithm: likelihood ratio regularized quantile regression

We solve an empirical version of this objective. We use our labeled source data $\{(X_i, Y_i) : i \in [n_1]\}$ to estimate $\mathbb{E}_1[\ell_\alpha(h(X), S(X, Y))]$, our additional unlabeled source data $\mathcal{S}_3$ to estimate $\mathbb{E}_1[\beta^2 h(X)^2]$, and our unlabeled target data $\mathcal{S}_2$ to estimate $\lambda \mathbb{E}_2[-2\beta h(X)]$. Letting $\hat{\mathbb{E}}_1, \hat{\mathbb{E}}_2$, and $\hat{\mathbb{E}}_3$ denote empirical expectations over $\{(X_i, Y_i) : i \in [n_1]\}$, $\mathcal{S}_2$, and $\mathcal{S}_3$, respectively, we then solve the following empirical likelihood ratio regularized quantile regression problem, for $\lambda \geqslant 0$:

$$(\hat{h}, \hat{\beta}) \in \arg\min_{h \in \mathcal{H}, \beta \in \mathbb{R}} \left\{ \hat{L}_\lambda(h, \beta) := \hat{\mathbb{E}}_1[\ell_\alpha(h(X), S(X, Y))] + \lambda \hat{\mathbb{E}}_3[\beta^2 h(X)^2] - \lambda \hat{\mathbb{E}}_2[2\beta h(X)] \right\}.$$
$$\text{(Empirical-LR-QR)}$$

Our proposed threshold is $q = \hat{h}$. See Algorithm 1. In the following section, we justify this algorithm through a novel theoretical analysis of the test-time coverage.

## 4 Theoretical Results

### 4.1 Infinite sample setting

We first consider the infinite sample or "population" setting, characterizing the solutions of the LR-QR problem from (LR-QR) in an idealized scenario where the exact values of the expectations $\mathbb{E}_1, \mathbb{E}_2$ can be calculated. In this case, we will show that if the hypothesis class $\mathcal{H}$ is linear and contains the true likelihood ratio $r$, then the optimizer achieves valid coverage in the test domain. Let $r_{\mathcal{H}}$ be the projection of $r$ onto $\mathcal{H}$ in the Hilbert space induced by the inner product $\langle f, g \rangle = \mathbb{E}_1[fg]$.[3] The key step is the result below, which characterizes coverage weighted by $r_{\mathcal{H}}$.

**Proposition 4.1.** *Let $\mathcal{H}$ be a linear hypothesis class consisting of square-integrable functions with respect to $\mathbb{P}_{1,X}$. Then under regularity conditions specified in Appendix E (the conditions of Lemma L.3), if $(h^*, \beta^*) = (h_\lambda^*, \beta_\lambda^*)$ is a minimizer of the objective in Equation (LR-QR) with regularization strength $\lambda > 0$, then we have $\mathbb{E}_1[r_{\mathcal{H}}(X)\mathbf{1}[S(X, Y) \leqslant h^*(X)]] \geqslant 1 - \alpha$.*

The proof is given in Appendix I. As a consequence of Proposition 4.1, if $\mathcal{H}$ contains the true likelihood ratio $r$, so that $r_{\mathcal{H}} = r$, then in the infinite sample setting, the LR-QR threshold function $h^*$ attains valid coverage at test-time:

$$\mathbb{E}_1[r(X)\mathbf{1}[S(X, Y) \leqslant h^*(X)]] = \mathbb{P}_2[S(X, Y) \leqslant h^*(X)] \geqslant 1 - \alpha.$$

---

[3]Explicitly, given an orthonormal basis $\{\varphi_1, \ldots, \varphi_d\}$ for $\mathcal{H}$, we have $r_{\mathcal{H}} = \sum_{i=1}^{d} \langle r, \varphi_i \rangle \varphi_i$.

However, in practice, we can only optimize over finite-dimensional hypothesis classes, and as a result we must control the effect of mis-specifying $\mathcal{H}$. If $r$ is not in $\mathcal{H}$, we can derive a lower bound on the coverage as follows. First, write

$$\mathbb{E}_1[r(X)\mathbf{1}[S(X,Y) \leqslant h^*(X)]]$$
$$= \mathbb{E}_1[r_{\mathcal{H}}(X)\mathbf{1}[S(X,Y) \leqslant h^*(X)]] + \mathbb{E}_1[(r(X) - r_{\mathcal{H}}(X))\mathbf{1}[S(X,Y) \leqslant h^*(X)]].$$

By Proposition 4.1, the first term on the right-hand side is at least $1 - \alpha$. Since the random variable $\mathbf{1}[S(X,Y) \leqslant h^*(X)]$ is $\{0,1\}$-valued, the second term on the right-hand side is at least $-\mathbb{E}_1[(r(X) - r_{\mathcal{H}}(X))_+]$, where $(x)_+ = \max\{0, x\}$ for $x \in \mathbb{R}$. We set our threshold function $q$ equal equal $h^*$, so that our conformal prediction sets equal $C^*(x) = \{y \in \mathcal{Y} : S(x,y) \leqslant h^*(x)\}$ for all $x \in \mathcal{X}$. Thus, we have the lower bound

$$\mathbb{P}_2[Y \in C^*(X)] = \mathbb{E}_1[r(X)\mathbf{1}[S(X,Y) \leqslant h^*(X)]] \geqslant (1-\alpha) - \mathbb{E}_1[(r(X) - r_{\mathcal{H}}(X))_+].$$

Geometrically, this coverage gap is the result of restricting to $\mathcal{H}$. This error decreases if $\mathcal{H}$ is made larger, but in the finite sample setting, this comes at the risk of overfitting.

## 4.2 Finite sample setting

From the analysis of the infinite sample regime, it is clear that if the hypothesis class $\mathcal{H}$ is made larger, the test-time coverage of the population level LR-QR threshold function $h^*$ moves closer to the nominal value. However, in the finite sample setting, optimizing over a larger hypothesis class also presents the risk of overfitting. By tuning the regularization parameter $\lambda$, we are trading off the estimation error incurred for the first term of Equation (LR-QR), namely $(\hat{\mathbb{E}}_1 - \mathbb{E}_1)[\ell_\alpha(h(X), S(X,Y))]$, and the error incurred for the second and third terms of Equation (LR-QR), namely $\lambda(\hat{\mathbb{E}}_3 - \mathbb{E}_3)[\beta^2 h(X)^2] + \lambda(\hat{\mathbb{E}}_2 - \mathbb{E}_2)[-2\beta h(X)]$. Heuristically, for a fixed $h$, the former should be proportional to $1/\sqrt{n_1}$, and the latter should be proportional to $\lambda(1/\sqrt{n_3} + 1/\sqrt{n_2})$. Thus, if we pick $\lambda$ to make these two errors of equal order, it will be proportional to $\sqrt{(n_2 + n_3)/n_1}$.

Put differently, in order to ensure that the Empirical LR-QR threshold $\hat{h}$ from Equation (Empirical-LR-QR) has valid test coverage, one must choose the regularization $\lambda$ based on the relative amount of labeled and unlabeled data. The unlabeled datapoints carry information about the covariate shift $r$, because $r$ depends only on the distribution of the features. The labeled datapoints provide information about the *conditional $(1 - \alpha)$-quantile function* $q_{1-\alpha}$, which depends only on the conditional distribution of $S|X$. When $\lambda$ is large, our optimization problem places more weight on approximating $r$ (the minimizer of $\mathbb{E}_1[(\beta h(X) - r(X))^2]$ in $\beta h$), and if $\lambda$ is small, we instead aim to approximate $q_{1-\alpha}$ (the minimizer of $\mathbb{E}_1[\ell_\alpha(h(X), S(X,Y))]$ in $h$). Therefore, if the number of unlabeled datapoints $(n_2 + n_3)$ is large compared to the number of labeled datapoints $(n_1)$, our data contains much more information about the covariate shift $r$, and we should set $\lambda$ to be large. If instead $n_1$ is very large, the quantile function $q_{1-\alpha}$ can be well-approximated from the labeled calibration datapoints, and we set $\lambda$ to be close to zero. In the theoretical results, we make this intuition precise.

In order to facilitate our theoretical analysis in the finite sample setting, we consider constrained versions of Equation (LR-QR) and Equation (Empirical-LR-QR). Fix a collection $\Phi = (\phi_1, \ldots, \phi_d)^\top$ of $d$ basis functions, where $\phi_i : \mathcal{X} \to \mathbb{R}$ for $i \in [d]$. Let $\mathcal{I} = [\beta_{\min}, \beta_{\max}] \subset \mathbb{R}$ be an interval with $\beta_{\min} > 0$. Let $\mathcal{H}_B = \{\langle \gamma, \Phi \rangle : \|\gamma\|_2 \leqslant B < \infty\}$ be the $B$-ball centered at the origin in the linear hypothesis class spanned by $\{\phi_1, \ldots, \phi_d\}$. We equip $\mathcal{H}_B$ with the norm $\|h\| = \|\gamma\|_2$ for $h = \langle \gamma, \Phi \rangle$.

At the population level, consider the following constrained LR-QR problem: $(h^*, \beta^*) \in \arg\min_{h \in \mathcal{H}_B, \beta \in \mathcal{I}} L_\lambda(h, \beta)$. Also consider the following empirical constrained LR-QR problem[4]:

$$(\hat{h}, \hat{\beta}) \in \arg\min_{h \in \mathcal{H}_B, \beta \in \mathcal{I}} \hat{L}_\lambda(h, \beta). \tag{2}$$

We begin by bounding the generalization error of an ERM $(\hat{h}, \hat{\beta})$ computed via Equation (2).

**Theorem 4.2** (Suboptimality gap of ERM for likelihood ratio regularized quantile regression). *Under the regularity conditions specified in Appendix E, and for appropriate choices of the optimization hyperparameters[5], for sufficiently large $n_1, n_2, n_3$, with probability at least $1 - \delta$, any optimizer*

---

[4]For brevity, this notation overloads the definition of $(\hat{h}, \hat{\beta})$ from (Empirical-LR-QR). From now on, $(\hat{h}, \hat{\beta})$ will refer to the definition from (2), and the one from (Empirical-LR-QR) will not be used again.

[5]Specifically, suppose that $\beta_{\min} \leqslant \beta_{\text{lower}}$, $\beta_{\max} \geqslant \beta_{\text{upper}}$, and $B \geqslant B_{\text{upper}}$, where the positive scalars $\beta_{\text{lower}}$, $\beta_{\text{upper}}$, and $B_{\text{upper}}$ are defined in Lemma L.4 in the Appendix, and depend on the data distribution and the choice of basis functions, but not on the data, the sample sizes, or the regularization parameter $\lambda$.

$(\hat{h}, \hat{\beta})$ of the empirical constrained LR-QR objective from (2) with regularization strength $\lambda > 0$ has suboptimality gap $L_\lambda(\hat{h}, \hat{\beta}) - L_\lambda(h^*, \beta^*)$ with respect to the population risk (LR-QR) bounded by

$$\mathcal{E}_{\text{gen}} := c\lambda\sqrt{1/n_2 + 1/n_3} + c'/\sqrt{n_1} + c''/\sqrt{\lambda n_1},$$

and $c, c', c''$ are positive scalars that do not depend on $\lambda$.

The proof is in Appendix J. The generalization error $\mathcal{E}_{\text{gen}}$ is minimized for an optimal regularization on the order of

$$\lambda^* \propto n_1^{-1/3} \left(1/n_2 + 1/n_3\right)^{-1/3}, \tag{3}$$

which yields an optimized upper bound of order $\mathcal{E}_{\text{gen}}^* = O\left(n_1^{-1/3}\left(1/n_2 + 1/n_3\right)^{1/6} + 1/\sqrt{n_1}\right)$. As can be seen from Appendix F, $c, c', c''$ depend only polynomially on the radius $B$.

As a corollary of Theorem 4.2, we have the following lower bound on the excess marginal coverage of our ERM threshold $\hat{h}$ in the covariate shifted domain. Let $r_B$ denote the projection of $r$ onto the closed convex set $\mathcal{H}_B$ in the Hilbert space induced by the inner product $\langle f, g \rangle = \mathbb{E}_1[fg]$.

**Theorem 4.3** (Main result: Coverage under covariate shift). *Under the same conditions as Theorem 4.2, consider the LR-QR optimizers $\hat{h}$ and $\hat{\beta}$ from (2) with regularization strength $\lambda > 0$. Given any $\delta > 0$, for sufficiently large $n_1, n_2, n_3$, we have with probability at least $1 - \delta$ that*[6]

$$\mathbb{P}_2\left[Y \in \hat{C}(X)\right] \geqslant (1 - \alpha) + 2\hat{\beta}\lambda\mathbb{E}_1[(r_B(X) - \hat{\beta}\hat{h}(X))^2] - \mathcal{E}_{\text{cov}} - (1 - \alpha)\mathbb{E}_1[|r(X) - r_B(X)|],$$

*where $\mathcal{E}_{\text{cov}} := A\left(1/n_2 + 1/n_3\right)^{1/4}\lambda + A'(\lambda n_1)^{-1/4} + \lambda^{1/2}/n_1^{1/4}$, $r_B$ denotes the projection of $r$ onto $\mathcal{H}_B$, and $A, A'$ are positive scalars that do not depend on $\lambda$.*

The proof is in Appendix K. This result states that our LR-QR method has nearly valid coverage at level $1 - \alpha$ under covariate shift, up to small error terms that we can control. The quantity $\mathcal{E}_{\text{cov}}$ vanishes as we collect more data. The term $\mathbb{E}_1[|r(X) - r_B(X)|]$ captures the level of mis-specification by not including the true likelihood ratio function $r$ in our hypothesis class $\mathcal{H}_B$. This can be decreased by making the hypothesis class $\mathcal{H}_B$ larger. Of course, this will also increase the size of the terms $A, A'$ in our coverage error, but in our theory we show that the dependence is mild. Indeed, the terms depend only on a few geometric properties of $\mathcal{H}_B$: they depend polynomially on the radius $B$, on the eigenvalues of the sample covariance matrix of the basis $\Phi(X)$ under the source distribution, and on a quantitative measure of linear dependence of the features; but not explicitly on the dimension of the basis. We also note that the dimension of the feature space $\dim(X)$ does not appear in our results; only $\dim(\mathcal{H})$ affects our bounds.

We highlight the term $2\hat{\beta}\lambda\mathbb{E}_1[(r_B(X) - \hat{\beta}\hat{h}(X))^2]$, which is an error term relating the projected likelihood ratio $r_B$ to the LR-QR solution $\hat{\beta}\hat{h}$. Crucially, this term is a non-negative quantity multiplied by $\lambda$, and so for appropriate $\lambda$ it may counteract in part the coverage error loss. Consistent with the above observations, we find empirically that choosing small nonzero regularization parameters improves coverage. Moreover, we find that choosing the regularization parameter to be on the order of the optimal value for $\mathcal{E}_{\text{cov}}$ is suitable choice across a range of experiments.

Our proofs are quite involved and require a number of delicate arguments. Crucially, they draw on a *novel analysis of coverage via stability bounds* from learning theory. Existing stability results cannot directly be applied, due to our use of a data-dependent regularizer. For instance, in classical settings, the optimal regularization tends to zero as the sample size goes to infinity, but this is not the case here. To overcome this challenge, we combine stability bounds [48, 49] with a novel conditioning argument, and we show that the values of $L$ at the minimizers of $\hat{L}$ and $L$ are close by introducing intermediate losses that sequentially swap out empirical expectations $\hat{\mathbb{E}}_1, \hat{\mathbb{E}}_2, \hat{\mathbb{E}}_3$ with their population counterparts. We then leverage the smoothness of $L$, to derive that the gradient of $L$ at $(\hat{\beta}, \hat{h})$ is small. Finally, we show that a small gradient implies the desired small coverage gap.

---

[6]The probability $\mathbb{P}_2\left[Y \in \hat{C}(X)\right]$ is over $(X, Y) \sim \mathbb{P}_2$, conditional on $\hat{C}$.

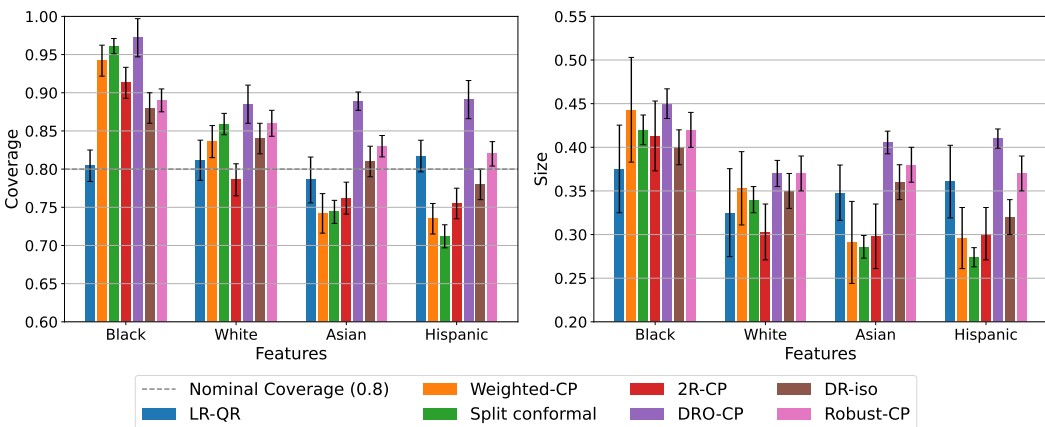

Figure 1: (Left) Coverage. (Right) Average prediction set size on the Communities and Crime dataset.

## 5 Experiments

We compare our method with the following baselines: (1) Split/inductive conformal prediction [31, 36]; (2) Weighted-CP: Weighted conformal prediction [55]; (3) 2R-CP: The *doubly robust* method from Yang et al. [64]; (4) DRO-CP: Distributionally robust optimization [7]; (5) DR-iso: Isotonic distributionally robust optimization [19]; (6) Robust-CP: Robust weighted conformal prediction [1].

### 5.1 Choosing the Regularization Parameter

Equation (3) suggests an optimal choice of the regularization parameter $\lambda$ in the LR-QR algorithm. Guided by this, we form a uniform grid of size ten from $\lambda^*/10$ to $\lambda^*$. We then perform three-fold cross-validation over the combined calibration and unlabeled target datasets (without using any labeled test data) as follows: we train the LR-QR threshold for each $\lambda$, and compute as a validation measure the $\ell^2$-norm of the gradient of the LR-QR objective on the held-out fold. We pick $\lambda$ with the smallest average validation measure across all folds.

This validation measure is motivated by our algorithmic development: the first-order conditions of the LR-QR objective play a fundamental role in ensuring valid coverage in the test domain. While the model is trained to satisfy these conditions on the observed data, we seek to ensure this property generalizes well to unseen data. Thus, our selection criterion is based on two key observations: (1) a small gradient of the LR-QR objective implies reliable coverage, and (2) the regularization parameter $\lambda$ balances the generalization error of the two terms in LR-QR. By minimizing this measure, we select a $\lambda$ that optimally trades off these competing factors.

Finally, we re-train the LR-QR threshold on the entire calibration and unlabeled target datasets using this best $\lambda$, and report coverage and interval size on the held-out labeled test set. This ensures that no test labels are used during hyperparameter tuning. Additionally, in Appendix B, we provide deeper insights on different regimes of regularization in practice through an ablation study.

### 5.2 Communities and Crime

We evaluate our methods on the *Communities and Crime* dataset [42], which contains 1994 datapoints corresponding to communities in the United States, with socio-economic and demographic statistics. The task is to predict the (real-valued) per-capita violent crime rate from a 127-dimensional input.

We first randomly select half of the data as a training set, and use it to fit a ridge regression model $\hat{f}$ as our predictor. We tune the ridge regularization with five-fold cross-validation. We use the remaining half to design four covariate shift scenarios, determined by the frequency of a specific racial subgroup (Black, White, Hispanic, and Asian). For each of these features, we find the median value $m$ over the remaining dataset. Datapoints with feature value at most $m$ form our *source* set, and the rest form our *target* set. In other words, in each scenario, the source set consists of data points with below-median frequency of the specified racial subgroup, while the target set contains those with above-median frequency. This creates a covariate shift between calibration and test, as the split procedure only observes the covariates and is independent of labels. We then further split the target set into roughly equal *unlabeled* and *labeled* subsets. The unlabeled subset and the calibration data (without the labels) is used to estimate $r$, while the labeled test subset is held out *only* for final evaluation. The same procedure is applied to each of the four racial subgroups, creating four distinct partitions.

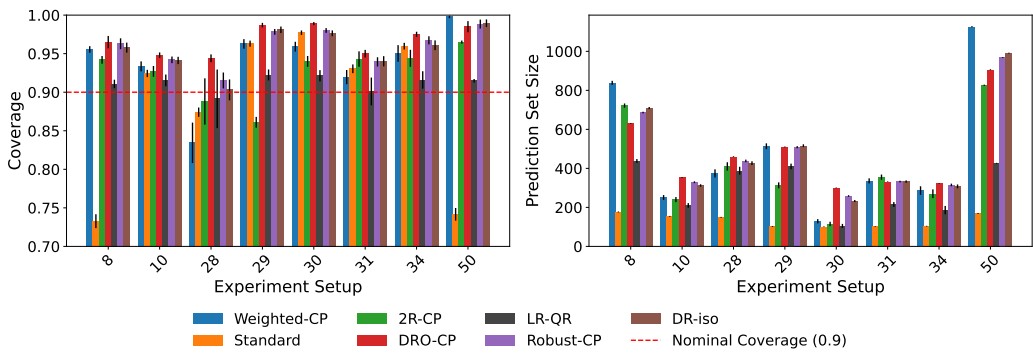

Figure 2: (Left) Coverage, (Right) Average prediction set size on the RxRx1 dataset from the WILDS repository.

**Experimental details.** The nonconformity score is $s(x, y) = |y - \hat{f}(x)|$. Several baselines require an estimate of the likelihood ratio $r$, which we obtain by training a logistic regression model $\hat{p}$ to distinguish unlabeled source and target data. We then set $\hat{r} = \frac{\hat{p}}{1-\hat{p}}$, where $\hat{p}(x)$ is the predicted probability that $x$ came from the target distribution. The hypothesis class $\mathcal{H}$ consists of all linear maps from the feature space to $\mathbb{R}$. All experimental results are averaged over 1000 random splits.

**Results.** Figure 1 displays the results. Notably, split conformal undercovers in two setups and overcovers in the other two. Methods that estimate $r$ and DRO fail to track the nominal coverage, particularly in the first setup on the left. However, the LR-QR method is closer to the nominal level of coverage, showing a stronger adaptivity to the covariate shift.

## 5.3 RxRx1 data - WILDS

Our next experiment uses the RxRx1 dataset [53] from the WILDS repository [25], which is designed to evaluate model robustness under distribution shifts. The RxRx1 task involves classifying cell images based on 1339 laboratory genetic treatments. These images, captured using fluorescent microscopy, originate from 51 independent experiments. Variations in execution and environmental conditions lead to systematic differences across experiments, affecting the distribution of input features (e.g., lighting, cell morphology) while the relationship between inputs and labels remains unchanged. This situation creates covariate shift where the marginal distribution of inputs shifts across domains, but the conditional distribution $\mathbb{P}_{Y|X}$ remains the same.

We use a ResNet50 model [20] trained by the WILDS authors on 37 of the 51 experiments. Using the other experiments, we construct 14 distinct evaluations, where each experiment is selected as the target dataset, and its data is evenly split into an unlabeled target set and a labeled test set. The labeled data from the other 13 experiments serves as the source dataset.

**Experimental details.** The nonconformity score is $s(x, y) = -\log f_x(y)$, where $f_x(y)$ is the probability assigned the image-label pair $(x, y)$. To estimate $r$, we train a logistic regression model $\hat{p}$ on top of the representation layer of the pretrained model to distinguish unlabeled source and target data, and we set $\hat{r} = \frac{\hat{p}}{1-\hat{p}}$. We set the hypothesis class $\mathcal{H}$ to be a linear head on top of the representation layer of the pretrained model. Experimental results are averaged over 50 random splits.

**Results.** Figure 2 presents the coverage and average prediction set size for all methods. To enhance visual interpretability, we display results for eight randomly selected settings out of the 14, with the full plot provided in Figure 3 in the Appendix. The x-axis shows the indices of the test condition. LR-QR adheres more closely to the nominal coverage value of 0.9 compared to other methods.

Notably, split conformal prediction, which assumes exchangeability between calibration and test data, shows under- and overcoverage due to the covariate shift. The coverage of weighted CP and 2R-CP is also far from the nominal level, showing that directly estimating the likelihood ratio and conditional quantile is insufficient to correct the coverage violations in the case of high-dimensional image data. Further, the superior coverage of LR-QR is not due to inflated prediction sets.

### 5.4 Multiple choice questions - MMLU

Finally, we evaluate all methods using the MMLU benchmark, which covers 57 subjects spanning a wide range of difficulties. To induce a covariate shift, we partition the dataset by subject difficulty: prompts from subjects labeled as *elementary* or *high school* are used for calibration, while those from *college* and *professional* subjects form the test set.

Motivated by the design from [26], we follow a prompt-based scoring scheme adapted for LLMs: we append the string "The answer is the option:" to the end of each MMLU question and feed the resulting prompt into the Llama 13B model without generating any output. We then extract the next-token logits corresponding to the first decoding position (i.e., immediately after the prompt) and consider the logits associated with the characters A, B, C, and D. These four logits are normalized using the softmax function to produce a probability vector over the answer options.

**Experimental details.** The nonconformity score is $s(x, y) = 1 - f(x)_y$, where $f(x)_y$ is the probability assigned to the correct answer. For $\hat{r}$ and $\mathcal{H}$, we compute prompt embeddings as follows. We extract the final hidden layer outputs from GPT-2 Small to obtain 768-dimensional embeddings. We then apply average pooling across all token embeddings in a prompt to obtain a single fixed-length vector representation for each input. We fit a probabilistic classifier $\hat{p}$ using logistic regression on the unlabeled pooled embeddings from the source and target data, and we set $\hat{r} = \frac{\hat{p}}{1-\hat{p}}$. We set $\mathcal{H}$ to be a linear head on top of the representation layer of the pretrained model.

**Results.** As shown in Table 1, our LR-QR method achieves near-nominal coverage and has the smallest average prediction set size among methods that achieve approximately 90% or higher coverage, demonstrating both validity and efficiency under covariate shift.

Table 1: Comparison of Methods by Coverage and Set Size (mean $\pm$ std)

| Metric | Nominal | LR-QR | DRO | WCP |
|---|---|---|---|---|
| Coverage (%) | $90.0 \pm 0.0$ | $89.6 \pm 1.2$ | $99.7 \pm 0.3$ | $86.5 \pm 1.5$ |
| Set Size | – | $3.38 \pm 0.15$ | $3.92 \pm 0.20$ | $3.31 \pm 0.12$ |

| Metric | SCP | DR-iso | Robust-CP | 2R-CP |
|---|---|---|---|---|
| Coverage (%) | $78.1 \pm 2.1$ | $96.3 \pm 0.6$ | $95.8 \pm 0.7$ | $96.9 \pm 0.5$ |
| Set Size | $2.60 \pm 0.10$ | $3.64 \pm 0.18$ | $3.56 \pm 0.14$ | $3.80 \pm 0.17$ |

## 6 Discussion

We proposed the LR-QR method to construct prediction sets under covariate shift. While we have provided strong guarantees on the coverage of our method, it would be desirable to have results that control of the slack in coverage in specific scenarios depending on the structure of the likelihood ratio and the hypothesis space. Our work concerns uncertainty quantification and may have positive social impact for reliable decision-making. We do not envision any negative social impact of our work.

## 7 Acknowledgments

ED and SJ were supported by NSF, ARO, ONR, AFOSR, and the Sloan Foundation. The work of HH, SK, and GP was supported by the NSF Institute for CORE Emerging Methods in Data Science (EnCORE) and the ASSET (AI-Enabled Systems: Safe, Explainable and Trustworthy) Center.

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

## A  Additional figures

## B  Ablation studies

Here we provide an ablation study for $\lambda$, the regularization strength that appears in the LR-QR objective. In the same regression setup as Section 5.2, instead of selecting $\lambda$ via cross-validation, here we sweep the value of $\lambda$ from 0 to 1, and we plot the coverage of the LR-QR algorithm on the test data. Here, note that the split ratios between train, calibration, and test (both labeled and unlabeled data) are fixed and similar to the setup in Section 5.2. We report the averaged plots over 100 independent splits.

Figure 4 displays the effect of different regimes of $\lambda$. At one extreme, when $\lambda$ is close to zero, the LR-QR algorithm reduces to ordinary quantile regression. In this regime, the LR-QR algorithm behaves similarly to the algorithm from [15], without the test covariate imputation. In other words, when we set $\lambda = 0$, we try to provide coverage with respect to all the covariate shifts in the linear hypothesis class that we optimize over. As we can see in Figure 4, this can lead to overfitting and undercoverage of the test labels. As we increase $\lambda$, as a direct effect of the regularization, the coverage gap decreases. This is primarily due to the fact that larger $\lambda$ restricts the space of quantile regression optimization in such a way that it does not hurt the test time coverage, since the regularization is designed to shrink the optimization space towards the true likelihood-ratio. Thus, the regularization improves the generalization of the selected threshold, as the effective complexity of the hypothesis class is getting smaller. That being said, this phenomenon is only applicable if $\lambda$ lies within a certain range; once $\lambda$ grows too large, due to the data-dependent nature of our regularization, the generalization error of the regularization term itself becomes non-negligible and hinders the precise test-time coverage of the LR-QR threshold. As is highlighted in Figure 4, our theoretical results suggest an optimal regime for $\lambda$ which can best exploit the geometric properties of the LR-QR threshold.

## C  Related work

The basic concept of prediction sets dates back to foundational works such as Wilks [63], Wald [62], Scheffe and Tukey [46], and Tukey [56, 57]. The early ideas of conformal prediction were developed in Saunders et al. [45], Vovk et al. [61]. With the rise of machine learning, conformal prediction has emerged as a widely used framework for constructing prediction sets [e.g., 2, 4, 8–11, 16–18, 27–31, 34–36, 43, 58, 59]. A wide range of predictive inference methods have been developed [e.g., 14, 24, 33, 37, 38, 40, 44, 47, 51].

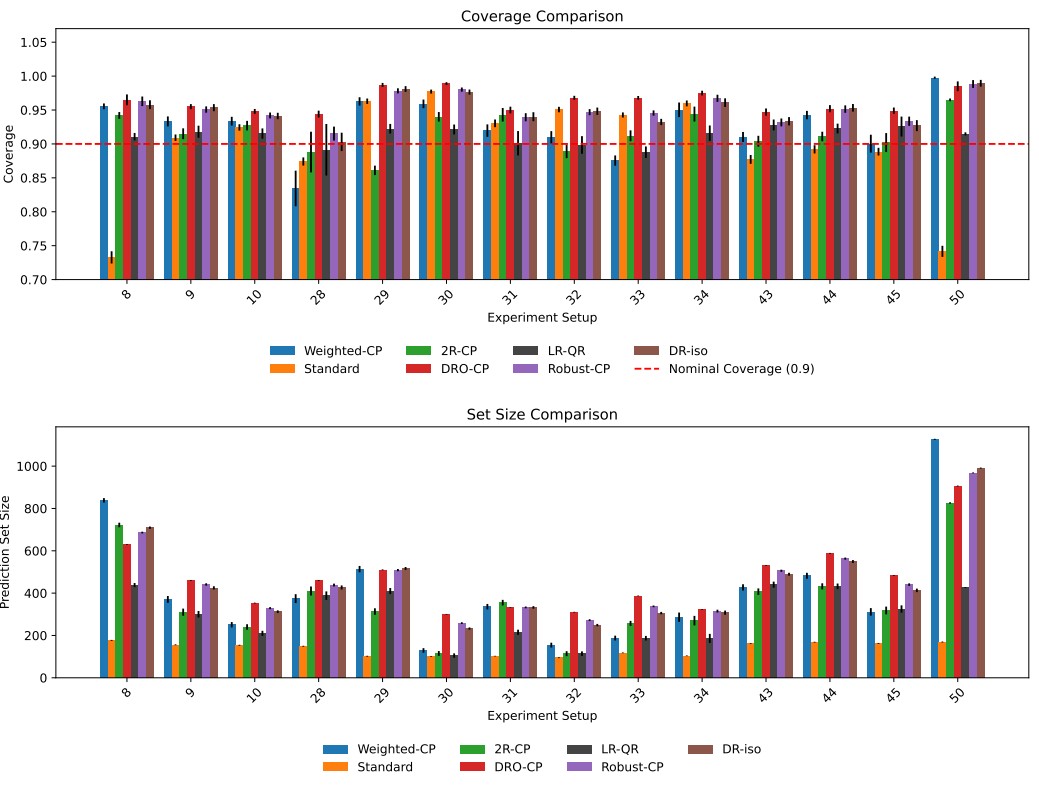

Figure 3: (Left) Coverage, (Right) Average prediction set size.

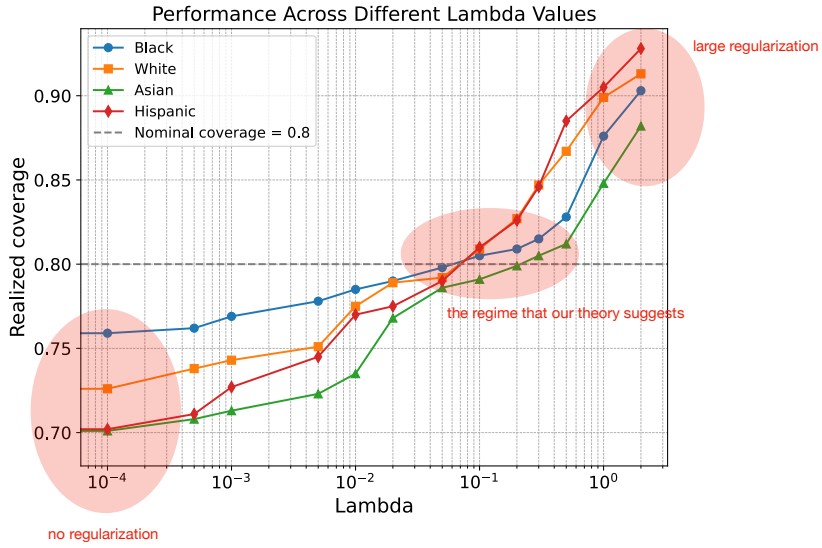

Figure 4: Ablation study for the effect of $\lambda$ on LR-QR performance in the experimental setup of Section 5.2.

Numerous works have addressed conformal prediction under various types of distribution shift [37, 38, 40, 51, 55]. For example, Tibshirani et al. [55] investigated conformal prediction under covariate shift, assuming the likelihood ratio between source and target covariates is known. Lei and Candès [32] allowed the likelihood ratio to be estimated, rather than assuming it is known. Park et al. [37] developed prediction sets with a calibration-set conditional (PAC) property under covariate shift. [3] present the nonexchangeable conformal prediction algorithm for arbitrary distribution shifts, assuming that the optimal weights for their method are known. Qiu et al. [40], Yang et al. [64] developed prediction sets with asymptotic coverage that are doubly robust in the sense that their coverage error is bounded by the product of the estimation errors of the quantile function of the score and the likelihood ratio. Cauchois et al. [7] construct prediction sets based on a distributionally robust optimization approach. Gui et al. [19] develop methods based on an isotonic regression estimate of the likelihood ratio. They provide theoretical guarantees for the difference between the population-level distributionally robust risk and its empirical counterpart. However, their results do not directly lead to coverage guarantees under distribution shift in our setting, as that would further require characterizing the effect of estimating the likelihood ratio.

Qin et al. [39] combine a parametric working model with a resampling approach to construct prediction sets under covariate shift. Bhattacharyya and Barber [6] analyze weighted conformal prediction in the special case of covariate shifts defined by a finite number of groups. Ai and Ren [1] reweight samples to adapt to covariate shift, while simultaneously using distributionally robust optimization to protect against worst-case joint distribution shifts. Kasa et al. [23] construct prediction sets by using unlabeled test data to modify the score function used for conformal prediction.

Our algorithm works by constructing a novel regularized regression objective, whose stationary conditions ensure coverage in the test domain. We can minimize the objective by estimating certain expectations of the data distribution—which implicitly involve estimating only certain functionals of the likelihood ratio. We further show that the coverage is retained in finite samples via a novel analysis of coverage leveraging stability bounds [48, 49]. We illustrate that our algorithms behave better in high-dimensional datasets than existing methods.

# D  Notation and conventions

Constants are allowed to depend on dimension only through properties of the population and sample covariance matrices of the features, and the amount of linear independence of the features; see the quantities $\lambda_{\min}(\Sigma)$, $\lambda_{\max}$, $c_{\min}$, $c_{\max}$, and $c_{\text{indep}}$ defined in Appendix E. In the Landau notation ($o$, $O$, $\Theta$), we hide constants. We say that a sequence of events holds with high probability if the probability of the events tends to unity. We define $\mathcal{S}_1$ as the features of the labeled calibration dataset. All functions that we minimize can readily be verified to be continuous, and thus attain a minimum over the compact domains over which we minimize them; thus all our minimizers will be well-defined. We may not mention this further. We denote by $\mathbf{1}[A]$ the indicator of an event $A$. Recall that $\mathcal{H}$ denotes the linear hypothesis class $\mathcal{H} = \{\langle \gamma, \Phi \rangle : \gamma \in \mathbb{R}^d\}$. This defines a one-to-one correspondence between $\mathbb{R}^d$ and $\mathcal{H}$. This enables us to view functions defined on $\mathbb{R}^d$ equivalently as defined on $\mathcal{H}$. In our analysis, we will use such steps without further discussion. Unless stated otherwise, $\mathcal{H}$ is equipped with the norm $\|h\| := \|\gamma\|_2$ for $h = \langle \gamma, \Phi \rangle$. Given a differentiable function $\varphi : \mathcal{H} \to \mathbb{R}$, its directional derivative at $f = \langle \gamma, \Phi \rangle \in \mathcal{H}$ in the direction defined by the function $g \in \mathcal{H}$ is defined as $\frac{d}{d\varepsilon}\big|_{\varepsilon=0} \varphi(f + \varepsilon g)$. Note that if we write $g = \langle \tilde{\gamma}, \Phi \rangle$ for some $\tilde{\gamma} \in \mathbb{R}^d$, then the directional derivative of $\varphi$ at $f$ equals $\langle \tilde{\gamma}, \nabla_\gamma \varphi(\gamma) \rangle$, where $\nabla_\gamma \varphi(\gamma)$ denotes the gradient of $\varphi : \mathbb{R}^d \to \mathbb{R}$ evaluated at $\gamma \in \mathbb{R}^d$. When it is clear from context, we drop the subscript $\lambda$ from the risks $L_\lambda$ and $\hat{L}_\lambda$.

# E  Conditions

**Condition 1.** *Suppose $C_\Phi = \sup_{x \in \mathcal{X}} \|\Phi(x)\|_2$ is finite.*

**Condition 2.** *For the population covariance matrix $\Sigma = \mathbb{E}_1[\Phi\Phi^\top]$, we have $\lambda_{\min}(\Sigma) > 0$ and $\lambda_{\max}(\Sigma)$ is of constant order, not depending on the sample size, or any other problem parameter.*

**Condition 3.** *For the sample covariance matrix $\hat{\Sigma} = \frac{1}{n_3} \sum_{k=1}^{n_3} \Phi(x_k)\Phi(x_k)^\top$, we have both $\lambda_{\min}(\hat{\Sigma}) \geqslant c_{\min} > 0$ and $\lambda_{\max}(\hat{\Sigma}) \leqslant c_{\max}$ of constant order with probability $1 - o(n_3^{-1})$.*

**Condition 4.** *Defining $C_1$ as in (7) in Appendix H, assume there exists an upper bound $C_{1,\text{upper}}$ on $\mathbb{E}[C_1]$ of constant order.*

**Condition 5.** *The conditional density $f_{S|X=x}$ exists for all $x \in \mathcal{X}$, and $C_f = \sup_{x \in \mathcal{X}} \|f_{S|X=x}(s)\|_\infty$ is a finite constant.*

The following can be interpreted as an independence assumption on the basis functions.

**Condition 6.** *Suppose $\inf_{v \in S^{d-1}} \mathbb{E}_1[|\langle v, \Phi \rangle|] \geqslant c_{\text{indep}} > 0$ for some constant $c_{\text{indep}}$.*

**Condition 7.** *Suppose $\frac{\mathbb{E}_1[rh_0^*]}{\mathbb{E}_1[|h_0^*|^2]^{1/2}} \geqslant c_{\text{align}} > 0$ for some minimizer $h_0^*$ of the objective in Equation (18) with regularization $\lambda = 0$.*

**Condition 8.** *Suppose $\mathbb{E}_1[r^2]$ is finite.*

**Condition 9.** *The constant function $h : \mathcal{X} \to \mathbb{R}$ given by $h(x) = 1$ for all $x \in \mathcal{X}$ is in $\mathcal{H}$.*

The following ensures that the zero function $0 \in \mathcal{H}$ is not a minimizer of the objective in Equation (LR-QR).

**Condition 10.** *For each $\lambda \geqslant 0$, there exists $h \in \mathcal{H}$ and $\beta \in \mathbb{R}$ such that*
$$\mathbb{E}_1[\ell_\alpha(h, S)] + \lambda \mathbb{E}_1[(\beta h - r)^2] < \mathbb{E}_1[\ell_\alpha(0, S)] + \lambda \mathbb{E}_1[r^2].$$

## F Constants

The following are the constants that appear in Theorem 4.2:
$$\rho_1 := 2\beta_{\max}^2 BC_\Phi^2 + 2\beta_{\max}C_\Phi, \quad \mu_1 := 2\beta_{\min}^2 c_{\min}, \quad \rho_2 := (1-\alpha)C_\Phi,$$

$$\widetilde{C}_1 := \frac{4\rho_1^2}{\mu_1}, \quad \widehat{C}_2 := \frac{4\rho_2^2}{2\beta_{\min}^2 c_{\min}}, \quad A_1 := \sqrt{\frac{64\widetilde{C}_1 a_1}{\delta}}, \quad A_2 := \sqrt{\frac{128\widehat{C}_2 a_2}{\delta}}.$$

Further,

$$A_3 := (1-\alpha)(BC_\Phi + 1)\sqrt{\frac{1}{2}\log\frac{8}{\delta}}, \qquad A_4 := \sqrt{2}(\beta_{\max}BC_\Phi)\sqrt{\frac{1}{2}\log\frac{16}{\delta}} \max\{\beta_{\max}BC_\Phi, 4\},$$

$$A_5 := A_1 + A_4, \quad a_1 := 2C_\Phi(C_{2,\text{upper}} + C_{2,\max})(1 + \beta_{\max}BC_\Phi), \quad a_2 := (1-\alpha)C_\Phi(C_{1,\text{upper}} + C_{1,\max}).$$

The following are the constants that appear in Theorem 4.3:

$$A_6 := 2\beta_{\max}^2\sqrt{4B^2\lambda_{\max}(\Sigma)}, \quad A_7 := \sqrt{4B^2\beta_{\max}^2\lambda_{\max}(\Sigma)}, \quad A_8 := \sqrt{2B^2C_f\lambda_{\max}(\Sigma)}, \quad A_9 := A_6 + A_7,$$

and

$$A_{10} := A_9 A_5^{1/2}, \quad A_{11} := \max\{A_9 A_3^{1/2}, A_8 A_5^{1/2}\},$$
$$A_{12} := A_9 A_2^{1/2}, \quad A_{13} := A_8 A_3^{1/2}, \quad A_{14} := A_8 A_2^{1/2}.$$

## G Generalization bound for regularized loss

The following is a generalization of Shalev-Shwartz and Ben-David [48, Corollary 13.6].

**Lemma G.1** (Generalization bound for regularized loss; extension of [48])**.** *Fix a compact and convex hypothesis class $\tilde{\mathcal{H}}$ equipped with a norm $\|\cdot\|_{\tilde{\mathcal{H}}}$, a compact interval $\mathcal{I} \subseteq \mathbb{R}$, and a sample space $\mathcal{Z}$. Consider the objective function $f : \tilde{\mathcal{H}} \times \mathcal{I} \times \mathcal{Z} \to \mathbb{R}$ given by $(h, \beta, z) \mapsto f(h, \beta, z) := \mathcal{J}(h, \beta, z) + \mathcal{R}(h, \beta)$, where $\mathcal{R} : \tilde{\mathcal{H}} \times \mathcal{I} \to \mathbb{R}$ is a regularization function, and $\mathcal{J} : \tilde{\mathcal{H}} \times \mathcal{I} \times \mathcal{Z} \to \mathbb{R}$ can be decomposed as $\mathcal{J}(h, \beta, z) := \mathcal{J}_1(h, \beta, z_1) + \mathcal{J}_2(h, \beta, z_2)$ for two functions $\mathcal{J}_1, \mathcal{J}_2 : \tilde{\mathcal{H}} \times \mathcal{I} \times \mathcal{Z} \to \mathbb{R}$.*

*Given distributions $\mathcal{D}_1, \mathcal{D}_2$ on $\mathcal{Z}$, let $\mathcal{L} : \tilde{\mathcal{H}} \times \mathcal{I} \to \mathbb{R}$ be given for all $h, \beta$ by*
$$\mathcal{L}(h, \beta) = \mathbb{E}_{Z_1 \sim \mathcal{D}_1, Z_2 \sim \mathcal{D}_2}[f(h, \beta, Z_1, Z_2)]$$

*denote the population risk, averaging over independent datapoints $Z_1 \sim \mathcal{D}_1$ and $Z_2 \sim \mathcal{D}_2$. Suppose that for both $Z \sim \mathcal{D}_1$ and $Z \sim \mathcal{D}_2$, $|\mathcal{J}_1(h, \beta, Z)|$ and $|\mathcal{J}_2(h, \beta, Z)|$ are almost surely bounded by a quantity not depending on $h \in \tilde{\mathcal{H}}$ and $\beta \in \mathcal{I}$.*

Let $\hat{\mathcal{L}} : \tilde{\mathcal{H}} \times \mathcal{I} \to \mathbb{R}$ denote the empirical risk computed over $Z_{i,1} \overset{i.i.d.}{\sim} \mathcal{D}_1$, $i \in [m_1]$ and $Z_{j,2} \overset{i.i.d.}{\sim} \mathcal{D}_2$, $j \in [m_2]$, given by

$$\hat{\mathcal{L}}(h, \beta) := \frac{1}{m_1} \sum_{i=1}^{m_1} \mathcal{J}_1(h, \beta, Z_{i,1}) + \frac{1}{m_2} \sum_{j=1}^{m_2} \mathcal{J}_2(h, \beta, Z_{j,2}) + \mathcal{R}(h, \beta).$$

Assume that for each fixed $\beta \in \mathcal{I}$ and $z \in \mathcal{Z}$,

- $h \mapsto \mathcal{J}_1(h, \beta, z)$ is convex and $\rho$-Lipschitz with respect to the norm $\| \cdot \|_{\tilde{\mathcal{H}}}$,

- $h \mapsto \mathcal{J}_2(h, \beta, z)$ is convex and $\rho$-Lipschitz with respect to the norm $\| \cdot \|_{\tilde{\mathcal{H}}}$, and

- $h \mapsto \hat{\mathcal{L}}(h, \beta)$ is $\mu$-strongly convex with respect to the norm $\| \cdot \|_{\tilde{\mathcal{H}}}$ with probability $1 - o(m_1^{-1} + m_2^{-1})$,

where the deterministic values $\mu = \mu(\beta)$ and $\rho = \rho(\beta)$ may depend on $\beta$.

Let $(\hat{h}, \hat{\beta})$ denote an ERM, i.e., a minimizer of $\hat{\mathcal{L}}(h, \beta)$ over $\tilde{\mathcal{H}} \times \mathcal{I}$. Let $\hat{h}_\beta$ denote a minimizer of the empirical risk in $h$ for fixed $\beta$.

Suppose the stochastic process $\beta \mapsto W_\beta$ given by $W_\beta = \mathcal{L}(\hat{h}_\beta, \beta) - \hat{\mathcal{L}}(\hat{h}_\beta, \beta)$ for $\beta \in \mathcal{I}$ obeys $|W_\beta - W_{\beta'}| \leqslant K|\beta - \beta'|$ for all $\beta, \beta' \in \mathcal{I}$ for some random variable $K$, and suppose that the probability of $K_{m_1,m_2} \leqslant K_{\max}$ converges to unity as $m_1, m_2 \to \infty$, for some constant $K_{\max}$. Suppose that there exists a constant $C > 0$ such that for all $\beta \in \mathcal{I}$,

$$\frac{4\rho(\beta)^2}{\mu(\beta)} \leqslant C. \tag{4}$$

Then for sufficiently large $m_1, m_2$, with probability at least $1 - \delta$,

$$|\mathcal{L}(\hat{h}, \hat{\beta}) - \hat{\mathcal{L}}(\hat{h}, \hat{\beta})| \leqslant \sqrt{\frac{16CK_{\max}}{\delta}(m_1^{-1} + m_2^{-1})}.$$

*Remark* G.2. A special case is when we do not have any data from $\mathcal{D}_2$, and instead all $m_1$ datapoints are sampled i.i.d. from $\mathcal{D}_1$. In this case, defining with a slight abuse of notation $\mathcal{J} := \mathcal{J}_1$, the statement simplifies to the analysis of the empirical risk

$$\hat{\mathcal{L}}(h, \beta) := \frac{1}{m_1} \sum_{i=1}^{m_1} \mathcal{J}(h, \beta, Z_{i,1}) + \mathcal{R}(h, \beta).$$

If for each fixed $\beta \in \mathcal{I}$, we have that $h \mapsto \mathcal{J}(h, \beta, z)$ is convex and $\rho$-Lipschitz with respect to the norm $\| \cdot \|_{\tilde{\mathcal{H}}}$, and if $|\mathcal{J}(h, \beta, Z)|$ is almost surely bounded by a quantity not depending on $h \in \tilde{\mathcal{H}}$ and $\beta \in \mathcal{I}$ for $Z \sim \mathcal{D}_1 = \mathcal{D}_2$, then under the remaining assumptions, we obtain the slightly stronger bound

$$|\mathcal{L}(\hat{h}, \hat{\beta}) - \hat{\mathcal{L}}(\hat{h}, \hat{\beta})| \leqslant \sqrt{\frac{16CK_{\max}}{\delta m_1}}.$$

We omit the proof, because it is exactly as below.

*Remark* G.3. We relax the strong convexity assumption on the regularizer $\mathcal{R}$ from Shalev-Shwartz and Ben-David [48, Corollary 13.6], substituting it with the less restrictive condition of strong convexity of the empirical loss $\hat{\mathcal{L}}$. In order to use assumptions that merely hold with high probability, we impose a boundedness condition on $\mathcal{J}$.

*Proof.* Fix $\beta$ and let $E$ denote the event that $h \mapsto \hat{\mathcal{L}}(h, \beta)$ is $\mu$-strongly convex in $h$. By assumption, $E$ occurs with probability $1 - o(m_1^{-1} + m_2^{-1})$.

We modify the proof of Shalev-Shwartz and Ben-David [48, Corollary 13.6] as follows. Let $Z_1' \sim \mathcal{D}_1$ and $Z_2' \sim \mathcal{D}_2$ be drawn independently from all other randomness. For a fixed $i \in [m_1]$, let $h \mapsto \hat{\mathcal{L}}_{i,1}(h, \beta)$ denote the empirical risk computed from the sample $(Z_{1,1}, \ldots, Z_{i-1,1}, Z_1', Z_{i+1,1}, \ldots, Z_{m_1,1}) \cup (Z_{1,2}, \ldots, Z_{m_2,2})$, and let $\hat{h}_\beta^{(i)}$ denote an ERM for this

sample. Let $I$ be drawn from $[m_1]$ uniformly at random. The variables $J, \hat{\mathcal{L}}_{J,2}(h, \beta), \hat{h}_\beta^{(J)}$ are defined similarly but for the sample from $\mathcal{D}_2$.

Note that for fixed $\beta$, similarly to the argument in Shalev-Shwartz and Ben-David [48, Theorem 13.2], we have

$$\mathbb{E}[\mathcal{L}(\hat{h}_\beta, \beta)] = \mathbb{E}_{Z_1' \sim \mathcal{D}_1, Z_2' \sim \mathcal{D}_2}[\mathcal{J}_1(\hat{h}_\beta, \beta, Z_1') + \mathcal{J}_2(\hat{h}_\beta, \beta, Z_2') + \mathcal{R}(\hat{h}_\beta, \beta)]$$

$$= \mathbb{E}_{Z_1' \sim \mathcal{D}_1, Z_2' \sim \mathcal{D}_2}[\mathcal{J}_1(\hat{h}_\beta^{(I)}, \beta, Z_{I,1}) + \mathcal{J}_2(\hat{h}_\beta^{(J)}, \beta, Z_{J,2}) + \mathcal{R}(\hat{h}_\beta, \beta)]$$

and

$$\mathbb{E}[\hat{\mathcal{L}}(\hat{h}_\beta, \beta)] = \mathbb{E}[\mathcal{J}_1(\hat{h}_\beta, \beta, Z_{I,1}) + \mathcal{J}_2(\hat{h}_\beta, \beta, Z_{J,2}) + \mathcal{R}(\hat{h}_\beta, \beta)].$$

Therefore

$$\mathbb{E}[\mathcal{L}(\hat{h}_\beta, \beta) - \hat{\mathcal{L}}(\hat{h}_\beta, \beta)] = (\mathbb{E}[\mathcal{J}_1(\hat{h}_\beta^{(I)}, \beta, Z_{I,1}) - \mathcal{J}_1(\hat{h}_\beta, \beta, Z_{I,1})])$$

$$+ (\mathbb{E}[\mathcal{J}_2(\hat{h}_\beta^{(J)}, \beta, Z_{J,2}) - \mathcal{J}_2(\hat{h}_\beta, \beta, Z_{J,2})]).$$

Further, splitting the expectations over $E$ and its complement $E^c$, this further equals

$$(\mathbb{E}[(\mathcal{J}_1(\hat{h}_\beta^{(I)}, \beta, Z_{I,1}) - \mathcal{J}_1(\hat{h}_\beta, \beta, Z_{I,1}))\mathbf{1}[E]] + \mathbb{E}[(\mathcal{J}_1(\hat{h}_\beta^{(I)}, \beta, Z_{I,1}) - \mathcal{J}_1(\hat{h}_\beta, \beta, Z_{I,1}))\mathbf{1}[E^c]]) \tag{5}$$

$$+ (\mathbb{E}[(\mathcal{J}_2(\hat{h}_\beta^{(J)}, \beta, Z_{J,2}) - \mathcal{J}_2(\hat{h}_\beta, \beta, Z_{J,2}))\mathbf{1}[E]] + \mathbb{E}[(\mathcal{J}_2(\hat{h}_\beta^{(J)}, \beta, Z_{J,2}) - \mathcal{J}_2(\hat{h}_\beta, \beta, Z_{J,2}))\mathbf{1}[E^c]]).$$

On the event $E$, $h \mapsto \hat{\mathcal{L}}(h, \beta)$ is $\mu$-strongly convex. Now, consider the setting of Shalev-Shwartz and Ben-David [48, Corollary 13.6]. We claim that the arguments in their proof hold if we replace the regularizer $h \mapsto \lambda\|h\|^2$ by $h \mapsto \mathcal{R}(h, \beta)$, as they only leverage the strong convexity of the overall empirical loss $\hat{\mathcal{L}}$. Indeed, working on the event $E$, since $\hat{\mathcal{L}}$ is $\mu$-strongly convex, we have that $\hat{\mathcal{L}}(h) - \hat{\mathcal{L}}(\hat{h}_\beta) \geqslant \frac{1}{2}\mu\|h - \hat{h}_\beta\|^2$ for all $h \in \tilde{\mathcal{H}}$. Next, for any $h_1, h_2 \in \tilde{\mathcal{H}}$, we have

$$\hat{\mathcal{L}}(h_2) - \hat{\mathcal{L}}(h_1) = \hat{\mathcal{L}}_{I,1}(h_2) - \hat{\mathcal{L}}_{I,1}(h_1) + \frac{\mathcal{J}_1(h_2, \beta, Z_{I,1}) - \mathcal{J}_1(h_1, \beta, Z_{I,1})}{m_1}$$

$$- \frac{\mathcal{J}_1(h_2, \beta, Z_1') - \mathcal{J}_1(h_1, \beta, Z_1')}{m_1}.$$

Setting $h_2 = \hat{h}_\beta^{(I)}$ and $h_1 = \hat{h}$, since $\hat{h}_\beta^{(I)}$ minimizes $h \mapsto \hat{\mathcal{L}}_{I,1}(h, \beta)$, and using our lower bound on $\hat{\mathcal{L}}(h) - \hat{\mathcal{L}}(\hat{h}_\beta)$, we deduce

$$\frac{1}{2}\mu\|\hat{h}_\beta^{(I)} - \hat{h}_\beta\|^2 \leqslant \frac{\mathcal{J}_1(\hat{h}_\beta^{(I)}, \beta, Z_{I,1}) - \mathcal{J}_1(\hat{h}_\beta, \beta, Z_{I,1})}{m_1} - \frac{\mathcal{J}_1(\hat{h}_\beta^{(I)}, \beta, Z_1') - \mathcal{J}_1(\hat{h}_\beta, \beta, Z_1')}{m_1}. \tag{6}$$

Since by assumption, $h \mapsto \mathcal{J}_1(h, \beta, z)$ is $\rho$-Lipschitz, we have the bounds $|\mathcal{J}_1(\hat{h}_\beta^{(I)}, \beta, Z_{I,1}) - \mathcal{J}_1(\hat{h}_\beta, \beta, Z_{I,1})| \leqslant \rho|\hat{h}_\beta^{(I)} - \hat{h}_\beta|$ and $|\mathcal{J}_1(\hat{h}_\beta^{(I)}, \beta, Z_1') - \mathcal{J}_1(\hat{h}_\beta, \beta, Z_1')| \leqslant \rho|\hat{h}_\beta^{(I)} - \hat{h}_\beta|$. Plugging these into Equation (6), we obtain $\frac{1}{2}\mu\|\hat{h}_\beta^{(I)} - \hat{h}_\beta\|^2 \leqslant \frac{2\rho}{m_1}\|\hat{h}_\beta^{(I)} - \hat{h}_\beta\|$, so that $\|\hat{h}_\beta^{(I)} - \hat{h}_\beta\| \leqslant \frac{4\rho(\beta)}{\mu(\beta)m_1}$. Using once again that $h \mapsto \mathcal{J}_1(h, \beta, z)$ is $\rho$-Lipschitz, we find $|\mathcal{J}_1(\hat{h}_\beta^{(I)}, \beta, Z_{I,1}) - \mathcal{J}_1(\hat{h}_\beta, \beta, Z_{I,1})| \leqslant \frac{4\rho(\beta)^2}{\mu(\beta)m_1}$.

Similarly, on the event $E$, we have the bound $|\mathcal{J}_2(\hat{h}_\beta^{(J)}, \beta, Z_{J,2}) - \mathcal{J}_2(\hat{h}_\beta, \beta, Z_{J,2})| \leqslant \frac{4\rho(\beta)^2}{\mu(\beta)m_2}$. Thus the first and third terms are bounded in magnitude by $\frac{4\rho(\beta)^2}{\mu(\beta)m_1}$ and $\frac{4\rho(\beta)^2}{\mu(\beta)m_2}$, respectively. Due to (4), their sum is at most $C(m_1^{-1} + m_2^{-1})$.

By our assumption that $|\mathcal{J}_1(h, \beta, Z)|$ and $|\mathcal{J}_2(h, \beta, Z)|$ are almost surely bounded by a constant for both $Z \sim \mathcal{D}_1$ and $Z \sim \mathcal{D}_2$, and our assumption that $\mathbb{P}[E^c] = o(m_1^{-1} + m_2^{-1})$, the second term and fourth terms from (5) sum to $o(m_1^{-1} + m_2^{-1})$. Thus for for each $\beta$, for sufficiently large $m_1, m_2$, we have $\mathbb{E}[|W_\beta|] \leqslant 2C(m_1^{-1} + m_2^{-1})$. By Markov's inequality, for any fixed $t > 0$, $|W_\beta| > t$ with

probability at most $\frac{2C}{t}(m_1^{-1} + m_2^{-1})$. We now use chaining. Let $N$ be an $\varepsilon$-net for $\mathcal{I}$. Then using the fact that by assumption, the process $W$ is $K_{m_1,m_2}$-Lipschitz, and by a union bound,

$$\mathbb{P}\left[\sup_{\beta \in \mathcal{I}} |W_\beta| > K_{m_1,m_2}\varepsilon + t\right] \leqslant \mathbb{P}\left[\sup_{\beta \in N} |W_\beta| > t\right] \leqslant |N|\frac{2C}{t}(m_1^{-1} + m_2^{-1}).$$

Pick $N$ with $|N| = 1/\varepsilon$, and set $t = \frac{4C}{\delta}(m_1^{-1} + m_2^{-1})\frac{1}{\varepsilon}$. We deduce that

$$\sup_{\beta \in \mathcal{I}} |W_\beta| > K_{m_1,m_2}\varepsilon + \frac{4C}{\delta}(m_1^{-1} + m_2^{-1})\frac{1}{\varepsilon}$$

with probability at most $\frac{\delta}{2}$. Set $\varepsilon = \sqrt{\frac{4C}{K_{m_1,m_2}\delta}(m_1^{-1} + m_2^{-1})}$. We deduce that

$$\sup_{\beta \in \mathcal{I}} |W_\beta| > \sqrt{\frac{16CK_{m_1,m_2}}{\delta}(m_1^{-1} + m_2^{-1})}$$

with probability at most $\frac{\delta}{2}$. Since the probability of $K_{m_1,m_2} \leqslant K_{\max}$ converges to unity, for sufficiently large $m_1, m_2$,

$$\sup_{\beta \in \mathcal{I}} |W_\beta| > \sqrt{\frac{16CK_{\max}}{\delta}(m_1^{-1} + m_2^{-1})}$$

holds with probability at most $\delta$. Since $|W_{\hat\beta}| \leqslant \sup_{\beta \in \mathcal{I}} |W_\beta|$, we may conclude. $\square$

# H   Lipschitz process

**Lemma H.1** (Lipschitzness of minimizer of perturbed strongly convex objective)**.** *Let $\mathcal{C} \subseteq \mathbb{R}^d$ be a closed convex set. Suppose $\psi : \mathcal{C} \to \mathbb{R}$ is $\mu$-strongly convex and $g : \mathcal{C} \to \mathbb{R}$ is $L$-smooth. Suppose also that $\psi + g$ is convex. Let $x_\psi$ denote the minimizer of $\psi$ in $\mathcal{C}$, and let $x_{\psi+g}$ denote the minimizer of $\psi + g$ in $\mathcal{C}$. Then for any $x \in \mathcal{C}$,*

$$\|x_{\psi+g} - x_\psi\|_2 \leqslant \frac{1}{\mu}(L\|x_{\psi+g} - x\|_2 + \|\nabla g(x)\|_2).$$

*Proof.* Since $\psi$ is $\mu$-strongly convex and since $x_{\psi+g}, x_\psi$ are minimizers of $\psi + g, \psi$ respectively,

$$\begin{aligned}
\mu\|x_{\psi+g} - x_\psi\|_2^2 &\leqslant \langle \nabla\psi(x_{\psi+g}) - \nabla\psi(x_\psi), x_{\psi+g} - x_\psi\rangle \\
&= \langle \nabla(\psi + g)(x_{\psi+g}), x_{\psi+g} - x_\psi\rangle + \langle \nabla\psi(x_\psi), x_\psi - x_{\psi+g}\rangle \\
&\quad - \langle \nabla g(x_{\psi+g}), x_{\psi+g} - x_\psi\rangle \\
&\leqslant -\langle \nabla g(x_{\psi+g}), x_{\psi+g} - x_\psi\rangle \\
&= -\langle \nabla g(x_{\psi+g}) - \nabla g(x), x_{\psi+g} - x_\psi\rangle - \langle \nabla g(x), x_{\psi+g} - x_\psi\rangle,
\end{aligned}$$

so that by $L$-smoothness of $g$,

$$\mu\|x_{\psi+g} - x_\psi\|_2^2 \leqslant (L\|x_{\psi+g} - x\|_2 + \|\nabla g(x)\|_2)\|x_{\psi+g} - x_\psi\|_2,$$

which implies the result. $\square$

**Lemma H.2** (Lipschitzness of minimizer of perturbed ERM)**.** *Under Condition 1, with $\hat\Sigma$ from Condition 3, and with the notations of Lemma H.4, we have with respect to the norm $\|\cdot\|$ on $\mathcal{H}_B$ that $\beta \mapsto \hat{h}_\beta$ is $C_1$-Lipschitz on $\mathcal{I}$, and $\beta \mapsto \beta\hat{h}_\beta$ is $C_2$-Lipschitz on $\mathcal{I}$, where*

$$C_1 = (\beta_{\min}^2\lambda_{\min}(\hat\Sigma))^{-1}((2\beta_{\max}\lambda_{\max}(\hat\Sigma)B + C_\Phi) + 4\beta_{\max}\lambda_{\max}(\hat\Sigma)B), \quad C_2 = B + \beta_{\max}C_1. \quad (7)$$

*Proof.* First, consider $\hat{h}_\beta$. Fix $\beta > \beta'$ in $\mathcal{I}$. Recalling the definition of $\hat{L}$ from (Empirical-LR-QR), the difference between the objectives $\hat{L}(h, \beta)$ and $\hat{L}(h, \beta')$ is the quadratic

$$g(h) := \hat{L}(h, \beta) - \hat{L}(h, \beta') = \lambda\hat{\mathbb{E}}_3[(\beta^2 - (\beta')^2)h^2] + \lambda\hat{\mathbb{E}}_2[-2(\beta - \beta')h].$$

We claim that $g$ is $2\lambda(\beta^2 - (\beta')^2)\lambda_{\min}(\hat{\Sigma})$-strongly convex and $2\lambda(\beta^2 - (\beta')^2)\lambda_{\max}(\hat{\Sigma})$-smooth in $h$. To see this, write $h = \langle \gamma, \Phi \rangle$ for $\gamma \in \mathbb{R}^d$, and note that $g$ can be rewritten as

$$g(\gamma) = \lambda(\beta^2 - (\beta')^2)\gamma^\top \hat{\Sigma}\gamma - 2(\beta - \beta')\lambda\gamma^\top \hat{\mathbb{E}}_2[\Phi],$$

a quadratic whose Hessian equals $2\lambda(\beta^2 - (\beta')^2)\hat{\Sigma}$, which implies the claim.

Similarly, we claim that the function $\psi(h) := \hat{L}(h, \beta')$ is $2\lambda(\beta')^2\lambda_{\min}(\hat{\Sigma})$-strongly convex in $h$. To see this, again write $h = \langle \gamma, \Phi \rangle$ for $\gamma \in \mathbb{R}^d$, and note that $\psi$ can be rewritten as

$$\psi(\gamma) = \lambda(\beta')^2\gamma^\top \hat{\Sigma}\gamma + \hat{\mathbb{E}}_1[\ell_\alpha(\gamma^\top\Phi, S)] + \lambda\hat{\mathbb{E}}_2[-2\beta\gamma^\top\Phi].$$

By Lemma N.3, the second term is convex, and since the third term is linear, it too is convex. The Hessian of the quadratic first term is $2\lambda(\beta')^2\hat{\Sigma}$, from which it follows that $\psi$ is $2\lambda(\beta')^2\lambda_{\min}(\hat{\Sigma})$-strongly convex.

Thus $\psi$ and $g$ satisfy the conditions of Lemma H.1, which implies the bound

$$\|\hat{h}_\beta - \hat{h}_{\beta'}\| \leqslant (2\lambda(\beta')^2\lambda_{\min}(\hat{\Sigma}))^{-1}(\|\nabla g(\hat{h}_g)\|_2 + 2\lambda(\beta^2 - (\beta')^2)\lambda_{\max}(\hat{\Sigma}) \cdot \|\hat{h}_\beta - \hat{h}_g\|), \quad (8)$$

where $\hat{h}_g = \hat{h}_{g,\beta,\beta'}$ denotes the minimizer of $g$ in $\mathcal{H}_B$. Since

$$\nabla g(\gamma) = \lambda(\beta - \beta')((\beta + \beta')2\hat{\Sigma}\gamma - 2\hat{\mathbb{E}}_2[\Phi]),$$

and by $|\beta|, |\beta'| \leqslant \beta_{\max}$, $\|\gamma\| \leqslant B$, and Condition 1, we have

$$\|\nabla g(\gamma)\|_2 \leqslant \lambda(4\beta_{\max}\lambda_{\max}(\hat{\Sigma})B + 2C_\Phi)|\beta - \beta'|$$

for $\beta, \beta' \in \mathcal{I}$ and $h \in \mathcal{H}_B$. Plugging this into the bound (8) on $\|\hat{h}_\beta - \hat{h}_{\beta'}\|$ and using the fact that $\beta' \geqslant \beta_{\min}$ and $\|\hat{h}_\beta\|, \|\hat{h}_g\| \leqslant B$,

$$\|\hat{h}_\beta - \hat{h}_{\beta'}\| \leqslant$$
$$(2\lambda\beta_{\min}^2\lambda_{\min}(\hat{\Sigma}))^{-1}(\lambda(4\beta_{\max}\lambda_{\max}(\hat{\Sigma})B + 2C_\Phi)|\beta - \beta'| + 8\lambda\beta_{\max}\lambda_{\max}(\hat{\Sigma})B|\beta - \beta'|).$$

Thus we may take

$$C_1 = (\beta_{\min}^2\lambda_{\min}(\hat{\Sigma}))^{-1}((2\beta_{\max}\lambda_{\max}(\hat{\Sigma})B + C_\Phi) + 4\beta_{\max}\lambda_{\max}(\hat{\Sigma})B).$$

For the map $\beta \mapsto \beta\hat{h}_\beta$, fix $\beta > \beta'$ in $\mathcal{I}$, and write $\|\beta\hat{h}_\beta - \beta'\hat{h}_{\beta'}\| \leqslant |\beta - \beta'|\|\hat{h}_\beta\| + |\beta'|\|\hat{h}_\beta - \hat{h}_{\beta'}\|$. For the first term, note that since $\hat{h}_\beta \in \mathcal{H}_B$ implies $\|\hat{h}_\beta\| \leqslant B$, the first term is bounded by $B|\beta - \beta'|$. For the second term, note that since $|\beta'| \leqslant \beta_{\max}$ and since $\beta \mapsto \hat{h}_\beta$ is $C_1$-Lipschitz on $\mathcal{I}$, the second term is bounded by $\beta_{\max}C_1|\beta - \beta'|$. Summing, we deduce that $\beta \mapsto \beta\hat{h}_\beta$ is $C_2$-Lipschitz on $\mathcal{I}$, where $C_2 = B + \beta_{\max}C_1$. $\qquad\square$

**Lemma H.3** (Lipschitzness of minimizer of perturbed auxiliary ERM). *Under Condition 1, we have that $\beta \mapsto \tilde{h}_\beta$ is $C_1$-Lipschitz on $\mathcal{I}$, and $\beta \mapsto \beta\tilde{h}_\beta$ is $C_2$-Lipschitz on $\mathcal{I}$.*

*Proof.* The proof is almost identical to Lemma H.2. $\qquad\square$

Recalling $c_{\min}$ and $c_{\max}$ from Condition 3, define

$$C_{1,\max} = (\beta_{\min}^2 c_{\min})^{-1}((2\beta_{\max}c_{\max}B + C_\Phi) + 4\beta_{\max}c_{\max}B), \qquad C_{2,\max} = B + \beta_{\max}C_{1,\max}, \quad (9)$$

so that by Condition 3, $C_1 \leqslant C_{1,\max}$ and $C_2 \leqslant C_{2,\max}$ with probability tending to unity over the randomness in $\mathcal{S}_3$.

We now compute the Lipschitz constants of the processes used in the proof of Theorem 4.2.

Recall $\bar{L}$ from (13), $\hat{L}$ from (Empirical-LR-QR), $\tilde{L}$ from (12), and $\mathcal{H}_B = \{\langle \gamma, \Phi \rangle : \|\gamma\|_2 \leqslant B < \infty\}$ from Section 4. For any fixed $\beta \in \mathcal{I}$, define $\hat{h}_\beta$ as the minimizer of $h \mapsto \hat{L}(h, \beta)$ over $\mathcal{H}_B$, which exists under the conditions of Theorem 4.2 due to our argument checking the convexity of $h \mapsto \hat{L}(h, \beta)$ in Term (I) in the proof of Theorem 4.2.

**Lemma H.4.** *Assume the conditions of Theorem 4.2. Define the stochastic processes $\bar{W}_\beta$ and $\tilde{W}_\beta$ on $\mathcal{I}$ given by $\beta \mapsto (\bar{L} - \hat{L})(\hat{h}_\beta, \beta)$ and $\beta \mapsto (\tilde{L} - \hat{L})(\hat{h}_\beta, \beta)$, respectively. Then $\bar{W}_\beta$ is $K_{1,\lambda}$-Lipschitz on $\mathcal{I}$ with probability tending to unity as $n_1, n_2, n_3 \to \infty$, and $\tilde{W}_\beta$ is $K_{2,\lambda}$-Lipschitz on $\mathcal{I}$ with probability tending to unity as $n_1, n_2, n_3 \to \infty$, where*

$$K_{1,\lambda} := 2C_\Phi(C_{2,\mathrm{upper}} + C_{2,\mathrm{max}})(1 + \beta_{\max} BC_\Phi)\lambda =: a_1\lambda,$$
$$K_{2,\lambda} := (1 - \alpha)C_\Phi(C_{1,\mathrm{upper}} + C_{1,\mathrm{max}}) =: a_2,$$

*with $C_{1,\mathrm{max}}$ and $C_{2,\mathrm{max}}$ are defined in (9) and where $C_{1,\mathrm{upper}}$ satisfies Condition 4 and $C_{2,\mathrm{upper}} := BC_\Phi + \beta_{\max}C_\Phi C_{1,\mathrm{upper}}$. In fact, $\bar{W}$ is $K_{1,\lambda}$-Lipschitz on $\mathcal{I}$ with probability tending to unity conditional on $\mathcal{S}_1$, and $\tilde{W}$ is $K_{2,\lambda}$-Lipschitz on $\mathcal{I}$ deterministically, when conditioning on $\mathcal{S}_2, \mathcal{S}_3$, when the event $C_1 \leqslant C_{1,\mathrm{max}}$ holds.*

*Proof.* We start with the process $\tilde{W}$. Consider $\beta, \beta' \in \mathcal{I}$. Note that for any $(h, \beta)$, using the definition of $\tilde{L}$ from (12), we have the identity

$$\tilde{L}(h, \beta) - \hat{L}(h, \beta) = \mathbb{E}_1[\ell_\alpha(h, S)] - \hat{\mathbb{E}}_1[\ell_\alpha(h, S)].$$

Thus we may write

$$\tilde{W}_\beta - \tilde{W}_{\beta'} = (\mathbb{E}_1[\ell_\alpha(\hat{h}_\beta, S)] - \mathbb{E}_1[\ell_\alpha(\hat{h}_{\beta'}, S)]) - (\hat{\mathbb{E}}_1[\ell_\alpha(\hat{h}_\beta, S)] - \hat{\mathbb{E}}_1[\ell_\alpha(\hat{h}_{\beta'}, S)]),$$

so that

$$|\tilde{W}_\beta - \tilde{W}_{\beta'}| \leqslant \mathbb{E}_1[|\ell_\alpha(\hat{h}_\beta, S) - \ell_\alpha(\hat{h}_{\beta'}, S)|] + \hat{\mathbb{E}}_1[|\ell_\alpha(\hat{h}_\beta, S) - \ell_\alpha(\hat{h}_{\beta'}, S)|] \qquad (10)$$

Note that we have the uniform bound

$$|\ell_\alpha(\hat{h}_\beta, S) - \ell_\alpha(\hat{h}_{\beta'}, S)| \leqslant (1 - \alpha)|\hat{h}_\beta - \hat{h}_{\beta'}|$$
$$\leqslant (1 - \alpha)C_\Phi\|\hat{h}_\beta - \hat{h}_{\beta'}\| \leqslant (1 - \alpha)C_\Phi C_1|\beta - \beta'|,$$

where in the first step we applied Lemma N.2, in the second step we used Condition 1 to apply Lemma N.4, and in the third step we used Lemma H.2. Thus the first term in Equation (10) is bounded by $(1 - \alpha)C_\Phi\mathbb{E}_1[C_1]|\beta - \beta'|$, and the second term in Equation (10) is bounded by $(1 - \alpha)C_\Phi\hat{\mathbb{E}}_1[C_1]|\beta - \beta'|$. Summing, we deduce that

$$|\tilde{W}_\beta - \tilde{W}_{\beta'}| \leqslant (1 - \alpha)C_\Phi(\mathbb{E}_1[C_1] + \hat{\mathbb{E}}_1[C_1])|\beta - \beta'|,$$

so that the process $\tilde{W}$ is $K_2$-Lipschitz with $K_2 := (1 - \alpha)C_\Phi(\mathbb{E}_1[C_1] + \hat{\mathbb{E}}_1[C_1])$.

We now condition on $\mathcal{S}_2, \mathcal{S}_3$. Observe that $C_1, C_2$ are $\mathcal{S}_3$-measurable (as $\hat{\Sigma}$ from Condition 3 is $\mathcal{S}_3$-measurable). Since $\mathbb{E}_1[C_1] \leqslant C_{1,\mathrm{upper}}$, on the event that $C_1 \leqslant C_{1,\mathrm{max}}$, we have $K_2 \leqslant K_{2,\lambda}$, where $K_{2,\lambda} = (1 - \alpha)C_\Phi(C_{1,\mathrm{upper}} + C_{1,\mathrm{max}})$, as claimed.

We now continue with the process $\bar{W}$. Consider $\beta, \beta' \in \mathcal{I}$. Note that for any $(h, \beta)$, using the definition of $\bar{L}$ from Equation (13), we have the identity

$$\bar{L}(h, \beta) - \hat{L}(h, \beta) = (\lambda\mathbb{E}_3[\beta^2 h^2] + \lambda\mathbb{E}_2[-2\beta h]) - (\lambda\hat{\mathbb{E}}_3[\beta^2 h^2] + \lambda\hat{\mathbb{E}}_2[-2\beta h]).$$

Thus we may write

$$\bar{W}_\beta - \bar{W}_{\beta'} = \lambda(\mathbb{E}_3[\beta^2 \hat{h}_\beta^2] - \mathbb{E}_3[(\beta')^2 \hat{h}_{\beta'}^2]) + \lambda(\mathbb{E}_2[-2\beta\hat{h}_\beta] - \mathbb{E}_2[-2\beta'\hat{h}_{\beta'}])$$
$$- \lambda(\hat{\mathbb{E}}_3[\beta^2 \hat{h}_\beta^2] - \hat{\mathbb{E}}_3[(\beta')^2 \hat{h}_{\beta'}^2]) - \lambda(\hat{\mathbb{E}}_2[-2\beta\hat{h}_\beta] - \hat{\mathbb{E}}_2[-2\beta'\hat{h}_{\beta'}]),$$

so that

$$|\bar{W}_\beta - \bar{W}_{\beta'}| \leqslant \lambda\mathbb{E}_3[|\beta^2 \hat{h}_\beta^2 - (\beta')^2 \hat{h}_{\beta'}^2|] + 2\lambda\mathbb{E}_2[|\beta\hat{h}_\beta - \beta'\hat{h}_{\beta'}|]$$
$$+ \lambda\hat{\mathbb{E}}_3[|\beta^2 \hat{h}_\beta^2 - (\beta')^2 \hat{h}_{\beta'}^2|] + 2\lambda\hat{\mathbb{E}}_2[|\beta\hat{h}_\beta - \beta'\hat{h}_{\beta'}|] \qquad (11)$$

The integrands of the first and third terms of Equation (11) can be uniformly bounded as

$$|\beta^2 \hat{h}_\beta^2 - (\beta')^2 \hat{h}_{\beta'}^2| \leqslant |\beta\hat{h}_\beta - \beta'\hat{h}_{\beta'}| \cdot |\beta\hat{h}_\beta + \beta'\hat{h}_{\beta'}| \leqslant C_\Phi\|\beta\hat{h}_\beta - \beta'\hat{h}_{\beta'}\| \cdot C_\Phi\|\beta\hat{h}_\beta + \beta'\hat{h}_{\beta'}\|$$
$$\leqslant C_\Phi C_2|\beta - \beta'| \cdot 2C_\Phi\beta_{\max}B = 2\beta_{\max}BC_\Phi^2 C_2|\beta - \beta'|.$$

where in the first step we used difference of squares, in the second step we used Condition 1 to apply Lemma N.4, in the third step we applied Lemma H.2 to bound the first factor and the triangle inequality and the bounds $\beta \leqslant \beta_{\max}$ for $\beta \in \mathcal{I}$ and $\|h\| \leqslant B$ for $h \in \mathcal{H}_B$ to bound the second factor. The integrand of the second and fourth term in (11) can be bounded as $|\beta \hat{h}_\beta - \beta' \hat{h}_{\beta'}| \leqslant C_\Phi \|\beta \hat{h}_\beta - \beta' \hat{h}_{\beta'}\| \leqslant C_\Phi C_2 |\beta - \beta'|$, where in the first step we used Condition 1 to apply Lemma N.4, and in the second step we applied Lemma H.2.

Plugging these into our bound in Equation (11), we deduce

$$|\bar{W}_\beta - \bar{W}_{\beta'}| \leqslant (2C_\Phi(\mathbb{E}_2[C_2] + \hat{\mathbb{E}}_2[C_2]) + 2\beta_{\max} C_\Phi^2 B(\mathbb{E}_3[C_2] + \hat{\mathbb{E}}_3[C_2]))\lambda|\beta - \beta'|,$$

so that the process $\bar{W}$ is $K_1$-Lipschitz with

$$K_1 = (2C_\Phi(\mathbb{E}_2[C_2] + \hat{\mathbb{E}}_2[C_2]) + 2\beta_{\max} C_\Phi^2 B(\mathbb{E}_3[C_2] + \hat{\mathbb{E}}_3[C_2]))\lambda.$$

We now work conditional on $\mathcal{S}_1$. On the event that $C_1 \leqslant C_{1,\max}$ and $C_2 \leqslant C_{2,\max}$, and by Condition 4, we have $K_1 \leqslant K_{1,\max}$, where

$$\begin{aligned} K_{1,\lambda} &= (2C_\Phi(C_{2,\text{upper}} + C_{2,\max}) + 2\beta_{\max} C_\Phi^2 B(C_{2,\text{upper}} + C_{2,\max}))\lambda \\ &= 2C_\Phi(C_{2,\text{upper}} + C_{2,\max})(1 + \beta_{\max} BC_\Phi)\lambda. \end{aligned}$$

Since $C_1 \leqslant C_{1,\max}$ and $C_2 \leqslant C_{2,\max}$ with probability tending to one due to Condition 3, $K_1 \leqslant K_{1,\lambda}$ and $K_2 \leqslant K_{2,\lambda}$ both hold with probability tending to one if we uncondition on $\mathcal{S}_1$, and we are done. $\qquad\square$

## I  Proof of Proposition 4.1

Fix $\lambda \geqslant 0$. Under the assumptions of Lemma L.3, there exists a global minimizer $(h^*, \beta^*)$ of $L(h, \beta)$. The first order condition with respect to $\beta$ reads $2\lambda\mathbb{E}_1[h^*(X)(\beta^* h^*(X) - r(X))] = 0$. By Lemma N.5, the first order condition with respect to $h$ reads

$$\mathbb{E}_1[h^*(X)(\mathbb{P}_{S|X}[S(X,Y) \leqslant h^*(X)] - (1-\alpha))] + 2\lambda\mathbb{E}_1[\beta^* h(X)(\beta^* h^*(X) - r(X))] = 0$$

for all $h \in \mathcal{H}$. Setting $h = r_{\mathcal{H}}$ in the second equation, and subtracting $(\beta^*)^2$ times the first equation from the second, we deduce that

$$\begin{aligned} &\mathbb{E}_1[h^*(X)(\mathbb{P}_{S|X}[S(X,Y) \leqslant h^*(X)] - (1-\alpha))] \\ &+ 2\lambda\mathbb{E}_1[\beta^* \cdot r_{\mathcal{H}}(X) \cdot (\beta^* h^*(X) - r(X))] - 2\lambda\mathbb{E}_1[\beta^* \cdot \beta^* h^*(X) \cdot (\beta^* h^*(X) - r(X))] \\ &= \mathbb{E}_1[h^*(X)(\mathbb{P}_{S|X}[S(X,Y) \leqslant h^*(X)] - (1-\alpha))] \\ &+ 2\lambda\mathbb{E}_1[\beta^*(r_{\mathcal{H}}(X) - \beta^* h^*(X))(\beta^* h^*(X) - r(X))] \\ &= \mathbb{E}_1[h^*(X)\mathbb{P}_{S|X}[S(X,Y) \leqslant h^*(X)]] - (1-\alpha) - 2\lambda\beta^*\mathbb{E}_1[(r_{\mathcal{H}}(X) - \beta^* h^*(X))^2] = 0. \end{aligned}$$

Therefore,

$$\mathbb{E}_1[r_{\mathcal{H}}(X)\mathbb{P}_{S|X}[S(X,Y) \leqslant h^*(X)]] = (1-\alpha) + 2\lambda\beta^*\mathbb{E}_1[(r_{\mathcal{H}}(X) - \beta^* h^*(X))^2],$$

which implies the result.

## J  Proof of Theorem 4.2

Recall that $\mathcal{S}_1$ are the features of the labeled calibration dataset. We also recall the notation $\mathbb{E}_j$ and $\hat{\mathbb{E}}_j$ for $j = 1, 2, 3$ from Section 2. Given the unlabeled test data $\mathcal{S}_2$ and the unlabeled calibration data $\mathcal{S}_3$, define the auxiliary risks for $h \in \mathcal{H}_B, \beta \in \mathcal{I}$,

$$\tilde{L}(h, \beta; \mathcal{S}_2, \mathcal{S}_3) := \mathbb{E}_1[\ell_\alpha(h, S)] + \lambda\hat{\mathbb{E}}_3[\beta^2 h^2] + \lambda\hat{\mathbb{E}}_2[-2\beta h] \tag{12}$$

and

$$\bar{L}(h, \beta; \mathcal{S}_1) := \hat{\mathbb{E}}_1[\ell_\alpha(h, S)] + \lambda\mathbb{E}_3[\beta^2 h^2] + \lambda\mathbb{E}_2[-2\beta h]. \tag{13}$$

Let

$$(\tilde{h}, \tilde{\beta}) \in \arg\min_{h \in \mathcal{H}_B, \beta \in \mathcal{I}} \tilde{L}(h, \beta; \mathcal{S}_2, \mathcal{S}_3). \tag{14}$$

For convenience, we leave implicit the dependence of $\tilde{L}$ and $(\tilde{h}, \tilde{\beta})$ on $\mathcal{S}_2, \mathcal{S}_3$ and the dependence of $\bar{L}$ on $\mathcal{S}_1$.

In order to study the generalization error, we write

$$L(\hat{h}, \hat{\beta}) - L(h^*, \beta^*) = (L(\hat{h}, \hat{\beta}) - \tilde{L}(\hat{h}, \hat{\beta})) + (\tilde{L}(\hat{h}, \hat{\beta}) - \hat{L}(\hat{h}, \hat{\beta})) + (\hat{L}(\hat{h}, \hat{\beta}) - \hat{L}(\tilde{h}, \tilde{\beta}))$$
$$+ (\hat{L}(\tilde{h}, \tilde{\beta}) - \tilde{L}(\tilde{h}, \tilde{\beta})) + (\tilde{L}(\tilde{h}, \tilde{\beta}) - \tilde{L}(h^*, \beta^*)) + (\tilde{L}(h^*, \beta^*) - L(h^*, \beta^*)).$$

Since $(\hat{h}, \hat{\beta})$ is a minimizer of the risk $\hat{L}$, we have $\hat{L}(\hat{h}, \hat{\beta}) - \hat{L}(\tilde{h}, \tilde{\beta}) \leqslant 0$, and since $(\tilde{h}, \tilde{\beta})$ is a minimizer of the risk $\tilde{L}$, we have $\tilde{L}(\tilde{h}, \tilde{\beta}) - \tilde{L}(h^*, \beta^*) \leqslant 0$. Thus our generalization error is bounded by the remaining four terms:

$$L(\hat{h}, \hat{\beta}) - L(h^*, \beta^*) \leqslant (L(\hat{h}, \hat{\beta}) - \tilde{L}(\hat{h}, \hat{\beta})) + (\tilde{L}(\hat{h}, \hat{\beta}) - \hat{L}(\hat{h}, \hat{\beta}))$$
$$+ (\hat{L}(\tilde{h}, \tilde{\beta}) - \tilde{L}(\tilde{h}, \tilde{\beta})) + (\tilde{L}(h^*, \beta^*) - L(h^*, \beta^*))$$
$$=: (I) + (II) + (III) + (IV). \tag{15}$$

We study the generalization error by conditioning on the unlabeled calibration or test data. Then our regularization becomes data-independent. Conditional on $\mathcal{S}_1$, Term (I) can be handled with Lemma G.1 above. Conditional on $\mathcal{S}_2, \mathcal{S}_3$, Term (II) can be handled with Lemma G.1 above. Terms (III) and (IV) are empirical processes at fixed functions, conditional on $\mathcal{S}_2, \mathcal{S}_3$.

**Term (I):** We work conditional on $\mathcal{S}_1$. First, note that due to the definition of $\hat{L}$ from (Empirical-LR-QR), we can write for any $(h, \beta)$,

$$L(h, \beta) - \tilde{L}(h, \beta) = \bar{L}(h, \beta) - \hat{L}(h, \beta).$$

Since $\bar{L}(h, \beta) - \hat{L}(h, \beta)$ can be viewed as a difference of a population risk $\lambda \mathbb{E}_3[\beta^2 h^2] + \lambda \mathbb{E}_2[-2\beta h]$ and an empirical risk $\lambda \hat{\mathbb{E}}_3[\beta^2 h^2] + \lambda \hat{\mathbb{E}}_2[-2\beta h]$ with "regularizer" $\hat{\mathbb{E}}_1[\ell_\alpha(h, S)]$, this expression enables us to apply Lemma G.1 to bound $\bar{L}(\hat{h}, \hat{\beta}) - \hat{L}(\hat{h}, \hat{\beta})$.

Explicitly, we can write

$$\frac{1}{\lambda} \hat{L}(h, \beta) = \hat{\mathbb{E}}_3[\beta^2 h^2] + \hat{\mathbb{E}}_2[-2\beta h] + \frac{1}{\lambda} \hat{\mathbb{E}}_1[\ell_\alpha(h, S)].$$

Hence, fixing $\beta$, we can apply Lemma G.1, choosing $m_1 = n_3$ and $m_2 = n_2$. Further, we choose $\tilde{\mathcal{H}} := \mathcal{H}_B = \{\langle \gamma, \Phi \rangle : \|\gamma\|_2 \leqslant B < \infty\}$ with the norm $\langle \gamma, \Phi \rangle = \|\gamma\|_2$. Moreover, letting $z = (x'', x')$ for $x'', x' \in \mathcal{X}$, and $\xi = 1/\lambda$, we use the objective function given by $(h, z) \mapsto f_1(h, z) = \mathcal{J}(h, \beta, z) + \mathcal{R}(h, \beta)$, where $\mathcal{J}(h, \beta, z) = \mathcal{J}_1(h, \beta, z) + \mathcal{J}_2(h, \beta, z)$, and where

$$\mathcal{J}_1(h, \beta, z) = \beta^2 h(x'')^2, \qquad \mathcal{J}_2(h, \beta, z) = -2\beta h(x'), \qquad \mathcal{R}(h, \beta) = \xi \hat{\mathbb{E}}_1[\ell_\alpha(h, S)].$$

We now check the conditions of Lemma G.1.

*Boundedness:* Note that $|\mathcal{J}_1(h, \beta, z)| = |\beta|^2 |h(x'')|^2 \leqslant \beta_{\max}^2 (BC_\Phi)^2$, where in the second step we used $|\beta| \leqslant \beta_{\max}$ for $\beta \in \mathcal{I}$, and we used $h \in \mathcal{H}_B$ and Condition 1 to apply Lemma N.4. Similarly, note that $|\mathcal{J}_2(h, \beta, z)| = 2|\beta||h(x')| \leqslant 2\beta_{\max} BC_\Phi$, where in the second step we used $|\beta| \leqslant \beta_{\max}$ for $\beta \in \mathcal{I}$, and we used $h \in \mathcal{H}_B$ and Condition 1 to apply Lemma N.4. Thus $|\mathcal{J}_1(h, \beta, z)|$ and $|\mathcal{J}_2(h, \beta, z)|$ are both bounded by the sum $\beta_{\max}^2 (BC_\Phi)^2 + 2\beta_{\max} BC_\Phi$.

*Convexity:* Write $h = \langle \gamma, \Phi \rangle$ for $\gamma \in \mathbb{R}^d$. The map $h \mapsto \mathcal{J}_1(h, \beta, z)$ can equivalently be written as $\gamma \mapsto \beta^2 \gamma^\top \Phi(x'') \Phi(x'')^\top \gamma$, a quadratic whose Hessian equals the positive semidefinite matrix $2\beta^2 \Phi(x'') \Phi(x'')^\top$. Thus $h \mapsto \mathcal{J}_1(h, \beta, z)$ is convex. The map $h \mapsto \mathcal{J}_2(h, \beta, z)$ can equivalently be written as $\gamma \mapsto -2\beta \gamma^\top \Phi(x')$, which is linear, hence convex.

*Lipschitzness:* Write $h = \langle \gamma, \Phi \rangle$ for $\gamma \in \mathbb{R}^d$. The map $h \mapsto \mathcal{J}_1(h, \beta, z)$ can equivalently be written as $\gamma \mapsto \beta^2 \gamma^\top \Phi(x'') \Phi(x'')^\top \gamma$. The gradient of this quadratic is given by $\gamma \mapsto 2\beta^2 \Phi(x'') \Phi(x'')^\top \gamma$. The norm of this gradient can be bounded by

$$\|2\beta^2 \Phi(x'') \Phi(x'')^\top \gamma\|_2 \leqslant 2|\beta|^2 \|\Phi(x'')\|_2^2 \|\gamma\|_2 \leqslant 2\beta_{\max}^2 BC_\Phi^2,$$

where in the first step we applied the Cauchy-Schwarz inequality, in the second step we used $|\beta| \leqslant \beta_{\max}$ for $\beta \in \mathcal{I}$, $\|\gamma\|_2 \leqslant B$, and Condition 1. Next, the map $h \mapsto \mathcal{J}_2(h, \beta, z)$ can equivalently

be written as $\gamma \mapsto -2\beta\gamma^\top \Phi(x')$. The gradient of this linear map is given by $\gamma \mapsto -2\beta\Phi(x')$. The norm of this gradient can be bounded by $2|\beta|\|\Phi(x')\| \leqslant 2\beta_{\max}C_\Phi$, where we used $|\beta| \leqslant \beta_{\max}$ for $\beta \in \mathcal{I}$ and Condition 1. Thus the norm of each of these gradients is bounded by the sum $\rho_1 := 2\beta_{\max}^2 BC_\Phi^2 + 2\beta_{\max}C_\Phi$, and the maps $h \mapsto \mathcal{J}_1(h, \beta, z)$ and $h \mapsto \mathcal{J}_2(h, \beta, z)$ are both $\rho_1$-Lipschitz.

*Strong convexity:* Since $h \mapsto \ell_\alpha(h, s)$ is convex for all $s \in \mathbb{R}$ by Lemma N.3 and since $h \mapsto \hat{\mathbb{E}}_2[\beta h]$ is linear, the map $h \mapsto \xi\hat{\mathbb{E}}_1[\ell_\alpha(h, S)] - 2\hat{\mathbb{E}}_2[\beta h]$ is convex. Consider the map $h \mapsto \hat{\mathbb{E}}_3[\beta^2 h^2]$. Writing $h = \langle \gamma, \Phi \rangle$ for $\gamma \in \mathbb{R}^d$, this can be rewritten as $\gamma \mapsto \beta^2\gamma^\top\hat{\Sigma}\gamma$, a quadratic whose Hessian equals $2\beta^2\hat{\Sigma}$. By $\beta \geqslant \beta_{\min}$ for $\beta \in \mathcal{I}$ and Condition 3, it follows that with probability $1 - o(n_3^{-1}) = 1 - o(n_2^{-1} + n_3^{-1})$, the map $h \mapsto \hat{\mathbb{E}}_{2,3}[f_1(h, Z)]$ is $\mu_1$-strongly convex, where $Z = (X'', X')$ with $X'$ is uniform over $\mathcal{X}_2$ and $X''$ is uniform over $\mathcal{X}_3$, and where $\mu_1 := 2\beta_{\min}^2 c_{\min}$. In particular, $h \mapsto \frac{1}{\lambda}\hat{L}(h, \beta)$ is convex.

Let $\widetilde{C}_1 = \frac{4\rho_1^2}{\mu_1}$. Let $K_1$ denote the Lipschitz constant of the process $\bar{W}_\beta$, where $K_1 \leqslant K_{1,\lambda}$ with probability tending to unity conditional on $\mathcal{S}_1$ by Condition 4 and Lemma H.4. From Lemma G.1 applied with $\xi = 1/\lambda$, $\mathcal{L} = \frac{1}{\lambda}\bar{L}$, and $\hat{\mathcal{L}} = \frac{1}{\lambda}\hat{L}$, and $W = (\bar{L} - \hat{L})/\lambda$, we obtain that conditional on $\mathcal{S}_1$, for sufficiently large $n_2, n_3$, with probability at least $1 - \frac{\delta}{4}$, we have for Term (I) from (15),

$$\frac{1}{\lambda}\text{Term (I)} \leqslant \sqrt{\frac{16\widetilde{C}_1 K_{1,\lambda}/\lambda}{\delta/4}\left(\frac{1}{n_2} + \frac{1}{n_3}\right)}.$$

Thus

$$\text{Term (I)} \leqslant \sqrt{\frac{64\widetilde{C}_1\lambda K_{1,\lambda}}{\delta}\left(\frac{1}{n_2} + \frac{1}{n_3}\right)} = A_1\lambda\sqrt{\frac{1}{n_2} + \frac{1}{n_3}},$$

where we define $A_1 = \sqrt{\frac{64\widetilde{C}_1 a_1}{\delta}}$. Since the right-hand side does not depend on $\mathcal{S}_1$, the same bound holds when we uncondition on $\mathcal{S}_1$.

**Term (II):** We work conditional on $\mathcal{S}_2, \mathcal{S}_3$. The risks $\hat{L}$ and $\tilde{L}$ share the same data-independent regularization $\lambda\hat{\mathbb{E}}_3[\beta^2 h^2] + \lambda\hat{\mathbb{E}}_2[-2\beta h]$. Write $z = (x, s)$ for $x \in \mathcal{X}$ and $s \in [0, 1]$. Fixing $\beta$, we apply Lemma G.1 with the objective function $(h, z) \mapsto f(h, z) = \mathcal{J}(h, \beta, z) + \mathcal{R}(h, \beta)$, where

$$\mathcal{J}(h, \beta, z) = \ell_\alpha(h(x), s), \qquad \mathcal{R}(h, \beta) = \lambda\hat{\mathbb{E}}_3[\beta^2 h^2] + \lambda\hat{\mathbb{E}}_2[-2\beta h].$$

Since the empirical risk $\hat{L}$ is computed over the i.i.d. sample $Z_i = (X_i, S_i)$ for $i \in [n_1]$, we use the modified version of Lemma G.1 given in Remark G.2. In particular, we check boundedness, convexity, and Lipschitzness of $\mathcal{J}$ without writing it as a sum $\mathcal{J}_1 + \mathcal{J}_2$.

*Boundedness:* we have the uniform bound, for all $h, \beta, z$

$$|\mathcal{J}(h, \beta, z)| \leqslant (1 - \alpha)|h(x) - s| \leqslant (1 - \alpha)(|h(x)| + 1) \leqslant (1 - \alpha)(BC_\Phi + 1), \qquad (16)$$

where in the first step we used Lemma N.1, in the second step we used the triangle inequality and $s \in [0, 1]$, and in the third step we used $h \in \mathcal{H}_B$ and Condition 1 to apply Lemma N.4.

*Convexity:* By Lemma N.3, $h \mapsto \mathcal{J}(h, \beta, z)$ is convex.

*Lipschitzness:* Fix $h = \langle \gamma, \Phi \rangle$ and $h' = \langle \gamma', \Phi \rangle$ in $\mathcal{H}_B$, where $\gamma, \gamma' \in \mathbb{R}^d$. Note that

$$|\mathcal{J}(h, \beta, z) - \mathcal{J}(h, \beta, z)| = |\ell_\alpha(h(x), s) - \ell_\alpha(h'(x), s)|$$
$$\leqslant (1 - \alpha)|h(x) - h'(x)| \leqslant (1 - \alpha)C_\Phi\|h - h'\|,$$

where in the second step we used Lemma N.2, and in the third step we used Condition 1 to apply Lemma N.4. Thus $h \mapsto \mathcal{J}(h, \beta, z)$ is $\rho_2$-Lipschitz, where $\rho_2 := (1 - \alpha)C_\Phi$.

*Strong convexity:* To analyze $\mathcal{R}$, first observe that since $h \mapsto \lambda\hat{\mathbb{E}}_2[-2\beta h]$ is linear, it is convex. Writing $h = \langle \gamma, \Phi \rangle$ for $\gamma \in \mathbb{R}^d$, the term $h \mapsto \lambda\hat{\mathbb{E}}_3[\beta^2 h^2]$ in $\mathcal{R}$ can be rewritten as $\gamma \mapsto \lambda\beta^2\gamma^\top\hat{\Sigma}\gamma$, a quadratic whose Hessian equals $2\lambda\beta^2\hat{\Sigma}$. By $\beta \geqslant \beta_{\min}$ for $\beta \in \mathcal{I}$ and Condition 3, it follows

that with probability $1 - o(n_3^{-1})$ over $\mathcal{S}_2, \mathcal{S}_3$, the map $h \mapsto \mathcal{R}(h, \beta)$ is $\mu_2$-strongly convex, where $\mu_2(\lambda) := 2\lambda \beta_{\min}^2 c_{\min}$.

Let $\widetilde{C}_2(\lambda) = \frac{4\rho_2^2}{\mu_2(\lambda)}$. Let $K_2$ denote the Lipschitz constant of the process $W_\beta$; recall that conditional on $\mathcal{S}_2, \mathcal{S}_3$, $K_2 \leqslant K_{2,\lambda}$ deterministically on the event $C_1 \leqslant C_{1,\max}$ by Lemma H.4. By the version of Lemma G.1 given in Remark G.2, conditional on $\mathcal{S}_2, \mathcal{S}_3$, if $h \mapsto \mathcal{R}(h, \beta)$ is $\mu_2(\lambda)$-strongly convex, and if $C_1 \leqslant C_{1,\max}$, then for sufficiently large $n_1$, with probability at least $1 - \frac{\delta}{8}$, we have

$$\text{Term (II)} \leqslant \sqrt{\frac{16\widetilde{C}_2(\lambda)K_{2,\lambda}}{(\delta/8)n_1}} = \frac{A_2}{\sqrt{\lambda n_1}}, \tag{17}$$

where we define $A_2 = \sqrt{\frac{128\widehat{C}_2 a_2}{\delta}}$ and $\widehat{C}_2 = \frac{4\rho_2^2}{2\beta_{\min}^2 c_{\min}}$. Unconditioning on $\mathcal{S}_2, \mathcal{S}_3$, since $\mathcal{R}(h, \beta)$ is $\mu_2(\lambda)$-strongly convex with probability tending to unity by the above analysis, and since by Condition 3 we have $C_1 \leqslant C_{1,\max}$ with probability tending to unity, we deduce that for sufficiently large $n_1, n_2, n_3$, with probability at least $1 - \frac{\delta}{4}$, (17) still holds.

**Term (III):** We work conditional on $\mathcal{S}_2, \mathcal{S}_3$. Since $\tilde{h}$ from (14) lies in $\mathcal{H}_B$, we may use the bound in Equation (16) to obtain $\sup_{x \in \mathcal{X}} |\ell_\alpha(\tilde{h}, S)| \leqslant (1 - \alpha)(BC_\Phi + 1)$. Thus by Hoeffding's inequality [21], with probability at least $1 - \frac{\delta}{4}$ we have

$$(\hat{L} - \tilde{L})(\tilde{h}, \tilde{\beta}) = (\hat{\mathbb{E}}_1 - \mathbb{E}_1)[\ell_\alpha(\tilde{h}, S)] \leqslant \frac{(1 - \alpha)(BC_\Phi + 1)\sqrt{\frac{1}{2}\log\frac{2}{\delta/4}}}{\sqrt{n_1}}.$$

Thus we have Term (III) $\leqslant \frac{A_3}{\sqrt{n_1}}$, where we define $A_3 = (1 - \alpha)(BC_\Phi + 1)\sqrt{\frac{1}{2}\log\frac{8}{\delta}}$.

**Term (IV):** Note that we may write

$$(\tilde{L} - L)(h^*, \beta^*) = (\hat{\mathbb{E}}_2 - \mathbb{E}_2)[\lambda(\beta^* h^*)^2] + (\hat{\mathbb{E}}_3 - \mathbb{E}_3)[-2\lambda \beta^* h^*].$$

Since $\|h^*\| \leqslant B$ by $h^* \in \mathcal{H}_B$ and since Condition 1 holds, we may apply Lemma N.4 to deduce that $\sup_{x \in \mathcal{X}} |h^*(x)| \leqslant BC_\Phi$. Consequently, for $\beta \in \mathcal{I}$, we have the uniform bound $\sup_{x \in \mathcal{X}} |\beta h^*(x)| \leqslant \beta_{\max} BC_\Phi$. By Hoeffding's inequality [21], with probability at least $1 - \frac{\delta}{8}$, we have

$$|(\hat{\mathbb{E}}_2 - \mathbb{E}_2)[\lambda(\beta^* h^*)^2]| \leqslant \frac{\lambda(\beta_{\max} BC_\Phi)^2\sqrt{\frac{1}{2}\log\frac{2}{\delta/8}}}{\sqrt{n_2}}.$$

By another application of Hoeffding's inequality, with probability at least $1 - \frac{\delta}{8}$, we have

$$|(\hat{\mathbb{E}}_3 - \mathbb{E}_3)[-2\lambda \beta^* h^*]| \leqslant \frac{4\lambda(\beta_{\max} BC_\Phi)\sqrt{\frac{1}{2}\log\frac{2}{\delta/8}}}{\sqrt{n_3}}.$$

Summing, with probability at least $1 - \delta$ we have the bound

$$(\tilde{L} - L)(h^*, \beta^*) \leqslant \frac{\lambda(\beta_{\max} BC_\Phi)^2\sqrt{\frac{1}{2}\log\frac{16}{\delta}}}{\sqrt{n_2}} + \frac{4\lambda(\beta_{\max} BC_\Phi)\sqrt{\frac{1}{2}\log\frac{16}{\delta}}}{\sqrt{n_3}}.$$

Using the inequality $a + b \leqslant \sqrt{2}\sqrt{a^2 + b^2}$ for all $a, b \in \mathbb{R}$, we deduce Term (IV) $\leqslant A_4\lambda\sqrt{\frac{1}{n_2} + \frac{1}{n_3}}$, where we define

$$A_4 = \sqrt{2}(\beta_{\max} BC_\Phi)\sqrt{\frac{1}{2}\log\frac{16}{\delta}}\max\{\beta_{\max} BC_\Phi, 4\}.$$

Returning to the analysis of (15), and summing all four terms while defining $A_5 = A_1 + A_4$, with probability at least $1 - \delta$ we obtain a generalization error bound of

$$L(\hat{h}, \hat{\beta}) - L(h^*, \beta^*) \leqslant A_5\lambda\sqrt{\frac{1}{n_2} + \frac{1}{n_3}} + A_3\frac{1}{\sqrt{n_1}} + A_2\frac{1}{\sqrt{\lambda}}\frac{1}{\sqrt{n_1}}.$$

The result follows by taking $c = A_5$, $c' = A_3$, and $c'' = A_2$.

# K  Proof of Theorem 4.3

We use the following result to convert the generalization error bound in Theorem 4.2 to a coverage lower bound.

**Lemma K.1** (Bounded suboptimality implies bounded gradient for smooth functions). *Let $f : \mathbb{R}^{d'} \to \mathbb{R}$, for some positive $d'$. Suppose $x^*$ is a global minimizer of $f$. Suppose $x'$ is such that $f(x') \leqslant f(x^*) + \varepsilon$. Suppose $h \in \mathbb{R}^d$ is such that the map $g : \mathbb{R} \to \mathbb{R}$ given by $t \mapsto f(x' + th)$ is $L$-smooth, i.e. $|g''(h)|$ is uniformly bounded by $L$. Then*

$$|f'(x'; h)| = |\nabla f(x')^\top h| \leqslant \sqrt{2L\varepsilon}\|h\|_2.$$

*Proof.* Assume there exists $h$ and $\delta > 0$ with $f'(x'; h) > \delta\|h\|$. Setting $y = x' - th$,

$$f(x' - th) \leqslant f(x') - tf'(x'; h) + \tfrac{L}{2}t^2\|h\|^2.$$

Set $t = \delta/(L\|h\|)$ to obtain

$$f(x' - th) \leqslant f(x') - \tfrac{\delta^2}{L} + \tfrac{\delta^2}{2L} = f(x') - \tfrac{\delta^2}{2L}.$$

Since $f(x') \leqslant f(x^*) + \varepsilon$, we have $f(x' - th) \leqslant f(x^*) + \varepsilon - \tfrac{\delta^2}{2L}$. If $\delta > \sqrt{2L\varepsilon}$, then $f(x' - th) < f(x^*)$, a contradiction.

A similar argument with $f'(x'; h) < -\delta\|h\|$ and $y = x' + th$ yields the same contradiction. Hence $-\sqrt{2L\varepsilon}\|h\| \leqslant f'(x'; h) \leqslant \sqrt{2L\varepsilon}\|h\|$. $\qquad\qquad\square$

By Condition 1 and Condition 5, we may apply Lemma N.5 to deduce that the Hessian of our population risk $L$ from (LR-QR) in the basis $\{\phi_1, \ldots, \phi_d\}$ is the block matrix

$$\nabla^2 L(h, \beta) = \begin{bmatrix} \mathbb{E}_1[\Phi\Phi^\top(f_{S|X}(h) + 2\lambda\beta^2)] & \mathbb{E}_1[2\lambda\Phi^\top(2\beta h - r)] \\ \mathbb{E}_1[2\lambda\Phi(2\beta h - r)] & \mathbb{E}_1[2\lambda h^2] \end{bmatrix}.$$

Thus by $\beta \leqslant \beta_{\max}$, $\|h\| \leqslant B$ for $h \in \mathcal{H}_B$, Condition 5, and Jensen's inequality, we have the uniform bounds

$$\sup_{h \in \mathcal{H}_B, \beta \in \mathbb{R}} |\partial_\beta^2 L(h, \beta)| \leqslant 2\lambda\mathbb{E}_1[h^2] \leqslant 2\lambda B^2\lambda_{\max}(\Sigma) =: \nu_1$$

and

$$\sup_{h \in \mathcal{H}, \beta \in \mathcal{I}} \|\nabla_h^2 L(h, \beta)\|_2 = \|\mathbb{E}_1[\Phi\Phi^\top(f_{S|X}(h) + 2\lambda\beta^2)]\|_2 \leqslant (C_f + 2\lambda\beta_{\max}^2)\lambda_{\max}(\Sigma) =: \nu_2.$$

By Lemma L.3 and Lemma L.4, a global minimizer of the objective in Equation (LR-QR) exists, and since $\beta_{\min} \leqslant \beta_{\text{lower}}$, $\beta_{\max} \geqslant \beta_{\text{upper}}$, and $B \geqslant B_{\text{upper}}$, any such minimizer lies in the interior of $\mathcal{H}_B \times \mathcal{I}$. Thus we may apply Lemma K.1 to the objective function $L$. We utilize two directional derivatives in the space $\mathcal{H} \times \mathbb{R}$. The first is in the direction $0_{\mathcal{H}} \times 1$, the unit vector in the $\beta$ coordinate. Since $(\hat{h}, \hat{\beta}) \in \mathcal{H}_B \times \mathcal{I}$, the magnitude of the second derivative of $L$ along this direction is bounded by $\nu_1$.

The second is in the direction of the vector $r_B \times 0$, where $r_B$ the projection of $r$ onto the closed convex set $\mathcal{H}_B$ in the Hilbert space induced by the inner product $\langle f, g \rangle = \mathbb{E}_1[fg]$. Since $(\hat{h}, \hat{\beta}) \in \mathcal{H}_B \times \mathcal{I}$, the magnitude of the second derivative of $L$ along this direction is bounded by $\nu_2$.

Given $\hat{h}$, let $\widehat{\text{Cover}}(X) := \mathbb{P}\left[S \leqslant \hat{h}(X)|X\right] - (1 - \alpha)$. Now, on the event $E$ that $L(\hat{h}, \hat{\beta}) - L(h^*, \beta^*) \leqslant \mathcal{E}_{\text{gen}}$, we apply Lemma K.1 with $f$ being $(\gamma, \beta) \mapsto L(h_\gamma, \beta)$, $x^*$ being $(h^*, \beta^*)$, $x'$ being $(\hat{h}, \hat{\beta})$, $\varepsilon = \mathcal{E}_{\text{gen}}$, and the directions specified above, with their respective smoothness parameters derived above. Using the formulas for $\nabla L$ from Lemma N.5 and the bound $\|r_B\| \leqslant B$, we obtain that on the event $E$,

$$|2\lambda\mathbb{E}_1[\hat{h}(\hat{\beta}\hat{h} - r)]| \leqslant \mathcal{E}_1, \qquad |\mathbb{E}_1[r_B\widehat{\text{Cover}}] + \lambda\mathbb{E}_1[2\beta r_B(\hat{\beta}\hat{h} - r)]| \leqslant \mathcal{E}_2,$$

where $\mathcal{E}_1 = \sqrt{2\nu_1\mathcal{E}_{\text{gen}}}$, $\mathcal{E}_2 = \sqrt{2B^2\nu_2\mathcal{E}_{\text{gen}}}$.

For any $h$ and $\beta$, we may write

$$\mathbb{E}_1[r_B\widehat{\mathrm{Cover}}] = (\mathbb{E}_1[r_B\widehat{\mathrm{Cover}}] + \lambda\mathbb{E}_1[2\beta r_B(\beta h - r)])$$
$$- \lambda\mathbb{E}_1[2\beta(\beta h)(\beta h - r)] - \lambda\mathbb{E}_1[2\beta(r_B - \beta h)(\beta h - r)].$$

Evaluating at $(\hat{h}, \hat{\beta})$, the first term is at most $\mathcal{E}_2$ in magnitude, the second term is at most $\hat{\beta}^2\mathcal{E}_1$ in magnitude, and the third term equals $2\hat{\beta}\lambda\mathbb{E}_1[(r_B - \hat{\beta}\hat{h})^2]$. We deduce

$$\mathbb{E}_1[r_B\widehat{\mathrm{Cover}}] \geqslant 2\hat{\beta}\lambda\mathbb{E}_1[(r_B - \hat{\beta}\hat{h})^2] - \hat{\beta}^2\mathcal{E}_1 - \mathcal{E}_2.$$

Since $\widehat{\mathrm{Cover}} \in [-(1 - \alpha), \alpha]$,

$$|\mathbb{E}_1[r\widehat{\mathrm{Cover}}] - \mathbb{E}_1[r_B\widehat{\mathrm{Cover}}]| \leqslant (1 - \alpha)\mathbb{E}_1[|r - r_B|].$$

We deduce that

$$\mathbb{E}_1[r\widehat{\mathrm{Cover}}] \geqslant 2\hat{\beta}\lambda\mathbb{E}_1[(r_B - \hat{\beta}\hat{h})^2] - \hat{\beta}^2\mathcal{E}_1 - \mathcal{E}_2 - (1 - \alpha)\mathbb{E}_1[|r - r_B|].$$

We now bound the quantity $\hat{\beta}^2\mathcal{E}_1 + \mathcal{E}_2$. First, since $\sqrt{a + b} \leqslant \sqrt{a} + \sqrt{b}$ for all $a, b \geqslant 0$, Theorem 4.2 implies that

$$\sqrt{\mathcal{E}_{\mathrm{gen}}} \leqslant A_5^{1/2}\lambda^{1/2}\left(\frac{1}{n_2} + \frac{1}{n_3}\right)^{1/4} + \frac{A_3^{1/2}}{n_1^{1/4}} + A_2^{1/2}\frac{1}{\lambda^{1/4}}\frac{1}{n_1^{1/4}}.$$

We may write $\mathcal{E}_1 = \sqrt{2\nu_1\mathcal{E}_{\mathrm{gen}}} = \sqrt{4B^2\lambda_{\max}(\Sigma)} \cdot \lambda^{1/2}\sqrt{\mathcal{E}_{\mathrm{gen}}}$, so that for $\hat{\beta} \in \mathcal{I}$ we have

$$\hat{\beta}^2\mathcal{E}_1 \leqslant \beta_{\max}^2\sqrt{4B^2\lambda_{\max}(\Sigma)} \cdot \lambda^{1/2}\sqrt{\mathcal{E}_{\mathrm{gen}}} =: A_6\lambda^{1/2}\sqrt{\mathcal{E}_{\mathrm{gen}}}.$$

Using the inequality $\sqrt{a + b} \leqslant \sqrt{a} + \sqrt{b}$ for all $a, b \geqslant 0$, we may bound

$$\mathcal{E}_2 = \sqrt{2B^2\nu_2\mathcal{E}_{\mathrm{gen}}} \leqslant \sqrt{4B^2\beta_{\max}^2\lambda_{\max}(\Sigma)} \cdot \lambda^{1/2}\sqrt{\mathcal{E}_{\mathrm{gen}}} + \sqrt{2B^2C_f\lambda_{\max}(\Sigma)} \cdot \sqrt{\mathcal{E}_{\mathrm{gen}}}$$
$$=: A_7\lambda^{1/2}\sqrt{\mathcal{E}_{\mathrm{gen}}} + A_8\sqrt{\mathcal{E}_{\mathrm{gen}}},$$

Thus

$$\hat{\beta}^2\mathcal{E}_1 + \mathcal{E}_2 \leqslant A_6\lambda^{1/2}\sqrt{\mathcal{E}_{\mathrm{gen}}} + A_7\lambda^{1/2}\sqrt{\mathcal{E}_{\mathrm{gen}}} + A_8\sqrt{\mathcal{E}_{\mathrm{gen}}}$$
$$=: A_9\lambda^{1/2}\sqrt{\mathcal{E}_{\mathrm{gen}}} + A_8\sqrt{\mathcal{E}_{\mathrm{gen}}}.$$

Plugging in our bound on $\sqrt{\mathcal{E}_{\mathrm{gen}}}$ and grouping terms according to the power of $\lambda$, we deduce that $\hat{\beta}^2\mathcal{E}_1 + \mathcal{E}_2 \leqslant \mathcal{E}_{\mathrm{cov}}$, where $\mathcal{E}_{\mathrm{cov}}$ equals

$$A_{10}\left(\frac{1}{n_2} + \frac{1}{n_3}\right)^{1/4}\lambda + A_{11}\left(\frac{1}{n_1^{1/4}} + \left(\frac{1}{n_2} + \frac{1}{n_3}\right)^{1/4}\right)\lambda^{1/2} + A_{12}\frac{\lambda^{1/4}}{n_1^{1/4}} + \frac{A_{13}}{n_1^{1/4}} + A_{14}\frac{\lambda^{-1/4}}{n_1^{1/4}}$$

and where $A_{10}, \ldots, A_{14}$ are the positive constants given in Appendix F. It follows that on the event $E$,

$$\mathbb{E}_1[r\widehat{\mathrm{Cover}}] \geqslant (1 - \alpha) + 2\hat{\beta}\lambda\mathbb{E}_1[(r_B - \hat{\beta}\hat{h})^2] - \mathcal{E}_{\mathrm{cov}} - (1 - \alpha)\mathbb{E}_1[|r - r_B|].$$

By Theorem 4.2, $E$ occurs with probability $1 - \delta$ for sufficiently large $n_1, n_2, n_3$, and we may conclude.

## L  Unconstrained existence and boundedness

In this section, we prove apriori existence and boundedness of unconstrained global minimizers of the population objective Equation (LR-QR). We write $(h_\lambda^*, \beta_\lambda^*)$ for a minimizer of the unconstrained objective in Equation (LR-QR) with regularization strength $\lambda \geqslant 0$.

In Lemma L.1, we show that under Condition 10, we may eliminate $\beta$ from Equation (LR-QR), so that Equation (LR-QR) is equivalent to solving the following unconstrained optimization problem over $h$:

$$\min_{h \in \mathcal{H}\setminus\{0\}} \mathbb{E}_1[\ell_\alpha(h, S)] - \lambda\frac{\mathbb{E}_1[rh]^2}{\mathbb{E}_1[h^2]}. \tag{18}$$

**Lemma L.1.** *Under Condition 10, for $\lambda \geqslant 0$, given any minimizer $(h^*_\lambda, \beta^*_\lambda)$ of the objective in Equation* (LR-QR) *with regularization $\lambda$, $h^*_\lambda$ is a minimizer of the objective in Equation* (18) *with regularization $\lambda$. Conversely, if $h$ is a minimizer of the objective in Equation* (18) *with regularization $\lambda$, then there exists a minimizer $(h^*_\lambda, \beta^*_\lambda)$ of the objective in Equation* (LR-QR) *with regularization $\lambda$ such that $h^*_\lambda = h$.*

*Proof.* By Condition 10, the minimization in Equation (LR-QR) with regularization $\lambda$ can be taken over $\mathcal{H} \setminus \{0\}$. Further, since the projection of $r$ onto $\mathrm{span}\{h\} := \{ch : c \in \mathbb{R}\}$, for $h \neq 0$ is given by $\frac{\mathbb{E}_1[rh]}{\mathbb{E}_1[h^2]}h$, we may explicitly minimize the objective in Equation (LR-QR) over $\beta$ via

$$\ell_\alpha(h, S) + \lambda \min_{\beta \in \mathbb{R}} \mathbb{E}_1[(\beta h - r)^2] = \ell_\alpha(h, S) + \lambda \mathbb{E}_1\left[\left(\frac{\mathbb{E}_1[rh]}{\mathbb{E}_1[h^2]}h - r\right)^2\right]$$

$$= \ell_\alpha(h, S) + \lambda\left(\mathbb{E}_1[r^2] - \mathbb{E}_1\left[\left(\frac{\mathbb{E}_1[rh]}{\mathbb{E}_1[h^2]}h\right)^2\right]\right) = \ell_\alpha(h, S) + \lambda\left(\mathbb{E}_1[r^2] - \frac{\mathbb{E}_1[rh]^2}{\mathbb{E}_1[h^2]}\right),$$

where in the second step we applied the Pythagorean theorem. Since the term $\lambda \mathbb{E}_1[r^2]$ does not depend on the optimization variable $h$, we may drop it from the objective, which yields the objective in Equation (18). It follows that $h$ is a minimizer of the objective in Equation (18) iff $h = h^*_\lambda$ for some minimizer $(h^*_\lambda, \beta^*_\lambda)$ of the objective of Equation (LR-QR). $\square$

**Lemma L.2.** *Let $r_\mathcal{H}$ denote the projection of $r$ onto $\mathcal{H}$ in the Hilbert space induced by the inner product $\langle f, g \rangle = \mathbb{E}_1[fg]$. Then under Condition 5 and Condition 9, there exists $\theta^* > 0$ such that $\mathbb{E}_1[S] - \alpha^{-1}\mathbb{E}_1[\ell_\alpha(\theta^* r_\mathcal{H}, S)] > 0$.*

*Proof.* Define $g : \mathbb{R} \to \mathbb{R}$ by $g(\theta) = \mathbb{E}_1[S] - \alpha^{-1}\mathbb{E}_1[\ell_\alpha(\theta^* r_\mathcal{H}, S)]$. Clearly $g(0) = 0$. Note that by Condition 5, $\mathbb{P}_{S|X}[S = 0] = 0$, so that

$$g'(0) = -\alpha^{-1}\mathbb{E}_1[r_\mathcal{H}(\mathbb{P}_{S|X}[S \leqslant 0] - (1 - \alpha))] = \alpha^{-1}(1 - \alpha)\mathbb{E}_1[r_\mathcal{H}].$$

By Condition 9, $\mathbb{E}_1[r_\mathcal{H}] = \mathbb{E}_1[r_\mathcal{H} \cdot 1] = \mathbb{E}_1[r \cdot 1] = \mathbb{E}_1[r] = 1$, so $g'(0) > 0$. Thus there exists $\theta^* > 0$ such that $g(\theta^*) > g(0) = 0$, as claimed. $\square$

**Lemma L.3** (Existence of unconstrained minimizers). *Under Condition 2, Condition 5, Condition 6, Condition 7, Condition 8, Condition 9, and Condition 10, for each $\lambda \geqslant 0$, there exists a global minimizer $(h^*_\lambda, \beta^*_\lambda)$ of the objective in Equation* (LR-QR).

*Proof.* Fix $\lambda \geqslant 0$. By Condition 10 and Lemma L.1, it suffices to show that there exists a global minimizer of the objective in Equation (18). Let $G(h)$ denote the objective of Equation (18). Define the function $\tilde{h} = \theta^* r_\mathcal{H} \in \mathcal{H} \setminus \{0\}$, where $\theta^*$ is chosen to satisfy Lemma L.2. With $c_{\text{indep}}$ from Condition 6, define $\tilde{B}(\lambda) := 2c_{\text{indep}}^{-1}(1 + \alpha^{-1}\mathbb{E}_1[\ell_\alpha(\tilde{h}, S)]) > 0$ and

$$\tilde{b}(\lambda) := \frac{1}{2}\lambda_{\max}(\Sigma)^{-1/2}(\mathbb{E}_1[S] - \alpha^{-1}\mathbb{E}_1[\ell_\alpha(\tilde{h}, S)]) > 0.$$

We show that if $\|h\| \geqslant \tilde{B}(\lambda)$ or $\|h\| \leqslant \tilde{b}(\lambda)$, then $G(h) > G(\tilde{h})$. Consequently, the minimization in Equation (18) can be taken over the compact set $\{\langle \gamma, \Phi \rangle : \tilde{b}(\lambda) \leqslant \|\gamma\|_2 \leqslant \tilde{B}(\lambda)\} \subseteq \mathcal{H}$, so that by continuity of $G$ on $\mathcal{H} \setminus \{0\}$, a global minimizer $h^*_\lambda$ exists.

To see this, first suppose $\|h\| \geqslant \tilde{B}(\lambda)$. Then writing $h = \langle \gamma, \Phi \rangle$ for $\gamma \in \mathbb{R}^d$ and applying Lemma N.1, the triangle inequality, and $S \in [0, 1]$,

$$\mathbb{E}_1[\ell_\alpha(h, S)] \geqslant \alpha \mathbb{E}_1[|h - S|] \geqslant \alpha(\mathbb{E}_1[|h|] - \mathbb{E}_1[|S|]) \geqslant \alpha(\mathbb{E}_1[|\langle \gamma, \Phi \rangle|] - 1). \tag{19}$$

By Condition 6 and our assumption that $\|h\| \geqslant \tilde{B}(\lambda)$, this implies that $\mathbb{E}_1[\ell_\alpha(h, S)] \geqslant \alpha(\tilde{B}(\lambda)c_{\text{indep}} - 1)$. Further, by the Cauchy-Schwarz inequality,

$$\frac{\mathbb{E}_1[rh]^2}{\mathbb{E}_1[h^2]} \leqslant \sup_{\tilde{h}' \in \mathcal{H} \setminus \{0\}} \frac{\mathbb{E}_1[r\tilde{h}']^2}{\mathbb{E}_1[(\tilde{h}')^2]} \leqslant \mathbb{E}_1[r_\mathcal{H}^2].$$

Thus by Lemma N.1 and Condition 8, $G(h) \geqslant \alpha(\tilde{B}(\lambda)c_{\text{indep}} - 1) - \lambda\mathbb{E}_1[r_{\mathcal{H}}^2]$. To prove the inequality $G(h) > G(\tilde{h})$, it suffices to show that

$$\alpha(\tilde{B}(\lambda)c_{\text{indep}} - 1) - \lambda\mathbb{E}_1[r_{\mathcal{H}}^2] > \mathbb{E}_1[\ell_\alpha(\tilde{h}, S)] - \lambda\frac{\mathbb{E}_1[r\tilde{h}]^2}{\mathbb{E}_1[\tilde{h}^2]}.$$

Indeed, since $\tilde{h}$ is a scalar multiple of $r_{\mathcal{H}}$, we have $\mathbb{E}_1[r_{\mathcal{H}}^2] = \frac{\mathbb{E}_1[r\tilde{h}]^2}{\mathbb{E}_1[\tilde{h}^2]}$, so the inequality reduces to $\alpha(\tilde{B}(\lambda)c_{\text{indep}} - 1) > \mathbb{E}_1[\ell_\alpha(\tilde{h}, S)]$. This holds by our choice of $\tilde{B}(\lambda)$, which finishes the argument in this case.

Next, suppose $\|h\| \leqslant \tilde{b}(\lambda)$. By Lemma N.1, the triangle inequality, and $S \in [0, 1]$,

$$\mathbb{E}_1[\ell_\alpha(h, S)] \geqslant \alpha\mathbb{E}_1[|h - S|] \geqslant \alpha(\mathbb{E}_1[S] - \mathbb{E}_1[|h|]). \tag{20}$$

As above, the Cauchy-Schwarz inequality implies the bound $\frac{\mathbb{E}_1[rh]^2}{\mathbb{E}_1[h^2]} \leqslant \mathbb{E}_1[r_{\mathcal{H}}^2]$. We deduce that

$$G(h) \geqslant \alpha(\mathbb{E}_1[S] - \mathbb{E}_1[|h|]) - \lambda\mathbb{E}_1[r_{\mathcal{H}}^2].$$

Writing $h = \langle \gamma, \Phi \rangle$ for $\gamma \in \mathbb{R}^d$, our assumption that $\|h\| \leqslant \tilde{b}(\lambda)$ implies that

$$\mathbb{E}_1[|h|] \leqslant \mathbb{E}_1[|h|^2]^{1/2} = \mathbb{E}_1[\gamma^\top \Phi\Phi^\top \gamma]^{1/2} \leqslant \tilde{b}(\lambda)\lambda_{\max}(\Sigma)^{1/2},$$

which when plugged into our lower bound on $G(h)$ yields

$$G(h) \geqslant \alpha(\mathbb{E}_1[S] - \tilde{b}(\lambda)\lambda_{\max}(\Sigma)^{1/2}) - \lambda\mathbb{E}_1[r_{\mathcal{H}}^2].$$

To prove the inequality $G(h) > G(\tilde{h})$, it suffices to show that

$$\alpha(\mathbb{E}_1[S] - \tilde{b}(\lambda)\lambda_{\max}(\Sigma)^{1/2}) - \lambda\mathbb{E}_1[r_{\mathcal{H}}^2] > \mathbb{E}_1[\ell_\alpha(\tilde{h}, S)] - \lambda\frac{\mathbb{E}_1[r\tilde{h}]^2}{\mathbb{E}_1[\tilde{h}^2]}.$$

As above, since $\tilde{h}$ is a scalar multiple of $r_{\mathcal{H}}$, we have $\mathbb{E}_1[r_{\mathcal{H}}^2] = \frac{\mathbb{E}_1[r\tilde{h}]^2}{\mathbb{E}_1[\tilde{h}^2]}$, so the inequality reduces to

$$\alpha(\mathbb{E}_1[S] - \tilde{b}(\lambda)\lambda_{\max}(\Sigma)^{1/2}) > \mathbb{E}_1[\ell_\alpha(\tilde{h}, S)].$$

This holds for our choice of $\tilde{b}(\lambda)$, finishing the proof. $\qquad\square$

**Lemma L.4** (Bounds on unconstrained minimizers). *Under the conditions used in Lemma L.3, for all $\lambda > 0$, for any minimizer $(h_\lambda^*, \beta_\lambda^*)$ of the objective in Equation* (LR-QR)*, we have that $\|h_\lambda^*\| \in (B_{\text{lower}}, B_{\text{upper}})$ and $\beta_\lambda^* \in (\beta_{\text{lower}}, \beta_{\text{upper}})$, where*

$$B_{\text{lower}} = \frac{1}{2}\lambda_{\max}(\Sigma)^{-1/2}(\mathbb{E}_1[S] - \alpha^{-1}\mathbb{E}_1[\ell_\alpha(\theta^* r_{\mathcal{H}}, S)]) > 0, \tag{21}$$

$$B_{\text{upper}} = 2c_{\text{indep}}^{-1}(\alpha^{-1}\mathbb{E}_1[\ell_\alpha(\theta^* r_{\mathcal{H}}, S)] + 1), \qquad \beta_{\text{lower}} = \frac{c_{\text{align}}}{B_{\text{upper}}\lambda_{\max}(\Sigma)^{1/2}} > 0,$$

$$\beta_{\text{upper}} = \frac{\mathbb{E}_1[r^2]^{1/2}}{B_{\text{lower}}\lambda_{\min}(\Sigma)^{1/2}},$$

*and where $\theta^* > 0$ is as in Lemma L.2 and $r_{\mathcal{H}}$ denotes the projection of $r$ onto $\mathcal{H}$ in the Hilbert space induced by the inner product $\langle f, g \rangle = \mathbb{E}_1[fg]$.*

*Proof.* In order to derive our bounds, we consider the reparametrized optimization problem

$$\min_{h \in \mathcal{H} \setminus \{0\}} \xi\mathbb{E}_1[\ell_\alpha(h, S)] - \frac{\mathbb{E}_1[rh]^2}{\mathbb{E}_1[h^2]} \tag{22}$$

for $\xi \geqslant 0$. We claim that for $\xi > 0$, any minimizer of the objective in Equation (22) is of the form $h_{1/\xi}^*$. To see this, note that for $\xi > 0$, the objective of Equation (18) with regularization $\lambda = 1/\xi$ can be obtained by scaling the objective of Equation (22) by the positive factor $1/\xi$. Next, by Condition 10, we may apply Lemma L.1 to deduce that $h \in \mathcal{H} \setminus \{0\}$ is a minimizer of the objective in Equation (18) with regularization $\lambda = 1/\xi$ iff $h = h_{1/\xi}^*$.

In particular, by Lemma L.3, for all $\xi > 0$, there exists a global minimizer of Equation (22) with regularization $\xi$. In the case that $\xi = 0$, it is clear that any minimizer $h_\infty^*$ of the objective in Equation (22) with regularization $\xi = 0$ has the form $h_\infty^* = \theta r_{\mathcal{H}}$ for some scalar $\theta > 0$.

Since there exists a minimizer of the objective in Equation (22) for all regularizations $\xi$ in the interval $[0, \infty)$, we may apply Lemma M.1 to deduce that for all $\xi > 0$ we have $\mathbb{E}_1[\ell_\alpha(h_{1/\xi}^*, S)] \leqslant \mathbb{E}_1[\ell_\alpha(h_\infty^*, S)]$.

We prove lower and upper bounds on $\|h_{1/\xi}^*\|$ for all $\xi > 0$. We begin with the lower bound.

*Lower bound:* By (20), we have $\mathbb{E}_1[\ell_\alpha(h_{1/\xi}^*, S)] \geqslant \alpha(\mathbb{E}_1[S] - \mathbb{E}_1[|h_{1/\xi}^*|])$. Rearranging, we obtain the lower bound

$$\mathbb{E}_1[|h_{1/\xi}^*|] \geqslant \mathbb{E}_1[S] - \alpha^{-1}\mathbb{E}_1[\ell_\alpha(h_\infty^*, S)].$$

By Lemma L.2, there exists $\theta^* > 0$ such that $\mathbb{E}_1[S] - \alpha^{-1}\mathbb{E}_1[\ell_\alpha(\theta^* r_{\mathcal{H}}, S)] > 0$. Setting $h_\infty^* = \theta^* r_{\mathcal{H}}$ and plugging in the expression for $B_{\text{lower}}$ given in (21), our lower bound becomes $\mathbb{E}_1[|h_{1/\xi}^*|] > \lambda_{\max}(\Sigma)^{1/2}B_{\text{lower}}$. We now convert this $L^1$ norm bound to an $L^2$ norm bound as follows. Write $h_{1/\xi}^* = \langle \gamma_{1/\xi}^*, \Phi \rangle$ for $\gamma_{1/\xi}^* \in \mathbb{R}^d$. By the Cauchy-Schwarz inequality, we obtain the upper bound

$$\mathbb{E}_1[|h_{1/\xi}^*|] \leqslant \mathbb{E}_1[|h_{1/\xi}^*|^2]^{1/2} = \mathbb{E}_1[(\gamma_{1/\xi}^*)^\top \Phi\Phi^\top \gamma_{1/\xi}^*]^{1/2} \leqslant \lambda_{\max}(\Sigma)^{1/2}\|\gamma_{1/\xi}^*\|_2.$$

Combining this with the lower bound $\mathbb{E}_1[|h_{1/\xi}^*|] > \lambda_{\max}(\Sigma)^{1/2}B_{\text{lower}}$, we deduce that $\|h_{1/\xi}^*\| = \|\gamma_{1/\xi}^*\|_2 > B_{\text{lower}}$, as claimed.

*Upper bound:* We prove the upper bound in a similar manner. By the first two steps in (19), and using $S \in [0, 1]$, we have

$$\mathbb{E}_1[\ell_\alpha(h_{1/\xi}^*, S)] \geqslant \alpha(\mathbb{E}_1[|h_{1/\xi}^*|] - \mathbb{E}_1[|S|]) \geqslant \alpha(\mathbb{E}_1[|h_{1/\xi}^*|] - 1).$$

Rearranging, we obtain the upper bound $\mathbb{E}_1[|h_{1/\xi}^*|] \leqslant \alpha^{-1}\mathbb{E}_1[\ell_\alpha(h_\infty^*, S)] + 1$. Write $h_{1/\xi}^* = \langle \gamma_{1/\xi}^*, \Phi \rangle$ for $\gamma_{1/\xi}^* \in \mathbb{R}^d$. Since we have already established that $\|h_{1/\xi}^*\| > B_{\text{lower}} > 0$, we know that $\gamma_{1/\xi}^* \neq 0$. Thus we may write

$$\mathbb{E}_1[|h_{1/\xi}^*|] = \mathbb{E}_1[|\langle \gamma_{1/\xi}^*, \Phi \rangle|] = \|\gamma_{1/\xi}^*\|_2 \mathbb{E}_1\left[\left|\left\langle \frac{\gamma_{1/\xi}^*}{\|\gamma_{1/\xi}^*\|_2}, \Phi \right\rangle\right|\right].$$

By Condition 6, this is at least $\|\gamma_{1/\xi}^*\|_2 c_{\text{indep}}$. Combining these upper and lower bounds on $\mathbb{E}_1[|h_{1/\xi}^*|]$, we obtain $\|\gamma_{1/\xi}^*\|_2 c_{\text{indep}} \leqslant \alpha^{-1}\mathbb{E}_1[\ell_\alpha(h_\infty^*, S)] + 1$. Isolating $\|\gamma_{1/\xi}^*\|_2$, we have

$$\|h_{1/\xi}^*\| = \|\gamma_{1/\xi}^*\|_2 \leqslant c_{\text{indep}}^{-1}(\alpha^{-1}\mathbb{E}_1[\ell_\alpha(h_\infty^*, S)] + 1) < B_{\text{upper}},$$

as claimed.

Having established $0 < B_{\text{lower}} < \inf_{\lambda > 0}\|h_\lambda^*\| \leqslant \sup_{\lambda > 0}\|h_\lambda^*\| < B_{\text{upper}} < \infty$, we turn to upper and lower bounds on $\beta_\lambda^*$. As shown in the proof of Lemma L.1, if $(h_\lambda^*, \beta_\lambda^*)$ is a minimizer of the objective in Equation (LR-QR) with regularization $\lambda$, then $\beta_\lambda^* = \frac{\mathbb{E}_1[rh_\lambda^*]}{\mathbb{E}_1[|h_\lambda^*|^2]}$. By Condition 7, $\frac{\mathbb{E}_1[rh_0^*]}{\mathbb{E}_1[|h_0^*|^2]^{1/2}} \geqslant c_{\text{align}} > 0$ for some minimizer $(h_0^*, \beta_0^*)$ of the objective in Equation (LR-QR) with regularization 0. By Condition 10 and Lemma L.1, $h$ is a minimizer of the objective in Equation (18) with regularization $\lambda \geqslant 0$ iff $h = h_\lambda^*$ for some minimizer $(h_\lambda^*, \beta_\lambda^*)$ of the objective in Equation (LR-QR). Thus by Lemma L.3, for all $\lambda \geqslant 0$, there exists a global minimizer of Equation (18), and we may apply Lemma M.1 to Equation (18) to deduce that for any $\lambda \geqslant 0$ we have $\frac{\mathbb{E}_1[rh_\lambda^*]}{\mathbb{E}_1[|h_\lambda^*|^2]^{1/2}} \geqslant c_{\text{align}} > 0$. Consequently, by our bounds on $h_\lambda^*$, Condition 8, and the Cauchy-Schwarz inequality, if we write $h_\lambda^* = \langle \gamma_\lambda^*, \Phi \rangle$ for $\gamma_\lambda^* \in \mathbb{R}^d$, then we have

$$\beta_\lambda^* \geqslant \frac{c_{\text{align}}}{\mathbb{E}_1[|h_\lambda^*|^2]^{1/2}} = \frac{c_{\text{align}}}{\mathbb{E}_1[(\gamma_\lambda^*)^\top \Phi\Phi^\top \gamma_\lambda^*]^{1/2}} > \frac{c_{\text{align}}}{B_{\text{upper}}\lambda_{\max}(\Sigma)^{1/2}} =: \beta_{\text{lower}}$$

and

$$\beta_\lambda^* \leqslant \frac{\mathbb{E}_1[r^2]^{1/2}}{\mathbb{E}_1[|h_\lambda^*|^2]^{1/2}} < \frac{\mathbb{E}_1[r^2]^{1/2}}{B_{\text{lower}}\lambda_{\min}(\Sigma)^{1/2}} =: \beta_{\text{upper}},$$

completing the proof. $\qquad\square$

# M  Monotonicity

**Lemma M.1.** *For some set $\mathcal{X}$ and $f, g : \mathcal{X} \to \mathbb{R}$, let $x(c) = \arg\min_{x \in \mathcal{X}}(f(x) + cg(x))$, where $f, g$ are such that for some interval $\mathcal{I} \subset \mathbb{R}$, the minimum is attained for all $c \in \mathcal{I}$. Then $G : \mathcal{I} \to \mathbb{R}$, $G : c \mapsto g(x(c))$ is non-increasing in c.*

*Proof.* Let $c_1, c_2 \in \mathcal{I}$, $c_1 < c_2$. At $c = c_1$, the minimizer $x(c_1)$ satisfies:

$$f(x(c_1)) + c_1 g(x(c_1)) \leq f(x(c_2)) + c_1 g(x(c_2)).$$

At $c = c_2$, the minimizer $x(c_2)$ satisfies:

$$f(x(c_2)) + c_2 g(x(c_2)) \leq f(x(c_1)) + c_2 g(x(c_1)).$$

Adding the two inequalities, we find

$$\begin{aligned}
&\big[f(x(c_1)) + c_1 g(x(c_1))\big] + \big[f(x(c_2)) + c_2 g(x(c_2))\big] \\
&\leq \big[f(x(c_1)) + c_2 g(x(c_1))\big] + \big[f(x(c_2)) + c_1 g(x(c_2))\big].
\end{aligned}$$

Subtracting the common terms $f(x(c_1)) + f(x(c_2))$ leads to

$$c_1 g(x(c_1)) + c_2 g(x(c_2)) \leq c_2 g(x(c_1)) + c_1 g(x(c_2)).$$

Rearranging, and factoring out $c_1$ and $c_2$, we find

$$c_1\big[g(x(c_1)) - g(x(c_2))\big] - c_2\big[g(x(c_1)) - g(x(c_2))\big] \leq 0.$$

Thus, $(c_1 - c_2)\big[g(x(c_1)) - g(x(c_2))\big] \leq 0$. Since $c_2 - c_1 > 0$, the inequality implies $g(x(c_1)) \geq g(x(c_2))$, as desired. $\qquad\square$

# N  Helper lemmas

**Lemma N.1.** *If $\alpha \leqslant 0.5$, then $\alpha|c - s| \leqslant \ell_\alpha(c, s) \leqslant (1 - \alpha)|c - s|$ for all $c, s \in \mathbb{R}$.*

*Proof.* If $s \geqslant c$, then $\ell_\alpha(c, s) = (1 - \alpha)(s - c)$. Since $s - c \geqslant 0$ and $\alpha \leqslant 1 - \alpha$, we have $\alpha(s - c) \leqslant \ell_\alpha(c, s) \leqslant (1 - \alpha)(s - c)$, which implies $\alpha|c - s| \leqslant \ell_\alpha(c, s) \leqslant (1 - \alpha)|c - s|$. If $s < c$, then $\ell_\alpha(c, s) = \alpha(c - s)$. Since $c - s > 0$ and $\alpha \leqslant 1 - \alpha$, we have $\alpha(c - s) \leqslant \ell_\alpha(c, s) \leqslant (1 - \alpha)(c - s)$, which implies $\alpha|c - s| \leqslant \ell_\alpha(c, s) \leqslant (1 - \alpha)|c - s|$. $\qquad\square$

**Lemma N.2.** *If $\alpha \leqslant 0.5$, then the map $\mathbb{R} \to \mathbb{R}$ given by $c \mapsto \ell_\alpha(c, s)$ is $(1 - \alpha)$-Lipschitz.*

*Proof.* If $s \leqslant c_1 \leqslant c_2$, we have $0 \leqslant \ell_\alpha(c_2, s) - \ell_\alpha(c_1, s) = \alpha(c_2 - c_1)$, which by $\alpha \leqslant 0.5$ is at most $(1 - \alpha)(c_2 - c_1)$. Hence $|\ell_\alpha(c_2, s) - \ell_\alpha(c_1, s)| \leqslant (1 - \alpha)|c_2 - c_1|$. If $c_1 \leqslant s \leqslant c_2$ and $\ell_\alpha(c_2, s) \geqslant \ell_\alpha(c_1, s)$, then we have

$$0 \leqslant \ell_\alpha(c_2, s) - \ell_\alpha(c_1, s) = \alpha(c_2 - s) - (1 - \alpha)(s - c_1) \leqslant \alpha(c_2 - s) + \alpha(s - c_1) = \alpha(c_2 - c_1),$$

which by $\alpha \leqslant 0.5$ implies $|\ell_\alpha(c_2, s) - \ell_\alpha(c_1, s)| \leqslant (1 - \alpha)|c_2 - c_1|$. If $c_1 \leqslant s \leqslant c_2$ and $\ell_\alpha(c_2, s) \leqslant \ell_\alpha(c_1, s)$, then

$$\begin{aligned}
0 &\leqslant \ell_\alpha(c_1, s) - \ell_\alpha(c_2, s) = (1 - \alpha)(s - c_1) - \alpha(c_2 - s) \\
&\leqslant (1 - \alpha)(s - c_1) + (1 - \alpha)(c_2 - s) = (1 - \alpha)(c_2 - c_1),
\end{aligned}$$

hence $|\ell_\alpha(c_2, s) - \ell_\alpha(c_1, s)| \leqslant (1 - \alpha)|c_2 - c_1|$. Finally, if $c_1 \leqslant c_2 \leqslant s$, we have $0 \leqslant \ell_\alpha(c_1, s) - \ell_\alpha(c_2, s) = (1 - \alpha)(c_2 - c_1)$, hence $|\ell_\alpha(c_2, s) - \ell_\alpha(c_1, s)| \leqslant (1 - \alpha)|c_2 - c_1|$. $\qquad\square$

**Lemma N.3.** *The map $\mathcal{H} \to \mathbb{R}$ given by $h \mapsto \ell_\alpha(h(x), s)$ is convex for all $x \in \mathcal{X}$ and $s \in \mathbb{R}$.*

*Proof.* Write $h(x) = \langle \gamma, \Phi \rangle$ for $\gamma \in \mathbb{R}^d$. It suffices to show that the mapping $\mathbb{R}^d \to \mathbb{R}$ given by $\gamma \mapsto \ell_\alpha(\gamma^\top \Phi(x), s)$ is convex. But this map is the composition of the linear function $\mathbb{R}^d \to \mathbb{R}$ given by $\gamma \mapsto \gamma^\top \Phi(x)$ and the convex function $\mathbb{R} \to \mathbb{R}$ given by $c \mapsto \ell_\alpha(c, s)$, hence it is convex. $\qquad\square$

**Lemma N.4.** *Under Condition 1, if $h \in \mathcal{H}$, then $\sup_{x \in \mathcal{X}} |h(x)| \leqslant C_\Phi \|h\|$, where we use the norm given by $\|h\| = \|\gamma\|_2$ for $h = \langle \gamma, \Phi \rangle$. In particular, if $h \in \mathcal{H}_B$, then $\sup_{x \in \mathcal{X}} |h(x)| \leqslant BC_\Phi$.*

*Proof.* Writing $h = \langle \gamma, \Phi \rangle$ for $\gamma \in \mathbb{R}^d$, we have $\sup_{x \in \mathcal{X}} |h(x)| = \sup_{x \in \mathcal{X}} |\langle \gamma, \Phi(x) \rangle| \leqslant \sup_{x \in \mathcal{X}} \|\gamma\|_2 \|\Phi(x)\|_2 \leqslant C_\Phi \|h\|$, where in the second step we applied the Cauchy-Schwarz inequality. $\qquad\square$

**Lemma N.5.** *Consider the function $\varphi : \mathbb{R}^d \to \mathbb{R}$ given by $\varphi(\gamma) = \mathbb{E}_1[\ell_\alpha(h_\gamma(X), S)]$, where $h := h_\gamma : \mathcal{X} \to \mathbb{R}$ is given by $h(x) = \langle \gamma, \Phi(x) \rangle$ for all $x \in \mathcal{X}$. Then under Condition 1 and Condition 5, $\varphi$ is twice-differentiable, with gradient and Hessian given by*

$$\nabla_\gamma \varphi(\gamma) = \mathbb{E}_1[(\mathbb{P}_{S|X}[h(X) > S] - (1 - \alpha))\Phi(X)], \qquad \nabla_\gamma^2 \varphi(\gamma) = \mathbb{E}_1[f_{S|X}(h(X))\Phi(X)\Phi(X)^\top].$$

*Consequently, given $\tilde{\gamma} \in \mathbb{R}^d$, defining $g : \mathcal{X} \to \mathbb{R}$ as $g(x) = \langle \tilde{\gamma}, \Phi(x) \rangle$ for all $x \in \mathcal{X}$, the directional derivative of $\varphi : \mathcal{H} \to \mathbb{R}$ in the direction $g$ is given by $\langle \tilde{\gamma}, \nabla_\gamma \varphi(\gamma) \rangle = \mathbb{E}_1[(\mathbb{P}_{S|X}[h(X) > S] - (1 - \alpha))g(X)]$.*

*Proof.* For each $x \in \mathcal{X}$, define the function $\eta(\cdot; x) : \mathbb{R} \to \mathbb{R}$ given, for all $u$, by $\eta(u; s) = \mathbb{E}_{S|X=x}[\ell_\alpha(u, S)]$. For each $s \in \mathbb{R}$, define the function $\chi(\cdot; s) : \mathbb{R} \to \mathbb{R}$, where for all $u$, $\chi(u; s) = \alpha \mathbf{1}[u > s] - (1 - \alpha)\mathbf{1}[u \leqslant s]$.

By the definition of the pinball loss $\ell_\alpha(\cdot, \cdot)$, and since by Condition 5 the conditional density $f_{S|X=x}(\cdot)$ of $S|X = x$ exists for all $x \in \mathcal{X}$, the derivative of $\ell_\alpha(u, S)$ with respect to $u$ agrees with the random variable $\chi(u; S)$ almost surely with respect to the distribution $S|X = x$. Also, note that for fixed $u \in \mathbb{R}$, $|\chi(u; S)|$ is bounded by the constant $(1 - \alpha)$. By the dominated convergence theorem, it follows that $u \mapsto \eta(u; x)$ is differentiable, and that its derivative equals $\frac{\partial}{\partial u} \eta(u; x) = \mathbb{E}_{S|X=x}[\chi(u; S)]$, which, by the formula for $\chi(u; S)$, can be written as $\alpha \mathbb{P}_{S|X=x}[u > S] - (1 - \alpha)\mathbb{P}_{S|X=x}[u \leqslant S]$. Thus for all $u \in \mathbb{R}$ and $x \in \mathcal{X}$, we may write $\frac{\partial}{\partial u} \eta(u; x) = \mathbb{P}_{S|X=x}[u > S] - (1 - \alpha)$. Since by Condition 5 the conditional density $f_{S|X=x}$ of the distribution $S|X = x$ exists for all $x \in \mathcal{X}$, it follows that the cdf $u \mapsto \mathbb{P}_{S|X=x}[u > S]$ is differentiable for all $u \in \mathbb{R}$ and all $x \in \mathcal{X}$ with derivative given by $u \mapsto f_{S|X=x}(u)$. Thus the map $u \mapsto \frac{\partial}{\partial u} \eta(u; x)$ is differentiable for all $x \in \mathcal{X}$ with derivative given by $u \mapsto f_{S|X=x}(u)$. In particular, $\eta(\cdot; x)$ is twice-differentiable with second derivative given by $f_{S|X=x}(\cdot)$.

Next, for each $x \in \mathcal{X}$, define the function $\psi(\cdot; x) : \mathbb{R}^d \to \mathbb{R}$ given by $\psi(\gamma; x) = \mathbb{E}_{S|X=x}[\ell_\alpha(h_\gamma(x), S)]$, where $h = h_\gamma = \langle \gamma, \Phi \rangle$. For each $x \in \mathcal{X}$, let $\mathrm{ev}(\cdot; x) : \mathbb{R}^d \mapsto \mathbb{R}$ be given by $\mathrm{ev}(\gamma; x) = h_\gamma(x)$, where $h = h_\gamma = \langle \gamma, \Phi \rangle$. Then $\psi(\cdot; x)$ is given by the composition $\eta(\cdot; x) \circ \mathrm{ev}(\cdot; x)$. Since $\mathrm{ev}(\gamma; x) = \langle \gamma, \Phi(x) \rangle$, $\mathrm{ev}(\cdot; x)$ is linear, it is smooth. Its gradient is given by $\nabla_\gamma \mathrm{ev}(\gamma; x) = \Phi(x)$ for all $\gamma \in \mathbb{R}^d$, and its Hessian is zero. It follows that $\psi(\cdot; x)$ is twice-differentiable. By the chain rule, the gradient of $\psi(\cdot; x)$ is given by

$$\nabla_\gamma \psi(\gamma; x) = \left. \frac{\partial}{\partial u} \eta(u; x) \right|_{u=\mathrm{ev}(\gamma; x)} \cdot \nabla_\gamma \mathrm{ev}(\gamma; x) = (\mathbb{P}_{S|X=x}[h(x) > S] - (1 - \alpha))\Phi(x).$$

Since the map $\gamma \mapsto \mathbb{P}_{S|X=x}[h(x) > S] - (1 - \alpha)$ is given by the composition $\frac{\partial}{\partial u} \eta(\cdot; x) \circ \mathrm{ev}(\cdot; x)$, we may again apply the chain rule to deduce that the Hessian of $\psi(\cdot; x)$ is given by

$$\nabla_\gamma^2 \psi(\gamma; x) = \left. \frac{\partial^2}{\partial u^2} \eta(u; x) \right|_{u=\mathrm{ev}(\gamma; x)} \cdot \nabla_\gamma \mathrm{ev}(\gamma; x) \cdot \Phi(x)^\top = f_{S|X=x}(h(x))\Phi(x)\Phi(x)^\top.$$

Returning to our original function $\varphi$, note that by the tower property, $\varphi(\gamma) = \mathbb{E}_1[\psi(\gamma; X)]$. Note that $\|\nabla_\gamma \psi(\gamma; x)\|_2$ is at most

$$|\mathbb{P}_{S|X=x}[h(x) > S] - (1 - \alpha)| \|\Phi(x)\|_2 \leqslant (|\mathbb{P}_{S|X=x}[h(x) > S]| + (1 - \alpha))\|\Phi(x)\|_2 \leqslant (2 - \alpha)C_\Phi,$$

where in the first step we used the triangle inequality, and in the second step we used the fact that $\mathbb{P}_{S|X=x}[h(x) > S] \leqslant 1$ and Condition 1. Similarly, we may bound the Frobenius norm $\|\cdot\|_F$ of $\nabla_\gamma^2 \psi(\gamma; x)$ by

$$|f_{S|X=x}(h(x))| \|\Phi(x)\Phi(x)^\top\|_F \leqslant C_f \|\Phi(x)\|_2^2 \leqslant C_f C_\Phi^2,$$

where in the first step we used Condition 1, the identity $\|vv^\top\|_F = \|v\|_2^2$, and in the second step we used Condition 5. Since the entries of $\nabla_\gamma \psi(\cdot; x)$ and $\nabla_\gamma^2 \psi(\cdot; x)$ are bounded by constants, we may apply the dominated convergence theorem to deduce that $\varphi$ is twice-differentiable, with gradient given by $\nabla_\gamma \varphi(\gamma) = \mathbb{E}_1[\nabla_\gamma \psi(\gamma; X)]$ and Hessian given by $\nabla_\gamma^2 \varphi(\gamma) = \mathbb{E}_1[\nabla_\gamma^2 \psi(\gamma; X)]$.

Finally, since the directional derivative of $\varphi$ in the direction $g$ is defined as $\langle \tilde{\gamma}, \nabla_\gamma \varphi(\gamma) \rangle$, we may plug in our expression for the gradient to deduce

$$\begin{aligned}
\langle \tilde{\gamma}, \nabla_\gamma \varphi(\gamma) \rangle &= \langle \tilde{\gamma}, \mathbb{E}_1[(\mathbb{P}_{S|X}[h(X) > S] - (1 - \alpha))\Phi(X)] \rangle \\
&= \mathbb{E}_1[(\mathbb{P}_{S|X}[h(X) > S] - (1 - \alpha))\langle \tilde{\gamma}, \Phi(X) \rangle] \\
&= \mathbb{E}_1[(\mathbb{P}_{S|X}[h(X) > S] - (1 - \alpha))g(X)].
\end{aligned}$$

The result follows. $\qquad\square$

## O   Unbounded scores

In this section, we comment on our assumption that $S \in [0, 1]$ a.s. Given an arbitrary a.s. finite score $S$, and given the sigmoid function $g : \mathbb{R} \to \mathbb{R}$ given by $g(x) = \frac{1}{1 + e^{-x}}$, the composition $g(S)$ is a.s. $[0, 1]$-valued. As the proof of Theorem 4.2 only requires boundedness, this allows one to obtain the generalization bound for arbitrary score functions.

However, the proof of Theorem 4.3 places a boundedness assumption on the conditional density of $S|X = x$ in Condition 5, which precludes us from directly applying the transformation trick given above. We claim that Condition 5 can be replaced with the following alternate condition:

**Condition 11.** *(1) The conditional density $f_{S|X=x}$ exists for all $x \in \mathcal{X}$; (2) there exists a constant $C_f > 0$ and a real $k > 0$ such that for all $s \in (0, 1)$, we have*

$$f_{S|X=x}(s) \leqslant C_f(|s|^{-k} + |1 - s|^{-k}),$$

*uniformly in $x \in \mathcal{X}$; (3) the basis $\Phi$ obeys*

$$\sup_{\gamma \in S^{d-1}} \mathbb{E}_1[\langle \gamma, \Phi(X) \rangle^{-k}] < \infty,$$

*and (4) the quantity $B_{\text{upper}}$ defined in Lemma L.4 obeys $B_{\text{upper}} < C_\Phi^{-1}$, where $C_\Phi$ is defined in Condition 1.*

In other words, we can allow the conditional density of $S|X = x$ to diverge at a polynomial rate near $s = 0$ and $s = 1$, so long as $\Phi$ obeys a certain moment condition, and so long as apriori, the population LR-QR objective can be restricted to a sufficiently small ball.

One can consider point (3) of Condition 11 as a slight strengthening of Condition 6, a quantitative independence condition on the basis functions. Regarding point (4) of Condition 11, note that by inspecting the definition of $B_{\text{upper}}$, we see that an upper bound on $B_{\text{upper}}$ imposes (a) a lower bound on $c_{\text{align}}$ in Condition 7, as well as (b) an upper bound on $\mathbb{E}_1[\ell_\alpha(\theta^\star r_H, S)]$, which states that optimally scaling the projection $r_{\mathcal{H}}$ can yield a threshold function with low pinball loss.

Now, we sketch how utilizing Condition 11 implies Theorem 4.3 for unbounded scores. In the original proof, Condition 5 is used in order to control expressions of the form $|\mathbb{E}_1[\Phi(X)\Phi(X)^T f_{S|X}(h(X))]|$, uniformly for $\gamma \in \mathbb{R}^d$ with $0 < B_{\text{lower}} \leqslant \|\gamma\|_2 \leqslant B_{\text{upper}}$, where $h(X) = \langle \gamma, \Phi(X) \rangle$. By Condition 1 and Jensen's inequality, this is bounded by $\mathbb{E}_1[|f_{S|X}(h(X))|]$, up to constants. By point (2) of Condition 11, this in turn is bounded by $\mathbb{E}_1[|h(X)|^{-k} + |1 - h(X)|^{-k}]$, up to constants. By point (3) of Condition 11 and the bounds $0 < B_{\text{lower}} \leqslant \|\gamma\|_2 \leqslant B_{\text{upper}}$, the first term $\mathbb{E}_1[|h(X)|^{-k}]$ is uniformly bounded. Next, by the triangle inequality, the Cauchy-Schwarz inequality, Condition 1, and point (4) of Condition 11, we have

$$|1 - h(X)| \geqslant 1 - |h(X)| \geqslant 1 - C_\Phi B_{\text{upper}} > 0,$$

so we may uniformly bound the second term by

$$\mathbb{E}_1[|1 - h(X)|^{-k}] \leqslant (1 - C_\Phi B_{\text{upper}})^{-k}.$$

Putting these bounds together, we see that Condition 11 provides the desired uniform control.

Finally, if we utilize Condition 11 instead of Condition 5, then the sigmoid transformation allows us to generalize Theorem 4.2 beyond bounded scores. Note that for $g$, the conditional density $f_{g(S)|X=x}$ of the transformed score $g(S)$ obeys

$$f_{g(S)|X=x}(t) = \frac{1}{t(1-t)} f_{S|X=x} \left( \log \frac{t}{1-t} \right)$$

for all $t \in (0,1)$. Consequently, if the original density $f_{S|X=x}(s)$ is supported on $\mathbb{R}$ with polynomially-decaying tails as $s \to \pm\infty$, then the transformed density diverges like $\sim (t(1 - t))^{-(1+o(1))}$ as $t \to 0, 1$, which satisfies Condition 11 with $k = 1 + \varepsilon$ for any $\varepsilon > 0$.

