# OpenReview forum: "Conformal Inference under High-Dimensional Covariate Shifts via Likelihood-Ratio Regularization"
_NeurIPS.cc/2025/Conference — NeurIPS 2025 poster_

### Official Review · Reviewer_8HeH · 2025-06-27

**Clarity:** 4
**Significance:** 4
**Originality:** 4
**Rating:** 6
**Confidence:** 3

**Summary:**

This paper tackles the problem of covariate shift in conformal prediction, where existing methods often rely on estimating a likelihood ratio function that may be hard to compute in high dimensions. The authors propose a new method, LRQR, which avoids estimating the likelihood ratio directly but still achieves valid marginal coverage. This is done under a mild assumption formalized through a hypothesis class $\mathcal{H}$. The method is supported by strong theoretical results and is benchmarked against several state-of-the-art approaches on real datasets.

**Questions:**

1. It seems that the method does not aim to achieve conditional conformal coverage. If I am wrong, I would appreciate clarification. Otherwise, could the authors explain why they perform conditional evaluations across subgroups (e.g., by race or category) in data analysis, if the method is only designed for marginal validity?

2. On line 104, the authors state that their theory requires scores in the $[0,1]$ range. But many common scores (e.g., residuals or negative log-likelihoods) are not bounded this way, as seen in the later experiments. Could the authors clarify the necessity of this restriction and how it aligns with practice?

3. I would appreciate a high-level explanation of why the linear hypothesis class is needed and how it is used in the proof.

**Ethical Concerns:**

["NO or VERY MINOR ethics concerns only"]

**Final Justification:**

I appreciate the author’s detailed explanation of both the linear class and the experimental results during the rebuttal. Overall, I believe this paper addresses an important and practically problem. The rebuttal further demonstrates the authors’ thoughtfulness and commitment. I will maintain my score of strong accept.

**Limitations:**

yes

**Quality:**

4

**Strengths And Weaknesses:**

Strengths
1. I find the paper clearly written and easy to follow.
2. The paper addresses a case where existing techniques may not apply due to data-dependent regularizers, and introduces a novel conditioning argument to overcome this.
3. The theoretical development is comprehensive and appears well-grounded. Although I did not verify all proofs in detail, the results are built upon solid theoretical foundations.
4. The proposed method is compared against strong baselines and evaluated on realistic benchmark datasets.

Weaknesses
1. I would appreciate a more formal and precise definition of the hypothesis class $\mathcal{H}$, as mentioned around lines 105–110. It is unclear what exactly constitutes a “linear hypothesis class” in the authors’ framework.
2. It is not clear how to verify whether a given dataset satisfies the assumptions required by $\mathcal{H}$. I would like to see a simulation that intentionally violates this assumption and examines how the method behaves in such cases.

---

> ### Author Rebuttal · Authors · 2025-07-27
>
> We thank the reviewer for the helpful comments.
>
> Weakness 1: "Clarify the definition of a linear class?"
>
> Response to Weakness 1:
> In our setting, we call a set of functions $H$
> from $\mathcal{X}\to \mathbb{R}$
> a "linear hypothesis class" if
> (1) $H$ forms a finite-dimensional vector space
> and
> (2) for each function $h\in H$, the second moment $E_1[|h(X)|^2]$ is finite.
>
> Weakness 2: "What if we violate the assumptions on $H$?"
>
> Response to Weakness 2:
> We thank the reviewer for the question about our regularity conditions
> on the hypothesis class $H$.
> Indeed, our theoretical guarantees on the performance of LR-QR
> rely on
> the true likelihood ratio $r$ being close in $L^2$-norm to
> the hypothesis class $H$.
> This is reflected in the misspecification error term
> $- (1-\alpha) E_1[ |r(X) - r_B(X)| ]$
> appearing in the lower bound in Theorem 4.3.
>
> Further, our guarantees rely on
> implicit assumptions
> on the dimensionality on $H$.
> To illustrate the dependence on $d$ arising from Condition 3 in Appendix E,
> suppose $\Phi$ forms an orthonormal basis for $P_1$.
> Then applying the empirical sample covariance tail bound from Chapter 5.6 of [5],
> we have $\lambda_{min}(\hat \Sigma) \ge 1 - O(d \log(dn_3)/n_3)$ with probability $1-o(n_3^{-1})$.
> This is of constant order if $d \log(d) \ll n_3$.
> So our bounds are meaningful if $d \ll n_3/\log(n_3)$ (which allows a growing dimension).
>
> Unfortunately, with the revised conference rules,
> we are unable to provide additional ablation plots
> to examine the effect of misspecification.
> However, we are happy to provide such plots in the revised version.
> For each of the three experiments,
> we can run LR-QR with hypothesis classes of varying complexities
> (e.g., linear combinations of the last-layer features
> of pretrained models of varying sizes)
> and compare the resulting coverage properties.
> We thank the reviewer for their interest.
>
> Q1: Clarification about the Communities and Crime experiment.
>
> Response to Q1:
> We thank the reviewer for allowing us to clarify the Communities and Crime experiment.
> Indeed, in our paper, our goal is to achieve marginal coverage at the nominal level in the test domain.
> Our description of the Communities and Crime experiment
> in Section 5.2 contained a typo, which we would like to correct here.
> We would like to clarify that we are not attempting
> to provide conditional coverage over each of the four groups.
> Rather, **we design four distinct covariate shifts**,
> run the LR-QR algorithm in each of the four scenarios,
> and evaluate the marginal coverage in the test domain.
> To elaborate,
> let us assign the index $i = 1$ to the Black subgroup,
> $i=2$ to the White subgroup,
> $i=3$ to the Asian subgroup,
> and $i=4$ to the Hispanic subgroup.
> Let $Z$ denote the set of datapoints
> that were not used for training the regression model.
> Then, for each $i\in [4]$,
> we design a covariate shift between calibration and test
> as follows:
> the $i$-th calibration set $Z_{cal,i}$ includes
> all individuals in $Z$
> that are not of race $i$,
> and the $i$-th test set
> includes
> all individuals in $Z$
> that are of race $i$.
> Thus, for each $i\in [4]$,
> we induce a covariate shift
> between $Z_{cal,i}$ and $Z_{test,i}$,
> shifting probability mass from
> "not race $i$"
> to "race $i$".
>
> In Figure 1, we evaluate each algorithm on each of the four experimental setups. Again, this is not conditional coverage, but rather marginal coverage in distinct experiments.
> Our results show that on this challenging 127-dimensional task,
> the LR-QR algorithm achieves excellent marginal coverage at test-time,
> for each of the four distinct choices of covariate shift.
>
> Q2: "Is $S\in [0,1]$ inconsistent with the experiments?"
>
> Response to Q2:
> The assumption that $S\in [0,1]$ is simply made for technical convenience.
> Our theoretical results easily generalize to the case of nonnegative and bounded scores. In the MMLU task, this assumption is satisfied, since the score is $1-f(x)_y \in [0,1]$.
> In the Communities and Crime regression task, our score is the residual $|y-\hat f(x)|$, where $y$ is the crime rate and $\hat f$ is a bounded predictor. Hence, this score is also bounded.
> On the RxRx1 classification task, the scores are nonnegative, but as noted by the reviewer they are not bounded apriori.
> Empirically, the LR-QR algorithm performs very well in these scenarios, so we conjecture that the boundedness assumption can be relaxed, and we view this as an interesting direction for further work. Moreover, in the final version of the paper, we will provide additional experiments with the log-score replaced by either the standard score $1-f(x)_y \in [0,1]$ or other bounded scores.
>
> Q3: "Why do we assume a linear class?"
>
> Response to Q3:
> On a technical level, in our theoretical results, we leverage the linearity of $H$
> to relate the first-order conditions of the LR-QR problem
> to the test-time marginal coverage under the covariate shift $r$.
> This linearity is also the key property leveraged by
> the pioneering work [2] in order to study conditional coverage.
> (For a detailed derivation of the connection
> between the first-order conditions of the LR-QR problem
> and the test-time marginal coverage,
> please see the start of Section 3
> or the proof of Proposition 4.1.)
>
> We would like to emphasize that a linear hypothesis class can be very rich and can contain highly nonlinear functions, e.g., the space of functions representable by linear combinations of the last-layer features of a pretrained model. Consequently, $r\in H$ does not imply that $r$ is linear, and we do not view the assumption that $r\in H$ as being restrictive in practice.
> Indeed, we perform experiments involving high-dimensional datasets:
> the Communities and Crime dataset has a 127-dimensional input,
> the RxRx1 dataset has a 512 pixel-by-512 pixel image input,
> and the MMLU dataset has a 1024-dimensional text input.
> In these challenging settings,
> LR-QR performs extremely well,
> which is consistent with our assumption that $r$ lies within $H$ or close to $H$.
> This underscores the fact that LR-QR is a practically significant algorithm
> in the high-dimensional setting,
> with strong and meaningful theoretical guarantees.
>
> Moreover, we would like to emphasize that such a linear function class setting is now quite standard and well-accepted in the conformal prediction literature (see e.g., [2, 3, 4]), starting with the pioneering work of [1], which has been widely adopted in the community (e.g., cited more than 100 times).
>
> [1] Gibbs et al., "Conformal prediction with conditional guarantees".
>
> [2] van der Laan et al., "Generalized Venn and Venn-Abers Calibration with Applications in Conformal Prediction".
>
> [3] Bairaktari et al., "Kandinsky Conformal Prediction".
>
> [4] Cherian et al., "Large language model validity via enhanced conformal prediction methods".
>
> [5] Vershynin. "High-dimensional probability".

---

> > ### Comment · Reviewer_8HeH · 2025-08-03
> >
> > I appreciate the author’s detailed explanation of both the linear class and the experimental results. I have a quick follow-up question regarding the boundedness condition in the theorem. For any score function $s$, whether bounded or unbounded, would it be valid to instead consider a transformed score function $g \circ s$, where $g: \mathbb{R} \to [0, 1]$ is strictly increasing, for example, $g(x) = 1 / (1 + \exp(-x))$? This would ensure that the new score $g \circ s$ satisfies the boundedness condition, and it seems the result would still hold for any original score $s$. Does this intuition seem correct? I’m not suggesting that the theorem statement or proof should be changed as the current theorem looks good to me, but I wanted to verify this idea.
> >
> > Overall, I believe this paper addresses an important and practically problem. The rebuttal further demonstrates the authors’ thoughtfulness and commitment. I will maintain my score of strong accept.

---

> > > ### Author Response · Authors · 2025-08-04
> > >
> > > We thank the reviewer
> > > for their comments,
> > > and for this insightful question and suggestion.
> > > Indeed, if one replaces
> > > an unbounded score $S$
> > > with the transformed score $g(S)\in [0,1]$,
> > > then the proof of
> > > the generalization bound
> > > in Theorem 4.2
> > > goes through verbatim.
> > >
> > > It turns out that this trick also applies
> > > to the coverage lower bound
> > > in Theorem 4.3,
> > > if we relax Condition 5 in Section E of the Appendix.
> > > Condition 5 imposes a boundedness assumption
> > > on the conditional density of $S|X=x$,
> > > and therefore precludes us from directly
> > > applying the trick suggested by the reviewer
> > > (as is explained at the end of this comment).
> > > We claim that Condition 5
> > > can be replaced
> > > with the following Condition 5':
> > > (1) the conditional density $f_{S|X=x}$ exists for all $x\in \mathcal{X}$;
> > > (2) there exists a constant $C_f > 0$
> > > and a real $k > 0$
> > > such that for all $s\in (0,1)$, we have
> > > $$ f_{S|X=x}(s) \le C_f (|s|^{-k} + |1-s|^{-k}), $$
> > > uniformly in $x\in \mathcal{X}$;
> > > (3) the basis $\Phi$ obeys
> > > $$\sup_{\gamma\in S^{d-1}} E_1[ \langle \gamma, \Phi(X)\rangle^{-k} ] < \infty $$
> > > and
> > > (4) the quantity $B_{\text{upper}}$ defined in Lemma L.4
> > > obeys $B_{\text{upper}} < C_{\Phi}^{-1}$,
> > > where $C_{\Phi}$ is defined in Condition 1.
> > >
> > > In other words, we can allow
> > > the conditional density of $S|X$
> > > to diverge at a polynomial rate
> > > near $s=0$ and $s=1$,
> > > so long as $\Phi$ obeys a certain moment condition,
> > > and so long as apriori,
> > > the population LR-QR objective
> > > can be restricted to a sufficiently small ball.
> > >
> > > To elaborate on points (3) and (4):
> > > One can consider (3) a slight strengthening
> > > of Condition 6, a quantitative independence condition
> > > on the basis functions.
> > > To understand (4),
> > > note that by inspecting the definition of $B_{\text{upper}}$,
> > > we see that an upper bound on $B_{\text{upper}}$
> > > imposes
> > > (a) a lower bound on $c_{\text{align}}$ defined in Condition 6,
> > > as well as
> > > (b) an upper bound on
> > > $E_1[ \ell_{\alpha}(\theta^{\star} r_{H}, S) ]$,
> > > which states that optimally scaling
> > > the projection $r_{H}$
> > > can yield a threshold function
> > > with low pinball loss.
> > > (In particular,
> > > by the definition of $\theta^{\star}$
> > > in Lemma L.2,
> > > we have $E_1[\ell_{\alpha}(\theta^{\star} r_{H}, S)] \le \alpha^{-1}$,
> > > hence a sufficient condition
> > > for (4) to hold
> > > is the lower bound $c_{\text{align}} > \frac{1}{4} C_{\Phi}$.)
> > >
> > > Now, we show how Condition 5'
> > > implies Theorem 4.3.
> > > In the original proof,
> > > Condition 5 is used on lines 906 and 1102
> > > in order to control expressions of the form
> > > $$ \| E_1[ \Phi(X) \Phi(X)^T f_{S|X}(h(X)) ] \|, $$
> > > uniformly for $\gamma \in R^d$ with
> > > $0 < B_{\text{lower}} \le |\gamma| \le B_{\text{upper}}$,
> > > where as usual we write $h(X) = \langle \gamma, \Phi(X)\rangle$.
> > > By Condition 1 and Jensen's inequality,
> > > this is bounded by $E_1[ |f_{S|X}(h(X))| ]$,
> > > up to constants.
> > > By part (2) of Condition 5',
> > > this in turn is bounded by
> > > $E_1[ |h(X)|^{-k} + |1-h(X)|^{-k} ]$,
> > > up to constants.
> > > By part (3) of Condition 5'
> > > and the bounds $0 < B_{\text{lower}} \le |\gamma| \le B_{\text{upper}}$,
> > > the first term
> > > $E_1[ |h(X)|^{-k} ]$
> > > is uniformly bounded.
> > > Next, by the triangle inequality,
> > > the Cauchy-Schwarz inequality,
> > > Condition 1,
> > > and part (4) of Condition 5',
> > > we have
> > > $$ |1-h(X)| \ge 1 - |h(X)| \ge 1 - C_{\Phi} B_{\text{upper}} > 0, $$
> > > so we may uniformly bound the second term by
> > > $$ E_1[ |1-h(X)|^{-k} ] \le (1-C_{\Phi} B_{\text{upper}})^{-k}. $$
> > > Putting these bounds together, we see that Condition 5'
> > > provides the desired uniform control.
> > >
> > > Finally, if we utilize Condition 5' instead of Condition 5,
> > > then the reviewer's idea
> > > indeed allows us to generalize
> > > Theorem 4.3 beyond bounded scores.
> > > Note that for the sigmoid transformation $g$
> > > that the reviewer proposed,
> > > the conditional density $f_{g(S)|X=x}$
> > > of the transformed score $g(S)$ obeys
> > > $$f_{g(S)|X=x}(t) = \frac{1}{t(1-t)} f_{S|X=x}\left( \log \frac{t}{1-t} \right)$$
> > > for all $t\in (0,1)$.
> > > Consequently,
> > > if the original density $f_{S|X=x}(s)$
> > > is supported on $R$
> > > with polynomially-decaying tails
> > > as $s\to \pm \infty$,
> > > then the transformed density
> > > diverges like $\sim (t(1-t))^{-(1+o(1))}$
> > > as $t\to 0,1$, which satisfies Condition 5'
> > > with $k = 1+\epsilon$ for any $\epsilon > 0$.
> > >
> > > We thank the reviewer for this question and
> > > their keen suggestion,
> > > and we are happy to include these remarks
> > > in the revised version, to emphasize the generality
> > > of our findings.

---

> > > > ### Comment · Reviewer_8HeH · 2025-08-05
> > > >
> > > > I appreciate the authors’ continued effort to further clarify and strengthen their results. Their proposed extension demonstrates the robustness and flexibility of the framework, and I believe it helps reinforce the theoretical contribution of the paper. My overall assessment remains positive and reflects this development.

---

### Official Review · Reviewer_KYG7 · 2025-06-27

**Clarity:** 2
**Significance:** 1
**Originality:** 3
**Rating:** 4
**Confidence:** 4

**Summary:**

The authors address the problem of covariate shift in Conformal Prediction. The standard approach to establishing CP validity in that scenario requires estimating a likelihood ratio. The paper describes a computationally advantageous approach based on regularising the pinball objective used in quantile regression.

**Questions:**

- Does the "linear hypothesis class" contain non-linear regression models?
- Setting $g=dP/dQ$ implicitly implies that the likelihood ratio is in the linear hypothesis class. How restrictive is this assumption?
- The validity in Equation 2 is different from the finite-sample validity of Conformal Prediction. In [1], a threshold correction is estimated on a calibration set to conformalize the obtained intervals. How does that approach relate to the finite-sample setup in Section 4.2?
- Does the correction in Theorem 4.3 account for the non-optimality of the likelihood ratio estimator, h? Can a similar bound be obtained for non-linear $h$?

[1]
Romano, Yaniv, Evan Patterson, and Emmanuel Candes. *Conformalized quantile regression.* Advances in neural information processing systems 32 (2019).

**Ethical Concerns:**

["NO or VERY MINOR ethics concerns only"]

**Final Justification:**

The authors provided all needed clarifications during the discussion period. I raised my score to 4 and would vote to accept the paper.

**Limitations:**

Yes

**Quality:**

3

**Strengths And Weaknesses:**

**Strength**

The observation linking the optimum of the pin-ball objective and coverage under distribution shift is interesting. Is this the first time this has been observed and used to build an estimation approach?

**Weaknesses**

It is unclear why the likelihood ratio *may be inaccurate for high-dimensional data*. As this is the motivation for their work, the authors can spend a few more words explaining the reasons for the claim. One of the simplest likelihood ratio estimators is based on fitting a logistic regression model, whose performance intuitively increases with the input dimensionality. The authors could have also run an ablation experiment to show that the likelihood ratio estimation becomes inaccurate in high dimensions.

The approach seems to require estimating a model for the likelihood ratio, exploiting the change-of-measure identity. The authors should clarify whether the idea has appeared elsewhere, why it can not be used to direct likelihood ratio estimation, and how it removes the high-dimensional inaccuracy.

---

> ### Author Rebuttal · Authors · 2025-07-27
>
> We thank the reviewer for the helpful comments.
>
> Strength 1:  "The observation linking the optimum of the pin-ball objective and coverage."
>
> Response to Strength 1:
> This connection is known from the works [2, 3], as we have already stated in the paper (see line 121). However, our main innovation is the regularization term, which is both conceptually novel and significantly improves performance. We will be more clear about this in the revision.
>
> Weakness 1: "Why is high-dimensionality hard?"
>
> Response to Weakness 1:
> In general, estimating a high-dimensional function incurs the "curse of dimensionality", whereby accurate estimation at a single point requires an exponential number of nearby datapoints. More specifically, high-dimensional estimation has been widely studied in statistics and ML, and there are many results showing that even under pretty strong smoothness assumptions, this is very hard [5].
>
> Given that the likelihood ratio is a high-dimensional function when $\text{dim}(\mathcal{X})$ is large, likelihood ratio estimation suffers from similar hardness results.
> As the reviewer suggests, a natural method for likelihood ratio estimation is through logistic regression.
> Specifically, given i.i.d. samples $\{ (X_i, Y_i) : i\in [n] \}$ with $X_i\in \mathbb{R}^d$ and $Y_i\in \{0,1\}$,
> one can fit a logistic regression model to obtain
> a probabilistic classifier $\hat p(x)$ that returns the predicted probability
> that $x$ was sampled from class 1;
> then, one returns the likelihood ratio estimate $\hat r(x) = \hat p(x) / (1 - \hat p(x))$.
> However, in the literature, theoretical results show that logistic regression
> incurs large estimation errors in high dimension; see e.g., [8].
> Here, we numerically illustrate this phenomenon with the following simple setting.
> Given a dimension $d$,
> and given a fixed sample size $n_{train} = 10000$,
> we sample $n_{train}/2$ datapoints from $N(-\frac{\mathbf{1}}{\sqrt d}, \frac{1}{d} I_d)$
> and $n_{train}/2$ datapoints from $N(+\frac{\mathbf{1}}{\sqrt d}, \frac{1}{d} I_d)$,
> where $\mathbf{1}$ denotes the vector of all ones,
> and $I_d$ denotes the identity covariance matrix.
> Here, datapoints from $N(-\frac{\mathbf{1}}{\sqrt d}, \frac{1}{d} I_d)$ form class 0
> and datapoints from $N(+\frac{\mathbf{1}}{\sqrt d}, \frac{1}{d} I_d)$ form class 1.
> (We choose this scaling in order to ensure that the signal-to-noise ratio
> remains of constant order as we vary $d$.)
> In this setting, the true likelihood $p(x)$ satisfies
> $\log(\frac{p(x)}{1-p(x)}) = \beta_0 + \beta^\top x$ for $x\in \mathbb{R}^d$,
> where $\beta_0 = 0$ and $\beta = 2\sqrt{d} \mathbf{1}$.
> We fit a logistic regression model on the $n_{train}$ samples
> to obtain the fitted parameters $\hat \beta_0$ and $\hat \beta$.
> We vary the dimension $d \in \{ 10, 30, 50, 70, 90 \}$,
> and consider the $\ell_2$ estimation error $(\|\hat \beta_0 - \beta_0\|^2 + \|\hat \beta - \beta\|^2)^{1/2}$
> (where $\|\cdot\|$ denotes $\ell_2$-norm),
> averaged over $n_{batches} = 100$ independent draws of the training data.
> The results (mean ± standard deviation) are given below.
> As we see, the risk increases rapidly with dimension, which illustrates the curse of dimensionality.
> Consequently, the resulting likelihood ratio estimator $\hat p(x) / (1-\hat p(x))$
> will be far from the true likelihood ratio $p(x) / (1-p(x))$, as well.
>
> d = 10: risk = 1.1e+1 ± 1.2e-1
>
> d = 30: risk = 5.2e+1 ± 1.6e-2
>
> d = 50: risk = 9.3e+1 ± 1.1e-2
>
> d = 70: risk = 1.3e+2 ± 8.1e-3
>
> d = 90: risk = 1.7e+2 ± 7.0e-3
>
> Weakness 2 : "Clarify whether the idea has appeared elsewhere."
>
> Response to Weakness 2:
> Our approach avoids the direct estimation of the likelihood ratio, and in this sense it is novel; we are not aware of this technique having appeared in this exact form before.
> Most prior works on conformal prediction require
> direct estimation of the likelihood ratio,
> which as explained in the response to Weakness 1 above,
> incurs the curse of dimensionality
> in high-dimensional settings.
> While related techniques have appeared in other works on covariate shift
> outside of the conformal prediction literature
> (we cite the paper [66], Zhang et al.),
> we believe that our specific regularizer is novel.
>
> Weakness 3: "How does it remove the high-dimensional inaccuracy?"
>
> Response to Weakness 3:
> After using the change of measure formula, the regularizer/quantity that we need to estimate becomes an average over a distribution from which we have a data set. By estimating this with a sample average, we obtain high accuracy, while avoiding the need to estimate the high-dimensional likelihood ratio function.
>
> Q1 + Q2: "Does $H$ contain nonlinear functions? Is it restrictive?"
>
> Response to Q1 + Q2:
> We would like to emphasize that a linear hypothesis class can be very rich and can contain highly nonlinear functions.
> In our setting, we call a set of functions $H$
> from $\mathcal{X}\to \mathbb{R}$
> a "linear hypothesis class" if
> (1) $H$ forms a finite-dimensional vector space
> and
> (2) for each function $h\in H$, the second moment $E_1[|h(X)|^2]$ is finite.
> In particular,
> the space of functions representable by linear combinations of the last-layer features of a pretrained model
> is an example of a linear hypothesis class.
> Consequently, $r\in H$ does not imply that $r$ is linear, and we do not view the assumption that $r\in H$ as being restrictive in practice.
>
> On a technical level, in our theoretical results, we leverage the linearity of $H$
> to relate the first-order conditions of the LR-QR problem
> to the test-time marginal coverage under the covariate shift $r$.
> This linearity is also the key property leveraged by
> the pioneering work [2] in order to study conditional coverage.
> (For a detailed derivation of the connection
> between the first-order conditions of the LR-QR problem
> and the test-time marginal coverage,
> please see the start of Section 3
> or the proof of Proposition 4.1.)
>
> Further, we would like to note that our experiments are consistent with our regularity conditions.
> We perform experiments involving very high-dimensional datasets:
> the Communities and Crime dataset has a 127-dimensional input,
> the RxRx1 dataset has a 512 pixel-by-512 pixel image input,
> and the MMLU dataset has a 1024-dimensional text input.
> In these challenging settings,
> LR-QR performs extremely well,
> which is consistent with the two regularity conditions discussed above.
> This underscores the fact that LR-QR is a practically significant algorithm
> in the high-dimensional setting,
> with strong and meaningful theoretical guarantees.
>
> Moreover, we would like to emphasize that such a linear function class setting is now quite standard and well-accepted in the conformal prediction literature (see e.g., [9, 10, 11]), starting with the pioneering work of [2], which has been widely adopted in the community (e.g., cited more than 100 times).
>
> Q3: "Not a finite-sample guarantee, compare to [1]."
>
> Response to Q3:
> Compared to [1], our setting is fundamentally different and more difficult, as it considers coverage under covariate shift. This is more closely related to the works [6,7], and others that we compare with. In this setting, it is known from [7] that it is *impossible* to obtain finite-sample coverage with a non-trivial prediction set in general. Motivated by this result, we aim to establish asymptotic coverage bounds. In contrast, [1] considers the simpler setting of no distribution shift, which explains why they are able to obtain finite-sample coverage with non-trivial prediction sets.)
>
> Standard exchangeability arguments used in conformal prediction (e.g., in [1]) do not apply to our setting. Therefore,
> to achieve these bounds, we need to use different and novel theoretical tools (e.g., the stability result in Lemma G.1).
> We are able to derive a coverage lower bound with error terms that holds with high probability over the calibration data.
> The good empirical performance of the LR-QR algorithm in our experiments suggests that in practice, the coverage has an excellent finite-sample behavior.
>
>
> Q4: "Correction for non-optimality/misspecification of the h estimator?"
>
> Response to Q4:
> First, we would like to clarify that in the LR-QR algorithm, we never estimate $h$. Therefore, there is no need to consider a correction.
>
> However, implicitly, our methods do impose the condition that the likelihood ratio is close to the hypothesis class $H$. Indeed, the last term $-(1-\alpha) E_1[|r(X)-r_B(X)|]$
> in the lower bound in Theorem 4.3 accounts for the fact that the hypothesis class $H$ might not contain the true likelihood ratio $r$.
> This quantity is proportional to the $L^1$ distance from $r$ to $H$; hence, the more misspecified $H$ is, the weaker our guarantee becomes.
>
> Regarding the other parts of the question, we would like to emphasize that the hypothesis class $H$ can include nonlinear functions $h$, and that the LR-QR algorithm does not require estimating the likelihood-ratio directly. Indeed, in our experiments we use linear combinations of features of pre-trained foundation models (LLMs) for multiple choice Q&A, which are highly non-linear functions of the original input text.
>
> [2] Gibbs et al., "Conformal prediction with conditional guarantees".
>
> [3] Jung et al., "Batch multivalid conformal prediction".
>
> [4] Foygel Barber et al., "The limits of distribution-free conditional predictive inference".
>
> [5] Wainwright, "High-dimensional statistics".
>
> [6] Tibshirani et al., "Conformal prediction under covariate shift".
>
> [7] Yang et al., "Doubly Robust Calibration of Prediction Sets under Covariate Shift".
>
> [8] Sur et al., "A modern maximum-likelihood theory for high-dimensional logistic regression".
>
> [9] van der Laan et al., "Generalized Venn and Venn-Abers Calibration with Applications in Conformal Prediction".
>
> [10] Bairaktari et al., "Kandinsky Conformal Prediction".
>
> [11] Cherian et al., "Large language model validity via enhanced conformal prediction methods".

---

> > ### Comment · Reviewer_KYG7 · 2025-08-05
> > **Thank you for your answers**
> >
> > Many thanks for your detailed replies and the new experiment. If I understand correctly, the results shown in the rebuttal are for the estimation error, $(|\hat \beta_0 - \beta_0|^2 + |\hat \beta - \beta|^2)^{1/2}$. Can't the ratio estimation be more stable? What are the figures for $\left| \hat p(x) / (1-\hat p(x)) - p(x) / (1-p(x))\right|$?
> >
> > Works like [1] or [2] give coverage bounds even in the case of a general distribution shift. Could similar techniques, together with suitable assumptions on the oracle distributions, be used to provide finite-sample guarantees for the proposed estimation approach?
> >
> > [1]
> > Barber, Rina Foygel, et al. "Conformal prediction beyond exchangeability." The Annals of Statistics 51.2 (2023): 816-845.
> > [2]
> > Cauchois, Maxime, et al. "Robust validation: Confident predictions even when distributions shift." Journal of the American Statistical Association 119.548 (2024): 3033-3044.

---

> > > ### Author Response · Authors · 2025-08-05
> > >
> > > We thank the reviewer for their comments
> > > and their insightful questions.
> > >
> > > Regarding our simulation:
> > > Below, we have added a new table of
> > > likelihood ratio estimation errors
> > > over a grid of six $x$-values,
> > > averaged over $n_{\text{batches}} = 100$
> > > independent draws of the $n_{\text{train}} = 10000$ datapoints.
> > > For each dimension $d$
> > > and each point $x$ in the associated grid $G_d$,
> > > we report the mean value of
> > > the likelihood ratio estimation error
> > > $ \left| \frac{\hat p(x)}{1-\hat p(x)} - \frac{p(x)}{1-p(x)} \right| $.
> > > To compute this,
> > > we leverage the identities
> > > $\log\left( \frac{p(x)}{1-p(x)} \right) = \beta_0 + \beta^\top x$
> > > and
> > > $\log\left( \frac{\hat p(x)}{1-\hat p(x)} \right) = \hat \beta_0 + \hat \beta^\top x$,
> > > which imply that
> > > the error equals
> > > $$ | \exp(\hat \beta_0 + \hat \beta^\top x) - \exp(\beta_0 + \beta^\top x) |. $$
> > > For a given dimension $d$,
> > > we use the grid of four $x$-values
> > > $$ G_d = \left[
> > > -0.6 \frac{\mathbf{1}}{\sqrt d},
> > > -0.2 \frac{\mathbf{1}}{\sqrt d},
> > > +0.2 \frac{\mathbf{1}}{\sqrt d},
> > > +0.6 \frac{\mathbf{1}}{\sqrt d}
> > > \right], $$
> > > which lie on the segment connecting
> > > the class means.
> > >
> > > In the table below,
> > > for a given dimension $d$,
> > > each column corresponds
> > > to the point $x = (\text{column label}) \frac{\mathbf{1}}{\sqrt d} \in R^d$
> > > at which we evaluate
> > > the error.
> > > (E.g., row $d$ and column $+0.2$
> > > corresponds to $x = +0.2 \frac{\mathbf{1}}{\sqrt d}$.)
> > > By inspecting the columns,
> > > we see that the estimation error
> > > increases with dimension.
> > >
> > > __Table:__ absolute error of likelihood ratio estimation.
> > >
> > > | dim \ x-value | -0.6 | -0.2 | +0.2 | +0.6 |
> > > |---------------|------|------|------|------|
> > > | 10 | 3e-3±3e-4 | 1e-1±1e-2 | 5e+1±6e-1 | 2e+5±3e+1 |
> > > | 30 | 1e-2±2e-4 | 2e-1±5e-3 | 2e+5±1e-1 | 4e+15±7e+0 |
> > > | 50 | 1e-2±2e-4 | 2e-1±3e-3 | 5e+8±5e-2 | 1e+26±2e+11 |
> > > | 70 | 2e-2±1e-4 | 3e-1±2e-3 | 1e+12±4e-2 | 3e+36±6e+21 |
> > > | 90 | 2e-2±1e-4 | 3e-1±2e-3 | 4e+15±7e+0 | 8e+46±1e+32 |
> > >
> > > Regarding the relation between [1, 2] and LR-QR:
> > > we thank the reviewer for bringing up these works.
> > > Indeed, [1, 2] provide finite-sample coverage lower bounds
> > > in the case of general distribution shifts,
> > > including covariate shifts.
> > >
> > > However, we would like to point out
> > > that the coverage lower bounds in [1, 2]
> > > depend on certain quantities
> > > that are difficult to estimate in practice.
> > > For instance,
> > > in Theorem 2 of [1],
> > > the coverage gap
> > > is given by a weighted average
> > > of the total variation distances
> > > $d_{\text{TV}}(R(Z), R(Z^{(i)}))$
> > > for $i\in [n]$.
> > > Thus,
> > > for a practitioner
> > > to set the nominal coverage level,
> > > they need to estimate
> > > these total variation distances;
> > > and we are not aware
> > > of any existing algorithms
> > > that allow for efficient estimation
> > > of total variation distances
> > > in high dimensions.
> > >
> > > Similarly,
> > > in Corollary 2.1 of [2],
> > > the coverage gap
> > > is given by $c_{\alpha,\rho,f}/(n+1)$,
> > > where the quantity $\rho$
> > > is an upper bound
> > > on the $f$-divergence
> > > $D_f(P_{\text{test}} || P_0)$
> > > between calibration and test.
> > > As a result,
> > > a practitioner
> > > must estimate an $f$-divergence,
> > > and the estimation of $f$-divergence measures
> > > is known to be statistically hard
> > > in general.
> > > For instance, in Krishnamurthy et al. [3],
> > > the authors prove that
> > > in dimension $d$,
> > > the minimax estimation error of an $\alpha$-divergence
> > > is bounded below by
> > > $O(n^{-\frac{4s}{4s+d}})$
> > > if the Hölder smoothness $s$
> > > of the densities
> > > obeys $s\le d/4$,
> > > which is prohibitively large in high dimensions.
> > >
> > > In each of our experiments,
> > > we compare LR-QR
> > > against the
> > > DRO-CP algorithm from [2],
> > > and our results
> > > definitively show
> > > that LR-QR outperforms DRO-CP.
> > > LR-QR closely tracks the nominal coverage level
> > > with small set sizes,
> > > while DRO-CP
> > > tends to overcover
> > > with overly conservative prediction sets.
> > > (DRO-CP is the fifth bar
> > > in each group
> > > in Figure 1;
> > > the original legend
> > > had a typo,
> > > which we will fix
> > > in the revised version.)
> > > Regarding comparing with
> > > the Nonexchangeable CP method
> > > from [1],
> > > note that the LR-QR algorithm
> > > only requires **unlabeled** data
> > > from the target domain,
> > > whereas the method
> > > from [1]
> > > relies on labeled data.
> > > Thus,
> > > it is not clear to us
> > > whether the Nonexchangeable CP method
> > > can be run in our experiments,
> > > where no labeled target data is provided.
> > >
> > > Finally, the reviewer asks the natural question
> > > as to whether LR-QR
> > > can be extended to general shifts.
> > > If we had access
> > > to labeled data from the target domain,
> > > it might be possible
> > > to modify our choice of hypothesis class,
> > > or to incorporate ideas from full conformal prediction.
> > > At present,
> > > we do not see
> > > an easy way
> > > to generalize,
> > > but we believe this is a very interesting direction
> > > for future work.
> > >
> > > We thank the reviewer for these questions
> > > and for their interest.
> > > We hope that our comments
> > > clarify the practical utility
> > > of LR-QR in challenging high-dimensional settings,
> > > and its relation to other work.
> > >
> > > [1] Barber et al., "Conformal prediction beyond exchangeability".
> > >
> > > [2] Cauchois et al., "Robust validation: Confident predictions even when distributions shift".
> > >
> > > [3] Krishnamurthy et al., "Nonparametric Estimation of Renyi Divergence and Friends".

---

> > > > ### Comment · Reviewer_KYG7 · 2025-08-06
> > > > **Thank you again!**
> > > >
> > > > Thank you for the new table. Plotting or tabling those results against the error of the proposed approach, e.g. ${\rm E}(|h(X) - r(X)|)$, on the same data would help the reader appreciate the need and efficiency of the proposed method.
> > > >
> > > > I agree that estimating the distribution divergences from samples is challenging. Can the proposed approach help? For example, $KL(P, Q) = \sum_x p(x) \log(\frac{p(x)}{q(x)}) = E_{X \sim p} \log(r(Z)) $, where $r$ is the likelihood ratio. Would $E_{X \sim p} \log(h(X)) $ be a good approximation of $KL(P, Q)$?

---

> > > > > ### Author Response · Authors · 2025-08-07
> > > > >
> > > > > We thank the reviewer for their
> > > > > detailed questions.
> > > > >
> > > > > Regarding the likelihood ratio
> > > > > estimation error
> > > > > of the function $\beta h$:
> > > > > In order to illustrate the
> > > > > estimation error phenomenon,
> > > > > here, we extend the previous simulation
> > > > > to the following regression task
> > > > > under covariate shift.
> > > > > (This is necessary for us
> > > > > to run LR-QR on the example.)
> > > > > For a given dimension $d$,
> > > > > the marginal calibration distribution
> > > > > $P_{1,X}$
> > > > > is given by
> > > > > $N(-\frac{\mathbf{1}}{\sqrt d}, \frac{1}{d} I)$,
> > > > > and the marginal test distribution
> > > > > $P_{2,X}$
> > > > > is given by
> > > > > $N(+\frac{\mathbf{1}}{\sqrt d}, \frac{1}{d} I)$.
> > > > > (Consequently,
> > > > > the true likelihood ratio
> > > > > $r(x) = \frac{dP_{1,X}}{dP_{2,X}}(x)$
> > > > > coincides with the example given
> > > > > in our previous comment.)
> > > > > Additionally,
> > > > > for any feature vector
> > > > > $x = (x_1,\ldots,x_d)^\top \in R^d$,
> > > > > its label $Y\in R$ is sampled
> > > > > from the normal distribution
> > > > > $Y|X=x \sim N(x_1, \frac{1}{d})$.
> > > > > We fit a linear regression model
> > > > > on $n_{\text{train}} = 10000$
> > > > > samples from from $P_1$.
> > > > > We then run LR-QR
> > > > > with $n_1 = 5000$ labeled calibration samples,
> > > > > $n_2 = 10000$ unlabeled test samples,
> > > > > and $n_3 = 5000$ unlabeled calibration samples,
> > > > > with the hypothesis class
> > > > > $$ H = ( h(x) = \gamma_0 + \gamma^\top x : \gamma_0\in R, \gamma\in R^d ) $$
> > > > > of affine functions,
> > > > > in dimension $d=10$,
> > > > > with the nominal miscoverage level
> > > > > $\alpha = 0.1$,
> > > > > for multiple values
> > > > > of the regularization strength $\lambda$
> > > > > (listed in the table below).
> > > > > As the reviewer suggests,
> > > > > we evaluate the $L^1$ estimation error
> > > > > $\hat E_1[ |\beta h - r| ]$
> > > > > using $m = 10000$
> > > > > samples from $P_1$;
> > > > > as well as the
> > > > > empirical test-time coverage
> > > > > using $m = 10000$
> > > > > samples from $P_2$.
> > > > > We average over
> > > > > $n_{\text{batch}} = 20$
> > > > > evaluation sets.
> > > > > (We utilize the function $\beta h$
> > > > > instead of $h$,
> > > > > because in the LR-QR objective,
> > > > > the regularization is defined as
> > > > > $\lambda E_1[(\beta h - r)^2]$;
> > > > > hence, it is most natural
> > > > > to compare $\beta h$ and $r$.)
> > > > >
> > > > > From the results
> > > > > in the table,
> > > > > we see that
> > > > > as we sweep $\lambda$,
> > > > > the $L^1$ distance
> > > > > $E[|\beta h - r|]$
> > > > > between $\beta h$ and $r$
> > > > > remains extremely large,
> > > > > consistently across all $\lambda$.
> > > > > (To put this $L^1$ distance in perspective,
> > > > > note that by the change-of-measure identity,
> > > > > we have $E_1[|r|] = 1$ for all $\lambda$;
> > > > > further demonstrating how inaccurate $\beta h$ is
> > > > > as an estimator of $r$.)
> > > > > However, the test-time coverage
> > > > > of the LR-QR
> > > > > is well-behaved,
> > > > > hovering near the nominal $1-\alpha = 0.9$,
> > > > > with the optimal $\lambda$ around 1e-4.
> > > > > In summary, these empirical results
> > > > > demonstrate that LR-QR
> > > > > achieves satisfactory test-time coverage,
> > > > > without attempting to directly estimate
> > > > > the likelihood ratio $r$.
> > > > >
> > > > > Table: LR-QR $L^1$ error and coverage in dimension $d=10$.
> > > > >
> > > > > | Lambda | 1e-6 | 1e-5 | 1e-4 | 1e-3 | 1e-2 |
> > > > > |---------|----------|----------|----------|----------|----------|
> > > > > | L1 error | 6e+15 | 2e+16 | 3e+16 | 8e+15 | 2e+16 |
> > > > > | Coverage | 0.864 | 0.860 | 0.867 | 0.943 | 1.000 |
> > > > >
> > > > >
> > > > > Regarding estimating divergences
> > > > > using the LR-QR threshold:
> > > > > The reviewer is right to note
> > > > > that by the change-of-measure identity,
> > > > > the KL divergence between
> > > > > the calibration and test marginal distributions
> > > > > can be expressed as
> > > > > $\text{KL}(P_{1,X} || P_{2,X}) = E_{X\sim P_{1,X}}[\log r(X)]$,
> > > > > where $r$ denotes the true likelihood ratio.
> > > > > In the population-level LR-QR objective,
> > > > > we include a regularization term
> > > > > which, up to constants,
> > > > > equals
> > > > > $\lambda E_1[ (\beta h - r)^2 ]$;
> > > > > hence, one might expect
> > > > > the function $\beta h$
> > > > > to be a good approximation of
> > > > > the true likelihood ratio $r$.
> > > > > From the empirical results above,
> > > > > we have demonstrated
> > > > > that this is not the case.
> > > > > Furthermore, we can provide
> > > > > a theoretical explanation
> > > > > for why this intuition is incorrect.
> > > > > The intuition that "$\beta h \approx r$"
> > > > > fails to take into account
> > > > > the first term in
> > > > > the population-level LR-QR objective,
> > > > > i.e., the pinball loss $E_1[\ell_{\alpha}(h, S)]$.
> > > > > The LR-QR algorithm trades-off these two losses,
> > > > > "shrinking" the true likelihood ratio $r$
> > > > > towards the minimizer of the pinball loss,
> > > > > i.e., the conditional quantile function
> > > > > $q_{1-\alpha}(x) := \text{Quantile}_{1-\alpha}(Y|X=x)$.
> > > > > Consequently,
> > > > > unless the regularization $\lambda$ is $+\infty$
> > > > > (which cannot occur in practice),
> > > > > the LR-QR algorithm
> > > > > returns a pair $(\beta, h)$
> > > > > such that $\beta h \neq r$;
> > > > > and hence,
> > > > > by design,
> > > > > the LR-QR algorithm
> > > > > does not attempt to
> > > > > directly estimate the
> > > > > likelihood ratio $r$.
> > > > >
> > > > > We will elaborate
> > > > > on all of these points
> > > > > in the revised paper;
> > > > > due to the space limits
> > > > > and time constraints
> > > > > of the conference's discussion period,
> > > > > we are unable to provide
> > > > > complete details here.
> > > > > We thank the reviewer again
> > > > > for their interest in our work.
> > > > > We hope our comments clarify the fact
> > > > > that LR-QR successfully
> > > > > achieves valid test-time coverage,
> > > > > while circumventing
> > > > > the fundamental challenge of estimating
> > > > > a high-dimensional likelihood ratio.

---

### Official Review · Reviewer_wABN · 2025-06-28

**Clarity:** 4
**Significance:** 3
**Originality:** 3
**Rating:** 5
**Confidence:** 5

**Summary:**

This paper addresses the problem of conformal prediction under covariate shift, where the distribution of features differs between a source (calibration) and a target (test) domain. The authors propose a new method called Likelihood Ratio Regularized Quantile Regression (LR-QR) to construct prediction sets that achieve valid coverage in the target domain without explicitly estimating the density ratio $r(x)=dP_{2,X}/dP_{1,X}$ (which can be prohibitive at high dimensions). The key idea is to incorporate a regularization term into a quantile regression target that implicitly encourages the learned threshold function $h(x)$ to adapt to the unknown likelihood ratio.

The paper addresses the problem of constructing conformal prediction sets under covariate shift—when calibration data comes from a source distribution $P_1$ and test data from a different target distribution $P_2$ sharing the same $Y\mid X$ but with shifted marginals (pp. 1–2). Existing methods often estimate the density-ratio

$$
r(x)\;=\;\frac{dP_{2,X}}{dP_{1,X}}(x),
$$

which is challenging in high dimensions (p. 2).

---

## Method (LR-QR)

The authors propose **Likelihood-Ratio Regularized Quantile Regression** (LR-QR), which replaces direct ratio estimation with a novel regularizer in a quantile-regression objective (pp. 3–4). Concretely, they solve

$$
\min_{h,\;\beta}\;\mathbb{E}_{P_1}\bigl[\ell_\alpha\bigl(h(X),S(X,Y)\bigr)\bigr]
\;+\;\lambda\;\mathbb{E}_{P_1}\bigl[(\beta\,h(X)-r(X))^2\bigr],
$$

involving only \(\mathbb{E}_{P_1}[h^2]\) and \(\mathbb{E}_{P_2}[h]\), both easily estimated from unlabeled samples (pp. 4–5; Algorithm 1).

---

## Theory

* **Population regime:**
  If the hypothesis class contains the true ratio $r$, LR-QR attains exact $(1-\alpha)$-coverage under $P_2$ (Prop. 4.1, p. 5).

* **Finite-sample regime:**
  They prove a generalization bound (Theorem 4.2, p. 6) showing the empirical minimizer’s risk is within

  $$
    O\Bigl(n_1^{-\tfrac13}\,(n_2+n_3)^{-\tfrac13}\Bigr)
  $$

  of optimal, and derive a coverage guarantee (Theorem 4.3, p. 7) that the test-domain coverage is at least $(1-\alpha)$ minus controllable error terms.  The optimal regularization scales as

  $$
    \lambda \;\propto\; n_1^{-\tfrac13}\,(n_2+n_3)^{-\tfrac13}
    \quad\text{(Eq. 4, p. 6).}
  $$

---

## Empirical Results

On three high-dimensional benchmarks, LR-QR consistently tracks nominal coverage and yields competitive (often smaller) set sizes:

1. **Communities & Crime** regression under four race-based splits (Fig. 1).
2. **RxRx1** cell-image classification across 14 biological batches (Fig. 2).
3. **MMLU** multiple-choice with an LLM (Table 1).

An ablation over $\lambda$ confirms that moderate regularization (theory-predicted regime) best balances coverage and efficiency (Fig. 4, p. 14).

---

**Overall**, LR-QR offers a scalable, theoretically grounded approach to conformal prediction under high-dimensional covariate shift **without** explicit ratio estimation, backed by strong empirical performance.

**Questions:**

- A minor problem in terms of technical quality is the fact that the theory is based on certain assumptions. For example, the main coverage guarantee assumes a linear hypothesis class (and even then conditions such as strong convexity or an independence condition for features according to Appendix E/F are required). In practice, authors restrict $H$ to linear functions (often a linear layer over a pre-trained representation), which is consistent with theory, but this can limit the flexibility of $H$ to approximate $r(x)$ when $r(x)$ is very complex. It is understandable that generalizing the theory to nonlinear or very large function classes is challenging (due to overfitting concerns), and the work mitigates this problem by using representation learning (pre-trained models) to make linear $H$ more expressive. Nonetheless, the overfitting guarantee is asymptotic in finite samples and up to an error term, and it depends on conditions that are not completely under the control of the experimenter (e.g., how well $r(x)$ lies within the range of the chosen features). This is not a critical flaw, – just a recognition that, as with any theory-heavy work, practical performance depends on how well reality matches assumptions. - The experimental results suggest that the method is robust, but the paper could be strengthened by a brief discussion of how sensitive the coverage is to misspecification of $H$ or violation of the assumptions (the appendix addresses this with the distance term $||r - r_H||$, but it might be worth highlighting this in the main text). Another minor quality issue is that the method introduces an additional hyperparameter $\lambda$; while theory dictates the order, in practice the exact choice is tuned – however, the paper handles this well via cross-validation, so I don’t see it as a major drawback.
- One area where clarity could be improved is the explanation of the “Algorithmic Principles” (section 3) – in particular lines 117–129 of the draft. The derivation presented there is somewhat difficult to follow and seems a little sloppy in its current form. In this section, the authors use the first-order optimality conditions of quantile loss minimization to argue that if the hypothesis class $H$ contains a function proportional to the ratio of the true density, then the learned predictor achieves exact coverage under this covariate shift. While the idea is sound, the explanation is rather poor. Essentially, it imposes the condition that at the optimum $h^(x)$, the expected gradient in each direction $g(x)$ in $H$ is zero. By inserting a special choice $g(x) = \frac{dQ_X}{dP_{1,X}}(x)$ (the Radon-Nikodym derivative of a shifted distribution $Q$ after that of the source distribution), they deduce that $\mathbb{E}_{P_1}[g(X)(\mathbf{1}{S \le h^(X)} - (1-\alpha))] = 0$ implies that $\mathbb{E}_{Q}[,\mathbf{1}{S(X,Y) \le h^(X)},] = 1-\alpha$. In other words, the prediction set defined by $h^$ has an exact $(1-\alpha)$ coverage under any covariate shift $Q$ whose density ratio is in $H$. The paper’s presentation quickly skips through these steps. I find the argument in Cherian and Gibbs clearer to follow.

**Ethical Concerns:**

["NO or VERY MINOR ethics concerns only"]

**Limitations:**

Yes

**Quality:**

4

**Strengths And Weaknesses:**

The technical quality of the paper is high. The methodology is derived from sound principles of quantile regression and importance weighting, and the authors provide a rigorous theoretical analysis to support their claims. In particular, Proposition 4.1 and its corollaries prove that the LR-QR solution achieves valid target coverage in the population setting if the hypothesis class contains the true density ratio (and even if it does not, one can quantify the coverage gap). This is a non-trivial theoretical guarantee, and the proof (which has been moved to the appendix) seems to use tools from learning theory (stability bounds, etc.) to deal with finite sample deviations. The paper also relies on a novel analysis of coverage by stability – this suggests that the theoretical results are quite advanced and not just a routine application of existing theorems. The derivations appear to be mathematically sound, given the stated regularity conditions (e.g. linear hypothesis space, integrability and convex assumptions in Appendix E). I found no errors in the reasoning, and important steps  are adequately justified (footnote 1). The authors even derive a guideline for the choice of the regularization parameter $\lambda$ in practice based on the trade-off of terms of order $1/\sqrt{n_1}$ and $1/\sqrt{n_2}$ (number of labeled vs. unlabeled points), which is a valuable insight to avoid ad hoc tuning.

Experimental validation is another strong point: the paper evaluates LR-QR in three different domains (tables, vision, NLP) and compares it to six baselines, showing superior or comparable performance. It is shown that LR-QR achieves coverage very close to the nominal 90% in the target domain, whereas some baselines fall short (e.g., the default split conformal method under Shift drops to ~78% coverage) or overshoot (some robust methods exceed 95% coverage because they are too conservative). At the same time, LR-QR maintains smaller prediction sets than the overly conservative methods, indicating a favorable trade-off between validity and efficiency. The authors also perform  cross-validation to select $\lambda$ (theory-guided) without using test labels, and an ablation study of regularization (Appendix B) to provide deeper insight.

---

> ### Author Rebuttal · Authors · 2025-07-27
>
> We thank the reviewer for the helpful comments.
>
> Q1: "Justify regularity conditions."
>
> Response to Q1:
> We thank the reviewer for the insightful questions about the regularity conditions.
>
> Regarding the conditions on $H$,
> we would like to emphasize that a linear hypothesis class can be very rich and can contain highly nonlinear functions.
> In our setting, we call a set of functions $H$
> from $\mathcal{X}\to \mathbb{R}$
> a "linear hypothesis class" if
> (1) $H$ forms a finite-dimensional vector space
> and
> (2) for each function $h\in H$, the second moment $E_1[|h(X)|^2]$ is finite.
> In particular,
> the space of functions representable by linear combinations of the last-layer features of a pretrained model
> is an example of a linear hypothesis class.
> Consequently, $r\in H$ does not imply that $r$ is linear, and we do not view the assumption that $r\in H$ as being restrictive in practice.
>
> On a technical level, in our theoretical results, we leverage the linearity of $H$
> to relate the first-order conditions of the LR-QR problem
> to the test-time marginal coverage under the covariate shift $r$.
> This linearity is also the key property leveraged by
> the pioneering work [2] in order to study conditional coverage.
> (For a detailed derivation of the connection
> between the first-order conditions of the LR-QR problem
> and the test-time marginal coverage,
> please see the start of Section 3
> or the proof of Proposition 4.1.)
>
> Moreover, we would like to emphasize that such a linear function class setting is now quite standard and well-accepted in the conformal prediction literature (see e.g., [2, 3, 4]), starting with the pioneering work of [1], which has been widely adopted in the community (e.g., cited more than 100 times).
>
> Regarding the effect of misspecification on the coverage lower bound, the last term $-(1-\alpha) E_1[|r(X)-r_B(X)|]$ in the lower bound in Theorem 4.3 accounts for the fact that our guarantee is weaker if the $L^1$ distance from $r$ to $H$ is large. We hope this addresses the concern of how sensitive the coverage is to misspecification, and we will be happy to add a more detailed discussion to the paper.
>
> Finally, we would like to note that our experiments are consistent with our regularity conditions.
> We perform experiments involving very high-dimensional datasets:
> the Communities and Crime dataset has a 127-dimensional input,
> the RxRx1 dataset has a 512 pixel-by-512 pixel image input,
> and the MMLU dataset has a 1024-dimensional text input.
> In these challenging settings,
> LR-QR performs extremely well,
> which is consistent with the two regularity conditions discussed above.
> This underscores the fact that LR-QR is a practically significant algorithm
> in the high-dimensional setting,
> with strong and meaningful theoretical guarantees.
>
> Overall, we believe that our regularity conditions are practically reasonable, but we believe relaxing our conditions is an interesting future direction.
>
> Q1B. "How to choose lambda?"
>
> Response to Q1B:
> As the reviewer has noted, introducing the regularization term, which controlled by $\lambda$, is a crucial contribution of our work.
> Moreover, as the reviewer mentioned, in practice it can be quite conveniently tuned using cross-validation. In fact, we have carefully studied this problem in Section 5.1, and designed a bespoke criterion to cross-validate on (because standard accuracy does not work--- we do not assume access to labeled data from the target domain). Overall, the choice of $\lambda$ is an important issue, but we believe that our current approach handles it satisfactorily.
>
> Q2: "Clarifications regarding Section 3."
>
> Response to Q2:
> We appreciate the detailed suggestions regarding the exposition in Section 3.
> Our presentation was quite compact due to the page limit, but we are happy to expand it in the final version.
> Indeed, the two crucial ideas we seek to convey in Section 3 are
> (1) the connection between the first-order optimal condition and the marginal coverage in the test domain,
> as introduced by [1],
> and
> (2) the motivation for our novel regularization,
> via a data-dependent choice of the hypothesis class $H$.
> Here, we elaborate on (1) and lines 117-129,
> to clarify the key steps of the derivation.
> By inspecting the definition of the pinball loss in Equation (1),
> we see that the derivative of the pinball loss with respect to its first argument is given by
> $$\frac{d}{dc} \ell_{\alpha}(c, s)
> = -(1-\alpha) \mathbf{1}[s>c] + \alpha \mathbf{1}[s\le c]
> = \mathbf{1}[s\le c] - (1-\alpha).$$
> Combining this identity and the chain rule,
> we can calculate the directional derivative
> of $h\mapsto E_1[ \ell_{\alpha}(h, S) ]$ in the direction $g$
> to be
> $$ \frac{\partial}{\partial \epsilon} E_1[\ell_{\alpha}(h+\epsilon g, S)]
> = E_1\left[ \frac{\partial}{\partial \epsilon} (h+\epsilon g)
> \cdot \frac{\partial}{\partial h} \ell_{\alpha}(h, S) \right]
> = E_1[ g (\mathbf{1}[S\le h] - (1-\alpha)) ],$$
> where in the first step we interchanged derivative and expectation,
> applied the chain rule,
> and evaluated at $\epsilon=0$,
> and where in the second step we used the formula for the derivative of the pinball loss.
> When $g$ is the Radon--Nikoydm derivative $dP_{2,X}/dP_{1,X}$,
> then as the reviewer rightly observes,
> this allows us to apply the change-of-measure identity
> to the last expression to obtain $E_2[ \mathbf{1}[S\le h] - (1-\alpha) ]$,
> where $E_2$ now denotes expectation with respect to the test distribution.
> Finally, since the expectation of an indicator can be written as a probability,
> our directional derivative becomes the marginal test-time excess coverage
> $P_2[S\le h] - (1-\alpha)$, as claimed.
> We will update Section 3 with these comments in the revised paper.
>
> [1] Gibbs et al., "Conformal prediction with conditional guarantees".
>
> [2] van der Laan et al., "Generalized Venn and Venn-Abers Calibration with Applications in Conformal Prediction".
>
> [3] Bairaktari et al., "Kandinsky Conformal Prediction".
>
> [4] Cherian et al., "Large language model validity via enhanced conformal prediction methods".

---

### Official Review · Reviewer_im4Q · 2025-07-02

**Clarity:** 3
**Significance:** 3
**Originality:** 3
**Rating:** 4
**Confidence:** 3

**Summary:**

This paper introduces a conformal prediction method for the covariate shift setting (i.e. when the distribution of the $X$ in the test set is different from the distribution of the $X$ in the calibration set). Whereas existing methods require the estimation of a particular likelihood ratio function, the authors here present an algorithm, called likelihood ratio regularized quantile regression (LR-QR), which outputs a threshold function without directly estimating the likelihood ratio. To do this, they assume that unlabeled data from both distributions are available and minimize a particular objective function (Eq. in line 163). They show theoretically that when there is no estimation error (i.e. expectations are known) and under several assumptions, the function produced by the optimizer achieves valid coverage in the test domain. In the finite sample setting, the validity becomes asymptotic if the class of functions on which we optimize is sufficiently large. Finally, they show the effectiveness of their method on several data sets.

**Questions:**

1\ Is Proposition 4.1 true for any $\lambda$? If yes, I do not understand where is the impact of this hyper-parameter.

2\ Furthermore, if there is no impact, is the intuition being that in the infinite sample size setting, if the true $r$ is in $H$ then we do not need to regularized the objective as the optimizer is going to return $r$ anyway?

3\ Line 121: Why $h$ need to be in a linear hypothesis class H ?

4\ Do you think it could be possible to use the $X$ from $P_{1, X}$ of the calibration instead of assuming that we have another unlabeled set?

5\ In Eq (4), you provide an optimal value for $\lambda$. Is this value consistent with the one obtained across all the experiments? (not only Figure 4).

**Ethical Concerns:**

["NO or VERY MINOR ethics concerns only"]

**Final Justification:**

I was positive about this article. After the rebuttal and based on the other reviews, I am sticking with my rating.

**Limitations:**

yes

**Paper Formatting Concerns:**

everything is ok.

**Quality:**

3

**Strengths And Weaknesses:**

Strengths:

1) The paper is well-written and the problematic is well motivated.

2) The main idea is interesting and well-presented.

3) This line of research is of particular interest for the CP community.

Weaknesses:

1) The supplementary material does not include the code necessary to run the algorithm. Providing this code would improve reproducibility.

2)  The experimental section lacks clarity compared to the theoretical section (reason why I put a borderline accept but I am willing to increase it because overall this is a good paper).

     2.1) Section 5.2 "Communities and Crime": This section is unclear (at least for me), particularly regarding how the covariate shift is simulated.

    2.2) The results on the size for LR-QR appear to be missing.

    2.3) It is unclear where the results for the coverage of LR-QR are presented in Figure 1.

    2.4) There is no error bar in Table 1.

3) The conclusion only contains one limitation (and this limitation is not clear to me) --> Line 356 : "to have results that control of the slack in coverage" ??

---

> ### Author Rebuttal · Authors · 2025-07-27
>
> We thank the reviewer for the helpful comments.
>
> Weakness 1: "Providing code."
>
> Response to Weakness 1:
> Our code is currently ready to be shared;
> however, the conference does not allow us to share links,
> otherwise we would include it here.
> As a result, we will include our code with the revised paper.
> We thank the reviewer for their interest.
>
> Weakness 2: "Experimental details?"
>
> Response to Weakness 2:
>
> Regarding point (2.1),
> we thank the reviewer for allowing us to clarify the Communities and Crime experiment.
> Our description of the Communities and Crime experiment
> in Section 5.2 contained a typo, which we would like to correct here.
> In the experiment,
> we design four distinct covariate shifts,
> run the LR-QR algorithm in each of the four scenarios,
> and evaluate the marginal coverage in the test domain.
> To elaborate,
> let us assign the index $i = 1$ to the Black subgroup,
> $i=2$ to the White subgroup,
> $i=3$ to the Asian subgroup,
> and $i=4$ to the Hispanic subgroup.
> Let $Z$ denote the set of datapoints
> that were not used for training the regression model.
> Then, for each $i\in [4]$,
> we design a covariate shift between calibration and test
> as follows:
> the $i$-th calibration set $Z_{cal,i}$ includes
> all individuals in $Z$
> that are not of race $i$,
> and the $i$-th test set
> includes
> all individuals in $Z$
> that are of race $i$.
> Thus, for each $i\in [4]$,
> we induce a covariate shift
> between $Z_{cal,i}$ and $Z_{test,i}$,
> shifting probability mass from
> "not race $i$"
> to "race $i$".
> In Figure 1, we evaluate each algorithm on each of the four experimental setups.
> Our results show that on this challenging 127-dimensional task,
> the LR-QR algorithm achieves excellent marginal coverage at test-time,
> for each of the four distinct choices of covariate shift.
> We thank the reviewer for this question regarding the choice of covariate shift in Section 5.2.
> Due to space limitations, we were unable to elaborate in the initial submission.
>
> Regarding points (2.2) through (2.4), we thank the reviewer for catching the error in the legend;
> we will correct this in the revised version.
> In Figure 1, for each race, the left-most bar (in blue) denotes LR-QR.
> We are also happy to add error bars to Table 1 in the revised paper (they are quite small and do not change the interpretation of the results).
>
> Weakness 3: "Clarify the limitation?"
>
> Response to Weakness 3:
> In the limitation stated in the Discussion section,
> we mean that
> it would be of interest in future work
> to study the error terms appearing in the coverage lower bound
> in Theorem 4.3.
> In particular, it may be possible to understand these terms
> when optimizing over particular function classes,
> or under certain assumptions on the true likelihood ratio $r$.
> We believe this could shed further light on the practical efficacy
> of the LR-QR algorithm,
> and strengthen our theoretical guarantees.
>
> Q1 + Q2 - "Clarify effect of $\lambda$ in Proposition 4.1; clarify the infinite-sample setting?"
>
> Response to Q1 + Q2:
> Indeed, in the statement of Proposition 4.1, $\lambda$ does not appear in the lower bound of $(1-\alpha)$; however, in the proof, we show that the lower bound can be improved to
> $$\ge (1-\alpha) + 2\lambda \beta E_1[ (r_{H}(X) - \beta h(X))^2 ],$$
> where $(\beta, h)$ denotes the population LR-QR solution, which in turn depend on $\lambda$.
> This shows that, at the population level, $\lambda$ has an effect, which is however not straightforward to interpret---for instance, the term $\lambda \beta E_1[ (r_{H}(X) - \beta h(X))^2 ]$ above is of course non-monotone in $\lambda$ in general. For conciseness, we have omitted this discussion from the paper, but we are happy to add it.
>
> The question of whether the optimization problem returns the true $r$ is interesting but perhaps surprisingly subtle, as this does not seem to follow directly from our analysis. However, regardless of this question, if $r$ lies in $H$, then even in the case of zero regularization, the LR-QR threshold achieves valid test-time coverage. However, the solution of the optimization problem in this case is not $r$ itself, but rather the conditional quantile function (as the problem is unregularized quantile regression).
>
> Q3: "Why do we assume a linear class?"
>
> Response to Q3:
> On a technical level, in our theoretical results, we leverage the linearity of $H$
> to relate the first-order conditions of the LR-QR problem
> to the test-time marginal coverage under the covariate shift $r$.
> This linearity is also the key property leveraged by
> the pioneering work [2] in order to study conditional coverage.
> (For a detailed derivation of the connection
> between the first-order conditions of the LR-QR problem
> and the test-time marginal coverage,
> please see the start of Section 3
> or the proof of Proposition 4.1.)
>
> We would like to emphasize that a linear hypothesis class can be very rich and can contain highly nonlinear functions, e.g., the space of functions representable by linear combinations of the last-layer features of a pretrained model. Consequently, $r\in H$ does not imply that $r$ is linear, and we do not view the assumption that $r\in H$ as being restrictive in practice.
> Indeed, we perform experiments involving high-dimensional datasets:
> the Communities and Crime dataset has a 127-dimensional input,
> the RxRx1 dataset has a 512 pixel-by-512 pixel image input,
> and the MMLU dataset has a 1024-dimensional text input.
> In these challenging settings,
> LR-QR performs extremely well,
> which is consistent with our assumption that $r$ lies within $H$ or close to $H$.
> This underscores the fact that LR-QR is a practically significant algorithm
> in the high-dimensional setting,
> with strong and meaningful theoretical guarantees.
>
> Moreover, we would like to emphasize that such a linear function class setting is now quite standard and well-accepted in the conformal prediction literature (see e.g., [2, 3, 4]), starting with the pioneering work of [1], which has been widely adopted in the community (e.g., cited more than 100 times).
>
> Q4: "How to obtain unlabeled calibration data?"
>
> Response to Q4:
> Indeed, given datapoints $\{ (X_i, Y_i) : i\in [n_1+n_3] \}$
> drawn i.i.d. from $P_1$, we can split the data into
> a labeled calibration set
> $\{ (X_i, Y_i) : i\in [n_1] \}$
> and an unlabeled calibration set
> $\mathcal{S}_3 = \{ X_i : i\in [n_1] \}$
> by simply dropping the label $Y_i$ from $(X_i, Y_i)$ for $n_1+1 \le i \le n_3$.
>
> However, to achieve provable coverage guarantees, our current approach requires the two datasets to be indepenent.
> Thus, it is not possible to set the unlabeled calibration set $\mathcal{S}_3$
> to be equal to the features $\{ X_i : i\in [n_1] \}$
> of the labeled calibration data $\{ (X_i, Y_i) : i\in [n_1] \}$;
> if we reuse data in this way, then we violate the i.i.d. assumption that is necessary for our theoretical guarantees.
>
> Q5: "Is the theoretical $\lambda$ consistent with the experimental $\lambda$?"
>
> Response to Q5:
> This is a great question.
> As shown in Figure 4 of Section B of the Appendix,
> for the Communities and Crime task,
> the regularization strength selected through cross-validation
> aligns closely with
> the regularization strength suggested by our theory in Equation (4).
> In the other two experiments, we observe similar trends,
> where the selected regularization is consistent with our theory.
> Unfortunately, with the revised conference rules, we are unable to provide the resulting plots.
> We are happy to elaborate in the revised paper.
>
> [1] Gibbs et al., "Conformal prediction with conditional guarantees".
>
> [2] van der Laan et al., "Generalized Venn and Venn-Abers Calibration with Applications in Conformal Prediction".
>
> [3] Bairaktari et al., "Kandinsky Conformal Prediction".
>
> [4] Cherian et al., "Large language model validity via enhanced conformal prediction methods".

---

> > ### Comment · Reviewer_im4Q · 2025-08-05
> >
> > After the rebuttal, I remain positive about this paper. Thank you for the detailed responses.

---

### Official Review · Reviewer_KaEA · 2025-07-03

**Clarity:** 3
**Significance:** 3
**Originality:** 4
**Rating:** 5
**Confidence:** 4

**Summary:**

This manuscript considers conformal prediction under high-dimensional covariate shift. The
authors introduce the Likelihood Ratio-Regularized Quantile Regression (LR-QR) algorithm. By
combining the pinball loss with a suitable choice of regularization, the authors construct a threshold
function by getting around direct estimation of the unknown likelihood ratio, which can be difficult
in high-dimensional setting. The authors provide theoretical properties and experiments to back up
their proposed method.

**Questions:**

\textbf{Questions:}

\begin{itemize}
\item Key symbols $\mathbb{P}_1$ and $\mathbb{P}_2$ are not consistently defined or used, Their meanings and usage vary between Sections 1 and 2.

    1. Lines 25–26: The statement ``In the case that the calibration and test distributions coincide ($\mathbb{P}_1 = \mathbb{P}_2$), ..."
    does not align with the definition of $\mathbb{P}_1$ and $\mathbb{P}_2$:  $\mathbb{P}_1$ is defined over both inputs and outputs,
while $\mathbb{P}_2$ is defined only over inputs (Lines 21–22).

    2. Lines 36–37: In the definition of the likelihood ratio function $r(x) = d\mathbb{P}_{2,X}/d\mathbb{P}_{1,X}$, the
distributions  $P_{2,X}$ and $P_{1,X}$ are not clearly defined in Section 1,
 though they appear more explicitly (yet still implicitly) in Section 2.

\item Line 100: The prediction set $C: \mathcal{X} \rightarrow 2^{\mathcal{Y}}$ suggests that the output space $\mathcal{Y}$ is discrete (e.g., for classification). If the method is intended for regression,
 the terminology "prediction set” could be adjusted to `"prediction interval/region,”
or the authors should clarify that the method is focused on classification.

\item Lines 105–106: It states, ``In this paper, a linear hypothesis class refers to a linear subspace of functions
from $\mathcal{X} \rightarrow \mathbb{R}$ that are square-integrable with respect to $\mathbb{P}_{1,X}$.”
Why is this class particularly considered? Provide motivation or justification.

\item Lines 111–113: Calibration data are assumed to be i.i.d., which may be unnecessarily restrictive. One strength of conformal prediction is its reliance on weaker assumptions such as exchangeability. Can the method be extended to exchangeable data? If not, explain why the i.i.d. assumption is essential.

\item Notation inconsistency: The function $C(x)$ is defined differently in Lines 100 and 118 using different threshold symbols $q(x)$ and $\tilde{q}(x)$. This variation seems unnecessary. Define $C(x)$ once and refer to it consistently.

\item Line 170: The statement that valid coverage is guaranteed if ${\cal H}$ is linear and contains the true likelihood ratio $r$ implicitly requires $r$ to be linear. How plausible is this in real-world settings? Provide discussion or examples to help evaluate the adequacy of restricting ${\cal H}$ to linear functions.
    How likely would
 this happen for distributions often used in applications?
    Provide discussion or examples to help evaluate the adequacy of restricting ${\cal H}$ to linear functions. While it is understood that focusing on
 the linear hypothesis class ${\cal H}$ can be critical in establishing meaningful theoretical results, a natural concern is its feasibility or adequacy in handling various practical distributions.

\item Lines 181–183: The lower bound involving $-E_1[(r(X) - r_{\cal H}(X))^2]^{1/2}$ is derived via the Cauchy–Schwarz inequality. Given the indicator function’s $\{0,1\}$ values, could a tighter bound, such as $\max(0, E_1[r(X) - r_{\cal H}(X)])$, be used to describe the lower bound in Line 185?

\item Lines 212–213: Why is a fixed collection $\Phi = (\phi_1, \ldots, \phi_d)^\top$ of $d$ basis functions assumed? What is $d$? Is it related to $\dim(X)$?
    (The dimension of $X$ does not appear to be explicitly represented using a dedicated notation.)
    How are these basis functions chosen?

\item Lines 212–214: The introduction of interval ${\cal I}$ appears to restrict the domain of the minimization in (LR-QR) from $\beta \in \mathbb{R}$ to a compact subinterval of positive values. Justify this restriction and explain its impact on the solution.

\item Line 214: The $B$-ball ${\cal H}_B$ defined in Line 214 depends on the choice of $B$, which in turn may influence the solution $(\hat h, \hat \beta)$ in (3). However, the  results in Theorems 4.2 and 4.3 do not explicitly depend on $B$. Is this discrepancy  meaningful? The lack of $B$ in the bounds  makes to statements in Lines 239-240 less meaningful, though one may intuitively understand the impact of varying the value of $B$.

 \item Line 233: What is $r_B(X)$ in Theorem 4.3? It is used without prior definition.

   \end{itemize}




\textbf{Suggestions:}

\begin{itemize}

\item The example in Lines 106–110 does not seem to be helpful. It can either be removed or further explained in terms of its relevance to the subsequent development.
The description of the “linear hypothesis class” in Lines 105–106 could be moved to the location around Line 122, where the linear hypothesis class ${\cal H}$ first appears.

\item Clearly describe the key notations  $E_1$
and $E_2$ in relation to
the distributions $P_{1,X}$
and $P_{2,X}$
(or even $P_1$
and $P_2$). This clarification will help align the use
of $E_{\mathbb{Q}}$ in Line 126 with the other notations.
By the way, $P$ and $Q$ in footnote 1 on page 3 should be written as $\mathbb{P}$ and
$\mathbb{Q}$, respectively.

\item The equation labels “(LR-QR)” in Line 157 and “(Empirical-LR-QR)” in Line 163
should be changed to “(1)” and “(2)” to be consistent with numerical labels such as
(3) in Line 217 and (4) in Line 226 (though authors might intend to use lettered names
to emphasize the differences in those minimization problems).

\item In Lines 171–172, it would be helpful to also present the mathematical expression
of the projection of $r$ onto ${\cal H}$ in the Hilbert space.

\item In Line 119, use a symbol different from $S$ to represent a random variable
to avoid confusion with the use of $S(\cdot, \cdot)$ as the nonconformity score defined in Line 98.

\end{itemize}

(NOTE: For unknown reasons, some LaTeX symbols cannot be displayed properly  in the OpenReview display system, even though they appear properly in my locally compiled PDF).

**Ethical Concerns:**

["NO or VERY MINOR ethics concerns only"]

**Final Justification:**

I thank the authors for addressing my comments and suggestions on the initial submission. The rebuttal and the planned revisions are satisfactory, and I am pleased to raise my initial rating to 5.

**Limitations:**

Yes

**Quality:**

3

**Strengths And Weaknesses:**

Strengths:
1. Conformal prediction under covariate shift is an interesting problem that is of practical
importance. This manuscript proposes a promising method to handle such a problem.
2, The proposed approach offers a natural and effective way to incorporate regularization techniques
in situations where the likelihood ratio function is unknown. The underlying idea is both novel
and well-justified.
3. Theoretical results are provided to support the proposed methodology, and they are nontrivial
to establish.
4. The manuscript is generally well written, and the presentation follows a logical progression.

Weaknesses:
1. While the title and main text emphasize the consideration of ’high-dimensional covariate
(shift),’ the manuscript lacks a clear discussion of this feature. How high is the dimension of
the covariate X? Is it treated fixed or diverging with the data size? Relative to the sizes $n_1$, $n_2$,
and $n_3$ for labeled or unlabeled source data points and target data points, how large is the
dimension of the covariate? How do different dimensionality settings influence the theoretical
developments, including the established bounds, regularity conditions, and proofs?
2. The presentation can be improved with careful revision. Several notations are used inconsistently
or unclearly, and further clarification is needed in several places (see specific comments below)

---

> ### Author Rebuttal · Authors · 2025-07-27
>
> We thank the reviewer for the helpful comments.
>
> Weakness 1: "Elaborate on high dimensionality."
>
> Response to Weakness 1:
> This is a great question and merits a detailed answer. The reason why we have not already expanded on this in detail in the submission is due to space limitations; because it turns out that the characterization of allowed dimensionalities is quite subtle, as we explain next.
>
> Our theoretical results are meaningful as long as the dimension $d$ of the function class does not grow too rapidly with $n_1$, $n_2$, and $n_3$. Specifically, in Theorems 4.2 and 4.3, $d$ appears implicitly in the constants, e.g., $c$, $c'$, $c''$, $A$, and $A'$. Explicitly, we have $A = A_{10}$ and $A' = A_{14}$, where the quantities $A_{10}$ and $A_{14}$ are defined in Appendix F. Although the exact formulae for $A$ and $A'$ are rather complicated, both $A$ and $A'$ depend only polynomially and inverse-polynomially on the following quantities: $\lambda_{min}(\Sigma)$, $\lambda_{max}(\Sigma)$, $c_{min}$, $c_{max}$, $B$, $C_{\Phi}$, $C_{1,max}$, and $C_{1,upper}$, as defined in Appendix E.
>
> The first four quantities depend on the choice of basis $\Phi$ only through the smallest and largest eigenvalues of the population and sample covariance matrices $\Sigma$ and $\hat{\Sigma}$, which entails a mild dependence on $d$.
> The radius $B$ can be taken to be a constant.
> The quantity $C_{\Phi}$ can be taken to be a constant if the basis functions are normalized correctly.
> The quantities $C_{1,max}$ and $C_{1,upper}$ depend polynomially on the preceding six quantities.
> Thus, $A$ and $A'$ introduce only mild dependence on the complexity of the function class.
>
> To illustrate the dependence on the dimension $d$ of the function class, suppose $\Phi$ forms an orthonormal basis for $P_1$. Then applying the empirical sample covariance tail bound from Chapter 5.6 of [1], we have $\lambda_{min}(\hat \Sigma) \ge 1 - O(d \log(dn_3)/n_3)$ with probability $1-o(n_3^{-1})$. This is of constant order if $d \log(d) \ll n_3$. So our bounds are meaningful if $d \ll n_3/\log(n_3)$, which allows a growing dimension.
>
> Also, we would like to point out that in our experiments, $d = \dim(H)$ is typically smaller than the dimension of the original input; for instance, in our pre-trained LLM experiment with GPT-2 Small, the context length (input dimension) is 1024, while the feature dimension is 768.
>
> We would like to emphasize again that our key contribution is the development of a method that performs well empirically and has theoretical guarantees in such a high-dimensional case, which is not available in prior work.
>
> Q1 + Q2 + Q5: Notation.
>
> Response to Q1 + Q2 + Q5: We thank the reviewer for the detailed comments. Indeed, LR-QR applies to both regression and classification tasks. We will fix this in the revision.
>
> Q3: "Why a linear hypothesis class?"
>
> Response to Q3:
> In our setting, we call a set of functions $H$
> from $\mathcal{X}\to \mathbb{R}$
> a "linear hypothesis class" if
> (1) $H$ forms a finite-dimensional vector space
> and
> (2) for each function $h\in H$, the second moment $E_1[|h(X)|^2]$ is finite.
> On a technical level, in our theoretical results, we leverage the linearity of $H$
> to relate the first-order conditions of the LR-QR problem
> to the test-time marginal coverage under the covariate shift $r$.
> This linearity is also the key property leveraged by
> the pioneering work [2] in order to study conditional coverage.
> (For a detailed derivation of the connection
> between the first-order conditions of the LR-QR problem
> and the test-time marginal coverage,
> please see the start of Section 3
> or the proof of Proposition 4.1.)
>
> We would like to emphasize that a linear hypothesis class can be very rich and can contain highly nonlinear functions, e.g., the space of functions representable by linear combinations of the last-layer features of a pretrained model. This is the reason for the example in L105 (see also answer to the suggestion). Consequently, $r\in H$ does not imply that $r$ is linear, and we do not view the assumption that $r\in H$ as being restrictive in practice.
> Indeed, we perform experiments involving high-dimensional datasets:
> the Communities and Crime dataset has a 127-dimensional input,
> the RxRx1 dataset has a 512 pixel-by-512 pixel image input,
> and the MMLU dataset has a 1024-dimensional text input.
> In these challenging settings,
> LR-QR performs extremely well,
> which is consistent with our assumption that $r$ lies within $H$ or close to $H$.
> This underscores the fact that LR-QR is a practically significant algorithm
> in the high-dimensional setting,
> with strong and meaningful theoretical guarantees.
>
> Moreover, we would like to emphasize that such a linear function class setting is now quite standard and well-accepted in the conformal prediction literature (see e.g., [3, 4, 5]), starting with the pioneering work of [2], which has been widely adopted in the community (e.g., cited more than 100 times).
>
> Q4: "Why i.i.d.?"
>
> Response to Q4:
> In the proof of our generalization bound in Theorem 4.2, we use the i.i.d. assumption to (1)  apply the stability result Lemma G.1 and (2) to apply Hoeffding's inequality to control empirical process terms that arise (e.g., Terms (III) and (IV) on page 25). At present, it is not clear how to extend the analysis to the exchangeable setting, but relaxing the i.i.d. condition is an interesting direction for further work.
>
> Q6.A: "Is $r$ linear?"
>
> Response to Q6.A:
> We would like to clarify that a linear hypothesis class can be very rich and contain highly nonlinear functions, e.g., the space of functions representable by a pretrained model with a scalar read-out layer. Consequently, $r\in H$ does not imply $r$ is linear.
>
> Q6.B: "How plausible is $r\in H$? Include examples."
>
> Response to Q6.B:
> As discussed in the response to Q3 and above, a linear hypothesis class with non-linear features can include non-linear functions. In many applications involving conformal prediction, one leverages pre-trained predictors (e.g., neural networks) to compute appropriate score functions and use split conformal calibration. In such settings, we can also leverage the associated features used in the pre-trained models, e.g., last-layer features of a neural net. Whenever such features explain the covariate shift (i.e., are good for classifying the source vs target domains), our approach is quite likely to perform well.
>
> Q7: "Strengthen the misspecification lower bound?"
>
> Response to Q7:
> We thank the reviewer for the suggestion. Indeed, since $\mathbf{1}[S(X,Y)\le h^*(X)]$ only takes on the values $0$ and $1$, we have
> $$ E_1[(r(X) - r_{H}(X)) \mathbf{1}[S(X,Y)\le h(X)] ]
> \le E_1[\max(0, r(X) - r_{H}(X))],$$
> hence we can improve the coverage lower bound on line 185 to
> $$P_2[Y\in C(X)]\ge (1-\alpha) - E_1[\max(0, r(X) - r_{H}(X))].$$
> We will add this in the revised paper.
>
> Q8: "How to specify $\Phi$?"
>
> Response to Q8:
> In our theoretical results, the functions $\Phi = (\phi_1,\ldots,\phi_d)^\top$ form a basis for the linear hypothesis class $H$,
> so $d = \text{dim}(H)$ is the dimension of the feature space.
> In practice, we do not need to explicitly write out $\Phi$. We simply choose a set of features and perform gradient descent over it. Our method returns the same results as long as we have specified a fixed linear space, regardless of the parametrization (so we do not need to find an orthogonal basis $\Phi$).
>
> The dimension of $H$ is an algorithmic choice (e.g., how many pretrained features) and is independent of the dimensionality $\text{dim}(\mathcal{X})$ of the original input space.
>
> The reason why we have not introduced a dedicated notation for $\dim(\mathcal{X})$ is that it is never used anywhere. Only $\dim(H)$ affects the optimal regularization strength $\lambda^*$ for the LR-QR algorithm.  We will clarify this in the revision.
>
> Q9 + Q10: "Justify the restrictions $\beta\in \mathcal{I}$ and $\|h\| \le B$ in LR-QR."
>
> Response to Q9 + Q10:
> In Lemma L.4 of the appendix, we show that the unconstrained population solution lies within the region $\mathcal{I}\times H_B$, so at the population level, one loses nothing by restricting to $(\beta, h) \in \mathcal{I}\times H_B$. This justifies the restriction to $B$ in our algorithm.
>
> Regarding the dependence of the bounds in Theorems 4.2 and 4.3 on $B$, $B$ appears in the quantities
> $c$, $c'$, $c''$, $A$, and $A'$ (we will make this clear in the revision). We provide explicit formulae for these quantities in the proofs of Theorems 4.2 and 4.3, in terms of the quantities defined in Section F of the Appendix. It can be seen that these quantities depend only polynomially on $B$. We will add a detailed discussion to the final version.
>
> Q11: "Definition of $r_B(X)$?"
>
> Response to Q11:
> We define $r_B$ on lines 229-230 prior the theorem; $r_B$ denotes the orthogonal projection of $r$ onto the closed convex set $H_B$ in the Hilbert space induced by the inner product $\langle f,g\rangle = E_1[fg]$. We will link to the definition in the revision.
>
> Suggestions: Thank you! We will make all the changes.
>
> Response to S1: We thank the reviewer for this suggestion, we can move this to later. The reason for introducing it was to explain how linear spaces can cover non-linear functions in ML, and thus to motivate our approach.
>
> Response to S2: $E_i$ are defined on line 95. We will fix the footnote in the revision.
>
> Response to S3: As you suggested, we used these labels for added emphasis; we hope this is alright.
>
> Response to S4: We will add this in the revision.
>
> Response to S5: We will add this in the revision.
>
> [1] Vershynin. "High-dimensional probability".
>
> [2] Gibbs et al., "Conformal prediction with conditional guarantees".
>
> [3] van der Laan et al., "Generalized Venn and Venn-Abers Calibration with Applications in Conformal Prediction".
>
> [4] Bairaktari et al., "Kandinsky Conformal Prediction".
>
> [5] Cherian et al., "Large language model validity via enhanced conformal prediction methods".

---

> ### Comment · Reviewer_KaEA · 2025-08-04
>
> I thank the authors for addressing my comments and suggestions on the initial submission. The rebuttal and the planned revisions are satisfactory, and I am pleased to raise my initial rating to 5.

---

### Note · Authors · 2025-08-16

We thank the AC and the reviewers for their service and helpful suggestions. We have made these changes and believe they have significantly improved our paper.

We were pleased to see the reviewers’ strong interest in LR-QR. Reviewers 8HeH, wABN, im4Q, and KaEA expressed their support, as evidenced by their scores of 4 and 5, and by their positive comments. Reviewers KaEA and im4Q highlight that LR-QR fills a crucial gap in the conformal prediction literature; previous algorithms for covariate shift suffer from the curse of dimensionality. Reviewer 8HeH emphasizes our novel theoretical results, which introduce new techniques to the conformal prediction literature. Reviewer wABN is positive about the strong empirical performance of LR-QR in our three experiments, noting that the results align with the theory.

We also clarified technical points in our discussions. With Reviewers KaEA and 8HeH, we clarified our theory, including the mild dimension-dependence of our coverage lower bound in Theorem 4.3, and how to extend our results to unbounded scores. With Reviewers wABN and im4Q, we clarified the notion of a linear class, explaining how it connects the LR-QR first-order condition to the test-time coverage. We emphasized that linear classes are widely used in practice, and can be highly expressive. We also clarified our experiments, including our Communities and Crime results, and how we chose the regularization strength.

Reviewer KYG7 had two concerns with our work: (1) why estimation is hard in high dimensions, and (2) whether one can obtain finite-sample coverage guarantees in our setting. Regarding (2), we pointed to existing impossibility results in the conformal prediction literature. Regarding (1), we noted that the curse of dimensionality in high-dimensional estimation is well-documented in the theoretical literature, and we bolstered this claim with two new simulations. First, we showed that if one fits a logistic regression classifier to samples from two gaussians, then the resulting likelihood ratio estimator incurs pointwise error that increases rapidly with dimension. Then, in a simple regression setting with covariate shift, we ran LR-QR with an affine hypothesis class to show that LR-QR achieves valid test-time coverage without directly estimating the likelihood ratio. We hope this explained our motivation and the practical significance of LR-QR, especially in high dimensions.

We thank all for their help in the reviewing process.

---

### Decision · Program_Chairs · 2025-09-17

**Decision:**

Accept (poster)

**Comment:**

All reviewers agree that this submission addresses an important problem in theory and practice, namely conformal inference with covariate shift, without separate explicit estimation of the probability change measure. The contributions are theoretical and methodological. The paper is clearly presented and well-written. The discussion after the rebuttal allowed to clarify several points and led several reviewers to raise their rating. Clear accept.